# From Identifiable Causal Representations to Controllable Counterfactual Generation: A Survey on Causal Generative Modeling

**Aneesh Komanduri** *akomandu@uark.edu*
*Department of Electrical Engineering and Computer Science*
*University of Arkansas*

**Xintao Wu** *xintaowu@uark.edu*
*Department of Electrical Engineering and Computer Science*
*University of Arkansas*

**Yongkai Wu** *yongkaw@clemson.edu*
*Department of Electrical and Computer Engineering*
*Clemson University*

**Feng Chen** *feng.chen@utdallas.edu*
*Department of Computer Science*
*University of Texas at Dallas*

**Reviewed on OpenReview:** *https://openreview.net/forum?id=PUpZXvNqmb*

## Abstract

Deep generative models have shown tremendous capability in data density estimation and data generation from finite samples. While these models have shown impressive performance by learning correlations among features in the data, some fundamental shortcomings are their lack of explainability, tendency to induce spurious correlations, and poor out-of-distribution extrapolation. To remedy such challenges, recent work has proposed a shift toward causal generative models. Causal models offer several beneficial properties to deep generative models, such as distribution shift robustness, fairness, and interpretability. Structural causal models (SCMs) describe data-generating processes and model complex causal relationships and mechanisms among variables in a system. Thus, SCMs can naturally be combined with deep generative models. We provide a technical survey on causal generative modeling categorized into causal representation learning and controllable counterfactual generation methods. We focus on fundamental theory, methodology, drawbacks, datasets, and metrics. Then, we cover applications of causal generative models in fairness, privacy, out-of-distribution generalization, precision medicine, and biological sciences. Lastly, we discuss open problems and fruitful research directions for future work in the field.

## 1 Introduction

With the growing interest in using deep learning for domain-specific tasks in the natural and medical sciences, deep generative models have become a popular area of research. Deep generative models are a class of methods that either explicitly estimate the probability density or implicitly learn to sample from some data distribution. Such models have demonstrated impressive performance, especially in generating high-quality image samples. State-of-the-art deep learning models such as GPT-4 (OpenAI, 2023) and CLIP (Radford et al., 2021) have taken the advances in deep generative modeling to new heights. Recent research efforts in generative modeling focus on important properties of the data-generating process that lead to interpretable

representations and controllable generation (Locatello et al., 2019b). Despite such advances, most current methods neglect the fact that systems in the real world are complex, dynamic, and may have underlying causal dependencies. The failure to account for this has made deep learning models prone to shortcut learning and spurious associations (Arjovsky et al., 2020).

Many have argued that we must go beyond learning mere statistical correlations toward *causal models* that capture the causal-effect relationship between influential variables in a system. In the 1950s, Alan Turing asked what it would take for a computer to think like a human (Turing, 1950). Turing proposed what many today refer to as the "Turing Test." However, a more practical version of this test would be to ask the following question: how can machines represent causal knowledge of the world in a way that would enable them to access the necessary information quickly, answer questions accurately, and do so with ease (Pearl & Mackenzie, 2018)? Pearl's Causal Hierarchy (Pearl, 2019; Bareinboim et al., 2022) lays out the three levels of causal models that provide a mechanism to progressively get closer to modeling human-level reasoning: observations (seeing), interventions (doing), and counterfactuals (imagining). Intervening on causal variables of a system leads to the ability to reason about hypothetical scenarios in the world (counterfactuals).

Deep generative models such as variational autoencoders (VAE), generative adversarial networks (GANs), normalizing flows, and diffusion models have been at the forefront of machine learning for approximating complex data distributions. A traditional deep generative model would perform quite well in generating new data for an application such as generating novel images from brain Magnetic Resonance Imaging (MRI) data. However, understanding *what* and *how* underlying factors truly influence the generative process can only be captured by modeling the causal mechanisms between variables. If certain markers or labeled factors causally influence cancer diagnoses, incorporating them into a causal model will more robustly encode the system's information and its complex relationships. If we want to know the effect of changing certain factors on the structure of the brain MRI, we can easily perform interventions to infer such relationships. Further, one could "simulate" patients by generating counterfactual instances of tumors under different interventions that lie outside the support of the training data. This would greatly reduce the cost associated with collecting and annotating patient data with constraints such as privacy. Thus, a shift towards causal modeling is imperative to learn more robust models that reflect the complexities in the real world.

To improve the interpretability and robustness of generative models, recent research has focused on learning *causal* generative models using Pearl's structural causal model (SCM) formalism (Pearl, 2009). The SCM inherently describes a data-generating process consisting of a set of variables that influence each other through causal mechanisms. The SCM describes a generative model of the world and, as such, can be naturally utilized in deep generative modeling. Recent work has shown the effectiveness of neural models in parameterizing causal mechanisms between variables (Kocaoglu et al., 2018; Xia et al., 2021). Causal models offer several beneficial properties to deep generative models, such as distribution shift robustness, fairness, and interpretability (Scholkopf et al., 2021). The interpretability of models can be viewed in terms of both learning a semantically meaningful representation of a system and controllably generating realistic counterfactual data. In the context of representation learning, SCMs can be used to model the causal dependencies among encoded latent variables. In the context of controllable generation, SCMs can be used to model the mechanism that generates high-dimensional observational data as a function of causal variables to enable counterfactual inference.

We take this opportunity to survey the exciting and ever-growing field of causal generative models based on the Pearlian theory of structural causal models, discuss the benefits and drawbacks of different approaches, and highlight open problems and applications of causal generative models in a variety of domains for real-world impact. Unlike other surveys (Kaddour et al., 2022; Zhou et al., 2023), in addition to covering the methodology of causal generative models, we cover foundational identifiability theory in causal representation learning with a careful emphasis on intuition. We also outline several challenging open problems from the perspective of both application and theory. We focus our survey on generative models that utilize fundamental principles of causality to enable causal representation learning (CRL) and controllable counterfactual generation (CCG). We do not cover causality-inspired frameworks or causal discovery in this survey. Although other causal frameworks exist, such as Potential Outcomes (PO) (Rubin, 2005), we note that most methods in CRL and CCG utilize the SCM framework since they are concerned with explicitly modeling mechanisms among causal variables.

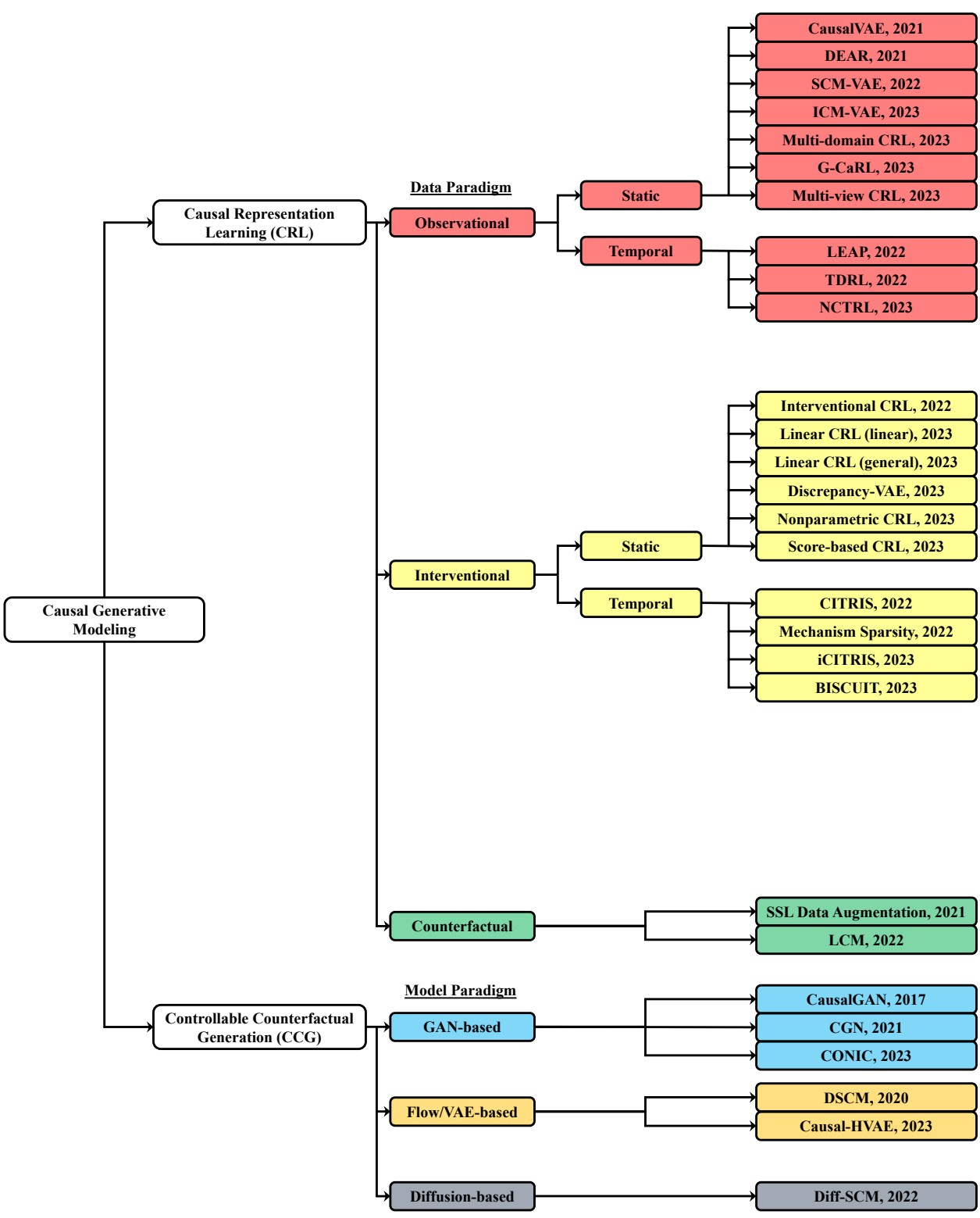

Figure 1: Taxonomy of Causal Generative Modeling

Causal representation learning (CRL) is concerned with learning semantically meaningful causally related latent variables and their causal structure from high dimensional data. We break down causal representation learning into the three different classes from Pearl's causal hierarchy based on the assumptions on the data-generating process to achieve identifiability. That is, one assumes access to either purely observational data, interventional data, or counterfactual data. Identifiability is an important property of representations learned by generative models since it ensures that we can provably recover the true generative factors. This is important for representation learning since this theoretical guarantee implies the model enjoys robustness and generalizability properties. Achieving identifiability in causal representation learning also enables isolated interventions on causal variables to observe causal effects downstream, which is often of great interest in domain-specific applications. Controllable counterfactual generation (CCG) methods focus more on modeling known causal variables (such as labels) and learning a mapping to the observed data. We break down controllable counterfactual generation methods by GAN-based, Flow/VAE-based, and Diffusion-based models. One of the main assumptions in CCG methods is access to causal variables in the form of labels, which makes modeling the relationships between causal variables and high-dimensional data convenient, efficient, and flexible. However, in causal representation learning, the causal variables and their structure are often unobserved and we must disentangle the factors before being able to generate counterfactual images. We emphasize that CRL methods are also capable of controllable generation through latent space manipulations and decoding. However, the focus of CRL is primarily on learning robust and interpretable representations for downstream tasks. Our survey focuses on feature-level causal variables and not on more general causal modeling schemes. The taxonomy of methods is shown in Figure 1.

**Contributions.** (1) We taxonomize causal generative modeling work into the identifiable causal representation learning and controllable counterfactual generation tasks. (2) We further categorize causal representation learning methods according to the type of data they assume access to from Pearl's Causal Hierarchy and break down controllable generation approaches based on the type of generative model utilized. (3) We discuss the existing evaluation metrics, benchmark datasets, and results of causal generative models. (4) Finally, we lay out current limitations and fruitful research directions in both causal representation learning and controllable counterfactual generation.

**Structure of survey.** The survey is structured as follows. In Section 2, we discuss the background necessary to understand the rest of the paper, including formal definitions of SCM, common deep generative models, and disentangled representation learning. In Section 3, we formulate the causal representation learning problem and discuss methods categorized according to the type of data they assume access to from Pearl's Causal Hierarchy. Then, in Section 4, we discuss the controllable counterfactual generation task and methods from GAN-based, Flow/VAE-based, and Diffusion-based frameworks. In Section 5, we give an overview of the most common metrics used in causal generative modeling. In Section 6, we discuss synthetic and real-world datasets that are often used in causal representation learning and controllable generation tasks. In Section 7, we highlight applications of causal generative models in trustworthy and robust AI, precision medicine, and biology. In Section 8, we outline several open research directions in causal generative modeling. Finally, we conclude the survey in Section 9.

## 2 Background

In this section, we introduce notation and key concepts in generative modeling and representation learning important for understanding the rest of the paper, including Pearl's structural causal model, variational autoencoders, normalizing flows, generative adversarial networks, diffusion probabilistic models, and disentangled representation learning. A list of the main notation used in the paper is provided in Table 1.

### 2.1 Structural Causal Model

All the methods discussed in this survey utilize the structural causal model (SCM) formalism from Pearl (2009), which formally describes the relationship among a set of variables. The SCM induces a hierarchy of causal reasoning referred to as *Pearl's Causal Hierarchy.*

Table 1: Main notation used for causal representation learning

| Notation | Description |
| --- | --- |
| $x$ | observed data |
| $z$ | endogenous latent causal variables |
| $\epsilon$ | exogenous noise variables |
| $y$ | labels (domain, class, time etc.) |
| $\mathcal{X} \subseteq \mathbb{R}^d$ | domain of observational data |
| $\mathcal{Z} \subseteq \mathbb{R}^n$ | domain of causal variables |
| $\mathcal{E} \subseteq \mathbb{R}^n$ | domain of noise variables |
| $\mathcal{Y}$ | domain of labels |
| $A$ | causal DAG adjacency matrix |
| $z_{\mathbf{pa}_i}$ | causal parents of variable $z_i$ |
| $f_i$ | causal mechanism deriving variable $z_i$ |
| $f_{RF}$ | reduced form mapping noise to causal variables |
| $\tilde{f}_i$ | new mechanism replacing $f_i$ after intervention |
| $F$ | set of all causal mechanisms |
| $\tilde{x}$ | counterfactual data |
| $\tilde{z}$ | post-intervention causal representation |
| $\tilde{\epsilon}$ | post-intervention noise variables |
| $\mathcal{C}$ | structural causal model |
| $\tilde{\mathcal{C}}$ | intervened structural causal model |
| $\mathbf{do}(\cdot)$ | perfect hard intervention |
| $g(\cdot)$ | mixing function |
| $p(z)$ (or $p_z$)[1] | joint distribution over set of variables $z$ |
| $\circ$ | composition of functions |
| $\odot$ | Hadamard product |

[1] *Note that the notations $p(z)$ and $p_z$ are used interchangeably throughout the survey and are equivalent*

**Definition 1 (Structural Causal Model)** *A structural causal model (SCM) is defined by a tuple $\mathcal{C} = \langle \mathcal{Z}, \mathcal{E}, F, p_{\mathcal{E}} \rangle$, where $\mathcal{Z}$ is the domain of $n$ endogenous causal variables $z = \{z_1, \ldots, z_n\}$, $\mathcal{E}$ is the domain of $n$ exogenous noise variables $\epsilon = \{\epsilon_1, \ldots, \epsilon_n\}$, and $F = \{f_1, \ldots, f_n\}$ is a collection of $n$ causal mechanisms of the form*

$$z_i = f_i(\epsilon_i, z_{\mathbf{pa}_i}) \tag{1}$$

*where $\forall i$, $f_i : \mathcal{E}_i \times \prod_{j \in \mathbf{pa}_i} \mathcal{Z}_j \to \mathcal{Z}_i$ are **causal mechanisms** that determine each causal variable as a function of the parents and noise, $z_{\mathbf{pa}_i}$ are the parents of causal variable $z_i$; and $p_{\mathcal{E}}(\epsilon)$ is a probability measure of $\mathcal{E}$.*

An SCM where the exogenous noise variables are jointly independent (no hidden confounders) is known as a Markovian model. The assumption of no hidden confounding is also referred to as *causal sufficiency*. We depict the causal structure of $z$ by a causal directed acyclic graph (DAG) $\mathcal{G}$ with adjacency matrix $A \in \{0, 1\}^{n \times n}$. A 2-variable SCM is shown in Figure 2a. The distribution over endogenous variables $p^{\mathcal{C}}(z)$ is called the entailed distribution. We have that the joint distribution of endogenous variables can be written as the following *causal Markov factorization*

$$p^{\mathcal{C}}(z) = \prod_{i=1}^{n} p^{\mathcal{C}}(z_i | z_{\mathbf{pa}_i}) \tag{2}$$

**Interventions.** An SCM enables one to go beyond observational distributions to construct interventional distributions that reflect changes in a system at the population level. Interventions enable one to answer the question: "What would $z_j$ be if we change $z_i$?" We can construct interventional distributions by making modifications to an SCM $\mathcal{C}$ and considering the entailed distribution.

Let $\mathcal{C} = \langle \mathcal{Z}, \mathcal{E}, F, p_{\mathcal{E}} \rangle$ be an SCM and $p^{\mathcal{C}}(z)$ be its entailed distribution. We replace any number of structural assignments to obtain a new SCM $\tilde{\mathcal{C}}$. Suppose we replace the mechanism that generates causal variable $z_i$

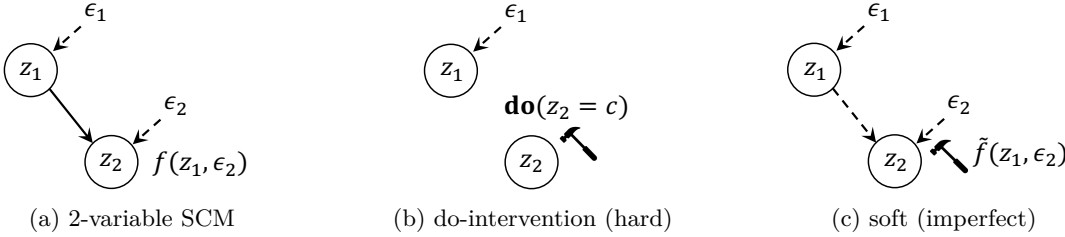

(a) 2-variable SCM          (b) do-intervention (hard)          (c) soft (imperfect)

Figure 2: Types of Interventions

by

$$z_i = \tilde{f}_i(z_{\tilde{\mathbf{pa}}_i}, \tilde{\epsilon}_i) \tag{3}$$

We call the entailed distribution of the new SCM an interventional distribution, which is denoted by

$$p^{\tilde{\mathcal{C}}}(z) = p^{\mathcal{C};do(z_i=\tilde{f}(z_{\tilde{\mathbf{pa}}_i}, \tilde{\epsilon}_i))}(z) \tag{4}$$

The two main types of interventions are *hard* interventions, which remove dependency from parents, and *soft* interventions, which alter the dependency on direct parents, as depicted in Figure 2.

- **Hard intervention.** A *hard* intervention, or *do-intervention*, modifies an SCM by deterministically replacing for a subset of causal variables the variable $z_i$ with a constant $c$, denoted $\mathbf{do}(z_j = c)$, such that the causal variable is no longer dependent on its parents. The interventional distribution given a hard intervention on variable $z_i$ can be expressed as:

$$p(z_1, \ldots, z_n | \mathbf{do}(z_i = c)) = \prod_{j \neq i} p(z_j | z_{\mathbf{pa}_j}) \delta_{z_i = c} \tag{5}$$

- **Soft intervention.** Unlike hard interventions, which remove all direct causal edges of the intervened target, a *soft* intervention, or *mechanism change*, simply changes the dependency of the causal variable on its direct parents, where the variable can still be dependent on a subset of the original parents.

We introduce two additional subclasses of interventions that are of special interest in the causal generative modeling literature: perfect and imperfect interventions (Peters et al., 2017).

- **Perfect intervention.** A *perfect* intervention modifies an SCM by replacing for a subset of causal variables the causal mechanism $f_i$ with a new mechanism $\tilde{f}_i$ that is no longer dependent on the original parents. With perfect interventions, all causal dependencies between the intervened target $z_i$ and its causes are removed, but the variable can still depend on its corresponding noise term, and its value may still be random. Thus, perfect interventions can be seen as a generalization of hard interventions where stochasticity is allowed.

- **Imperfect intervention.** An *imperfect* intervention modifies an SCM by replacing for a subset of causal variables the causal mechanism $f_i$ with a different causal mechanism $\tilde{f}_i$ that preserves the dependence on the causal variable's original parents. We note that imperfect interventions are a special case of soft interventions where the variable is still dependent on all of its original parents.

**Counterfactuals.** Beyond the ability to compute causal effects from interventional distributions, SCMs enable reasoning about counterfactuals. A counterfactual is a query at the unit level that asks the question: "Given an arbitrary unit, what would $z_j$ have been if $z_i$ were different?". Counterfactuals, unlike interventions, are relative to a factual observation. That is, given an observed set of variables that reflect reality, counterfactuals aim to infer what would happen if the original observations were intervened upon while keeping the original noise terms. Let $\mathcal{C} = \langle \mathcal{Z}, \mathcal{E}, F, p_{\mathcal{E}} \rangle$ be an SCM over variables $z$. Given some observations $z$, Pearl (2009) outlines the following three-step procedure to perform counterfactual inference:

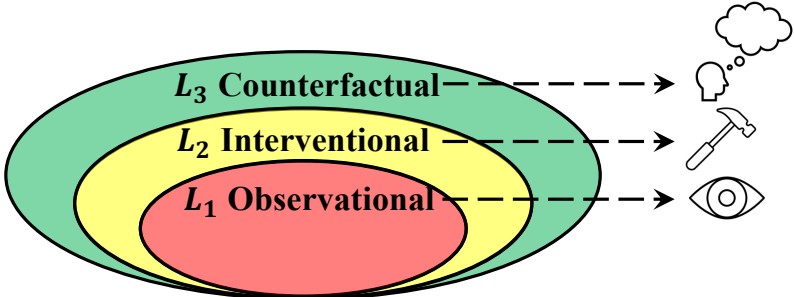

Figure 3: Pearl's Causal Hierarchy

- **Abduction:** Infer noise terms $\epsilon$ compatible with observations $z$ (i.e., $p(\epsilon|z)$)

- **Action:** Perform an intervention $\mathbf{do}(z_i = c)$ corresponding to the desired manipulation, which results in a modified interventional SCM $\tilde{\mathcal{C}} = \mathcal{C}_{\mathbf{do}(z_i=c)}$

- **Prediction:** Compute the counterfactual quantity based on the modified SCM $\tilde{\mathcal{C}}$ and $p(\epsilon|z)$

**Definition 2 (Pearl's Causal Hierarchy (Pearl, 2019))** *The observational ($L_1$), interventional ($L_2$), and counterfactual ($L_3$) distributions entailed by the SCM form a hierarchy in the sense that $L_1 \subset L_2 \subset L_3$, where each level encodes richer information that the previous level cannot express. The three layers are collectively referred to as Pearl's Causal Hierarchy (PCH).*

**On Pearl's Causal Hierarchy.** $L_1$ (observational) is concerned with modeling mere statistical correlation and does not in itself provide a causal perspective. $L_2$ (interventions) focuses on manipulating a system of causal variables to observe downstream effects on other variables. Interventions are often performed using the *do-operator* and are population-level operations. Finally, $L_3$ (counterfactuals) is the most complex layer and deals with retrospective thinking. That is, counterfactuals are unit-level queries that imagine alternative situations contrary to factual observations under a specific intervention. For instance, a question such as "Would I have gotten an A grade on my exam had I studied for one more hour?" would be a counterfactual query. An illustration of Pearl's causal hierarchy is shown in Figure 3.

## 2.2 Variational Autoencoders

The variational autoencoder (VAE) (Kingma & Welling, 2013), shown in Figure 4a, is a framework for optimizing latent variable models inspired by the theory of variational inference and energy-based models. Similar to a standard autoencoder, the VAE is composed of both an encoder and a decoder and is trained to minimize the reconstruction error. However, instead of encoding an input as a single point, the VAE encodes it as a distribution over the latent space. Formally, suppose the observed data $x$ is mapped to some low-dimensional latent space via an encoder $h : \mathcal{X} \to \mathcal{Z}$ and reconstructed by a decoder $g : \mathcal{Z} \to \mathcal{X}$. The goal of the VAE is to approximate the likelihood of the observed data, namely $p_\theta(x)$, parameterized by $\theta$. Given some prior over the latent variable $p_\theta(z)$, we can introduce a variational distribution $q_\phi(z|x)$ with parameters $\phi$ that approximates an intractable posterior $p_\theta(z|x)$. Since we cannot tractably estimate the exact likelihood, we alternatively maximize the evidence lower bound (ELBO), which is a lower bound to the log-likelihood of the data formulated as follows:

$$\log p_\theta(x) \geq \mathbb{E}_{q_\phi(z|x)}[\log p_\theta(x|z)] - \mathcal{D}_{KL}(q_\phi(z|x)||p_\theta(z)) \tag{6}$$

where the first term corresponds to the reconstruction loss (likelihood) and the second term encourages the learned posterior to match a suitable prior of the latent variables.

## 2.3 Normalizing Flows

Normalizing flows (Tabak & Turner, 2013; Tabak & Vanden-Eijnden, 2010; Papamakarios et al., 2021), shown in Figure 4b, are an expressive and general way of constructing flexible probability distributions over

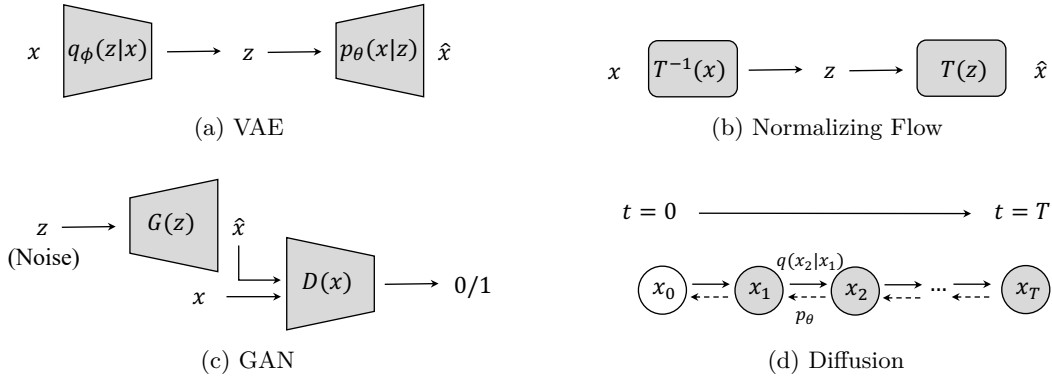

Figure 4: Overview of generative models (a) Variational Autoencoder (VAE), (b) Normalizing Flow, (c) Generative Adversarial Network (GAN), and (d) Diffusion models

continuous random variables. Flow-based models can transform a simple base distribution, e.g., Gaussian, into a complex distribution that resembles the distribution of the underlying data. Unlike VAEs and GANs, flow-based models explicitly learn the data distribution and thus the loss function is simply the negative log-likelihood. Consider the observed data $x$ and suppose, similar to the VAE, that we would like to estimate the likelihood of this data. In flow-based models, we express x as an invertible transformation of some real-valued variable $z$ as follows:

$$x = T(z) \tag{7}$$

where $z$ is sampled from some distribution $p(z)$, known as the *base distribution* of the flow-based model. Further, we require that $T$ is a diffeomorphic function and that $z$ is the same dimension as $x$. The density of $x$ is well-defined and can be estimated exactly by a change of variables:

$$p(x) = p(z)|\det J_T(z)|^{-1} \tag{8}$$

or, equivalently,

$$p(x) = p(T^{-1}(x))|\det J_T(T^{-1}(x))| \tag{9}$$

The Jacobian, which in practice can be efficiently computed by restricting it to a triangular structure, is a $d \times d$ matrix consisting of all partial derivatives of the transformation $T$

$$J_T(z) = \begin{bmatrix} \frac{\partial T_1}{\partial z_1} & \cdots & \frac{\partial T_1}{\partial z_d} \\ \vdots & \ddots & \vdots \\ \frac{\partial T_d}{\partial z_1} & \cdots & \frac{\partial T_d}{\partial z_d} \end{bmatrix} \tag{10}$$

Flows can be thought of as warping space to mold the base density $p(z)$, which is usually a simple distribution such as a Gaussian, into a complex distribution $p(x)$. The determinant of the Jacobian is the volumetric change in $z$ due to the transformation $T$. Flow-based models are capable of both generating new samples from the data distribution by *sampling* from the base distribution and computing the forward transformation $T$ and exact *density estimation* by computing the inverse transformation $T^{-1}$. Normalizing flows are easier to converge but less expressive when compared to GANs and VAEs because the transformations need to be invertible with a tractable Jacobian determinant.

## 2.4 Generative Adversarial Networks

Generative adversarial networks (GANs) (Goodfellow et al., 2014), shown in Figure 4c, are an example of implicit generative models, which are capable of sampling from probability distributions but cannot learn the likelihoods for data points. Thus, GANs are quite different than VAEs in their applications. GANs utilize an adversarial training procedure to produce realistic samples from distributions over a very high dimensional space, such as images. To learn the generator's distribution $p_g$ over data $x$, a prior is defined on input noise

variables $p(z)$ that is mapped to the data space via $G(z)$, where $G$ is a differentiable function (implemented as a multilayer perceptron) known as the generator network. The goal of $G$ is to generate samples close to the input distribution $p(x)$. Another function $D(x)$ (implemented as a multilayer perceptron), known as the discriminator network, is defined to represent the probability that $x$ indeed came from the data distribution $p_{\text{data}}$ rather than the generator $p_g$. The discriminator $D$ is trained to maximize the probability of assigning the correct label to both training examples from input distribution and samples generated from $G$. Simultaneously, $G$ is trained to minimize $\log(1 - D(G(z)))$. Formally, $G$ and $D$ play the following minimax zero-sum game:

$$\min_G \max_D \ \mathbb{E}_{x \sim p_{\text{data}}(x)}[\log D(x)] + \mathbb{E}_{z \sim p(z)}[\log(1 - D(G(z)))] \tag{11}$$

To optimize with respect to both networks, gradient descent is performed to update the generator and gradient ascent is performed to update the discriminator. GAN models are known for potentially unstable training and relatively less diversity in generation due to the adversarial training process.

## 2.5 Diffusion Models

Inspired by non-equilibrium statistical physics (Jarzynski, 1997)), diffusion models (Sohl-Dickstein et al., 2015; Ho et al., 2020), shown in Figure 4d, were proposed as a likelihood-based approach to achieve both flexible model structure and tractable estimation of probability distributions. Unlike VAEs, diffusion models are learned with a fixed procedure, and the latent variable has the same dimension as the original data. Diffusion models have shown impressive results in image generation tasks, even outperforming GANs in many cases (Dhariwal & Nichol, 2021). The idea of the diffusion model is to define a Markov chain of diffusion steps to slowly destroy the structure in a data distribution through a forward diffusion process by adding noise (Ho et al., 2020) and learn a reverse diffusion process that restores the structure of the data. Some proposed methods, such as DDIM (Song et al., 2021a), break the Markov assumption[1] to speed up the sampling in the diffusion process.

**Forward Diffusion.** Given some input data sampled from a distribution $x_0 \sim q(x)$, the forward diffusion process is defined by adding small amounts of Gaussian noise to the sample in $T$ steps, thereby producing noisy samples $x_1, \ldots, x_T$. The distribution of noisy sample $x_t$ is defined as a conditional distribution as follows:

$$q(x_t|x_{t-1}) = \mathcal{N}(x_t; \sqrt{1 - \beta_t}x_{t-1}, \beta_t I) \tag{12}$$

where $\beta_t \in (0, 1)$ is a variance parameter that controls the step size of noise. As $t \to \infty$, the input sample $x_0$ loses its distinguishable features. In the end, when $t = T$, $x_T$ is equivalent to an isotropic Gaussian. From Eq. (12), we can then define a closed-form tractable posterior over all time steps factorized as follows:

$$q(x_{1:T}|x_0) = \prod_{t=1}^{T} q(x_t|x_{t-1}) \tag{13}$$

Now, $x_t$ can be sampled at any arbitrary time step $t$ using the well-known reparameterization trick (Kingma & Welling, 2013; Rezende et al., 2014). Let $\alpha_t = 1 - \beta_t$ and $\bar{\alpha}_t = \prod_{i=1}^{t} \alpha_i$:

$$q(x_t|x_0) = \mathcal{N}(x_t; \sqrt{\bar{\alpha}_t}x_0, (1 - \bar{\alpha}_t)I) \tag{14}$$

**Reverse Diffusion.** In the reverse process to sample from $q(x_{t-1}|x_t)$, the goal is to recreate the true sample $x_0$ from a Gaussian noise input $x_T \sim \mathcal{N}(0, I)$. Unlike the forward diffusion, $q(x_{t-1}|x_t)$ is not tractable and thus requires learning a model $p_\theta$ to approximate the conditional distributions as follows:

$$p_\theta(x_{0:T}) = p(x_T) \prod_{t=1}^{T} p_\theta(x_{t-1}|x_t)$$

$$p_\theta(x_{t-1}|x_t) = \mathcal{N}(x_{t-1}; \mu_\theta(x_t, t), \Sigma_\theta(x_t, t)) \tag{15}$$

---

[1]In the context of diffusion models, the Markov assumption refers to the distribution over $x_t$ depending only on $x_{t-1}$

where $\mu_\theta$ is learned via neural networks, typically a UNet (Ronneberger et al., 2015). Further, it turns out that conditioning on the input $x_0$ yields a tractable reverse conditional probability

$$q(x_{t-1}|x_t, x_0) = \mathcal{N}(x_{t-1}; \tilde{\mu}(x_t, x_0), \tilde{\beta}_t I) \tag{16}$$

where $\tilde{\mu}$ and $\tilde{\beta}$ are the analytical mean and variance parameters derived as a result of conditioning on $x_0$. We can learn the reverse denoising process by optimizing the following variational lower bound:

$$\mathbb{E}[\log p_\theta(x_0)] \geq -\mathbb{E}\left[\log \frac{q(x_{1:T}|x_0)}{p_\theta(x_{0:T})}\right] = \mathcal{L}_{\text{VLB}} \tag{17}$$

For a detailed derivation, see Sohl-Dickstein et al. (2015). Diffusion models often require longer training and sampling times, making them computationally expensive.

## 2.6 Disentangled Representation Learning

For the longest time, the efficacy of machine learning models relied on human ingenuity and prior knowledge to perform feature engineering on data. The realization that such engineering is inefficient and demands a more automated approach gave rise to the field of *representation learning* (Bengio et al., 2013). Representation learning aims to extract meaningful information (or features) from data that can be used downstream to build robust classifiers or predictors. Thus, it is desirable to build representation-learning algorithms that incorporate certain priors about the world to automatically learn useful representations. In the probabilistic sense, which is what we focus on in this survey, a good representation is one that should ideally learn the posterior distribution of the underlying factors of variation for the observed input distribution. Further, a representation that is disentangled such that each factor is independently manipulable enables a robust representation useful for downstream tasks such as learning fair and robust predictors (Locatello et al., 2019a).

The problem of representation learning can be linked to nonlinear ICA (Hyvärinen & Pajunen, 1999). Suppose that our low-level observations $x \in \mathbb{R}^d$ are explained by a small number of latent variables $z = (z_1, \ldots, z_n)$, where $n \ll d$ and $x$ is generated by applying an injective (or sometimes diffeomorphic) map $g : \mathbb{R}^n \to \mathbb{R}^d$, also called the *mixing* function, to $z$ such that

$$x = g(z) \tag{18}$$

In nonlinear ICA or disentangled representation learning, $z_i$ are assumed to be mutually independent such that

$$p(z_1, \ldots, z_n) = \prod_{i=1}^{n} p(z_i) \tag{19}$$

Although there is no one commonly accepted definition of disentanglement, Locatello et al. (2020) propose to formalize the notion in the following general definition:

**Definition 3 ((Locatello et al., 2020))** *The goal of disentangled representation learning is to learn a mapping $g^{-1}(x)$ such that the effect of the different factors of variation is axis-aligned with different coordinates. That is, each factor of variation is associated with exactly one group of coordinates of $g^{-1}(x)$ and vice-versa, where the groups are non-overlapping. Thus, varying one factor of variation while keeping the others fixed results in a variation of exactly one group of coordinates.*

Several proposed VAE variants, such as $\beta$-VAE (Higgins et al., 2017) and FactorVAE (Kim & Mnih, 2018), have proposed objectives to disentangle the learned latent codes via independence constraints. However, such models fail to disentangle factors and do not address one major issue in representation learning: unidentifiability of representations.

### 2.6.1 The Identifiability Problem

A significant barrier to representation learning is the identifiability problem. Probabilistic generative models suffer from indeterminacy, which refers to the situation where the underlying factors of variation cannot be

uniquely inferred from empirical data. Characterizing and reducing these indeterminacies is the endeavor of *identifiability*. Identifiability is an important property to achieve in representation learning since it guarantees the stability and robustness of a model. When different representations can explain the same observational data equally well and cannot identify the true factors of variation, we say that the model is *unidentifiable*. In general, it is infeasible to always uniquely identify the underlying factors and remove all indeterminacies since it is often task-dependent. Thus, identifiable representation learning typically aims to recover the true factors (and mixing function) up to tolerable ambiguities (i.e., transformations, reordering, etc.). It is important to note that model identifiability is an asymptotic property that can be achieved in its strictest form only in the limit of an infinite amount of data (Xi & Bloem-Reddy, 2023). However, weaker identifiability results often suffice to recover ground-truth factors. The theory of identifiability stems from the literature on nonlinear independent component analysis (Hyvärinen & Pajunen, 1999). It has been shown that the identifiability of nonlinear ICA is, in fact, not possible unless certain restrictions are made on the mixing function or other auxiliary information is provided (Hyvarinen et al., 2018). Locatello et al. (2019b) extend this result to representation learning and show that learning disentangled representations in an unsupervised manner without any inductive biases is impossible. It turns out that the notion of disentanglement and identifiability are actually quite related. Formally, following Khemakhem et al. (2020) and Gresele et al. (2021), identifiability is defined as follows:

**Definition 4 (Identifiability)** *Let $\mathcal{G}$ be the set of all smooth, mixing functions $g : \mathbb{R}^n \to \mathbb{R}^d$, and $\mathcal{P}$ be the set of all smooth, factorized densities $p(z)$. Let $\Theta \subseteq \mathcal{G} \times \mathcal{P}$ be the domain of parameters, where the elements of $\Theta$ parameterize the mixing functions and densities, and let $\sim$ be an equivalence relation on $\Theta$. Then, the generative process in (18) is said to be $\sim$-identifiable on $\Theta$ if*

$$\forall \theta, \tilde{\theta} \in \Theta : \qquad p_\theta(x) = p_{\tilde{\theta}}(x) \implies \theta \sim \tilde{\theta} \tag{20}$$

If the true model belongs to $\Theta$, then identifiability implies that any model in $\Theta$ learned from infinite amounts of data will be $\sim$-equivalent to the true model. In the context of the VAE[2], if two different choices of model parameters $\theta$ and $\tilde{\theta}$ lead to the same marginal density $p_\theta(x)$, then they are equal and the joint distributions should match (i.e., $p_\theta(x, z) = p_{\tilde{\theta}}(x, z)$). If the joint distribution matches, then we have found the correct priors $p_\theta(z) = p_{\tilde{\theta}}(z)$ and correct posteriors $p_\theta(z|x) = p_{\tilde{\theta}}(z|x)$. Practically speaking, identifiability essentially guarantees that regardless of how many times we train the representation learner, we will always obtain the same unique solution (up to tolerable ambiguities). Thus, if identifiability didn't hold, learning representations would not be as useful for downstream tasks.

Linear ICA is well-known to be identifiable given non-Gaussian noise (Comon, 1994). That is, there exists a unique unmixing function that maps the given observations to a unique set of sources. However, in the general case, nonlinear ICA is ill-defined and not equivalent to blind-source separation (BSS). That is, in the $n = d$ case, different mixtures of $z_i$ and $z_j$ can be independent. Some classic counterexamples include the *Darmois construction* (Darmois, 1953; Hyvärinen & Pajunen, 1999) and *rotated Gaussian measure-preserving automorphisms* (Gresele et al., 2021). The following two examples show why nonlinear ICA is unidentifiable in the general case and the need for additional signals and strategies to resolve identifiability.

**Example 1 (Darmois Construction - (Hyvärinen & Pajunen, 1999))** The *Darmois* construction $g^D : \mathbb{R}^n \to (0, 1)^n$ is defined as the recursive application of the following conditional CDF:

$$g_i^D(x_{1:i}) = p(X_i \le x_i | x_{1:i-1}) = \int_{-\infty}^{x_i} p(x_i' | x_{1:i-1}) \, dx_i' \qquad (i = 1, \dots, n) \tag{21}$$

The *Darmois solution* maps any observed density to a uniform distribution using the CDF above, as shown in Figure 5a. The estimated sources are thus independent of the conditioning variables $x_{1:i-1}$. Thus, $y^D = g^D(x)$ are mutually independent uniform random variables that may not be meaningfully related to the true sources of variation. Let the mixing function be defined as $f^D = (g^D)^{-1}$ and the uniform density

---

[2]Recall that the VAE model learns a full generative model $p_\theta(x, z) = p_\theta(x|z)p_\theta(z)$ and an inference model $q_\phi(z|x)$ which approximates its posterior $p_\theta(z|x)$. However, we generally have no guarantees about what these learned distributions actually are and only the marginal distribution over $x$ is meaningful.

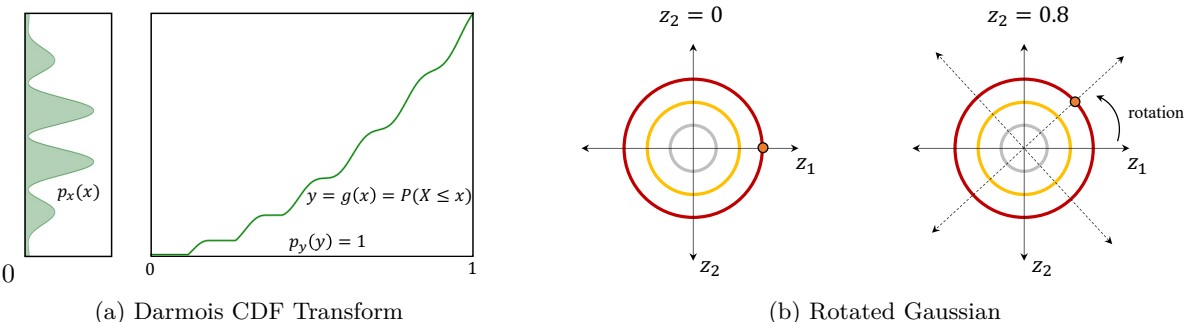

(a) Darmois CDF Transform          (b) Rotated Gaussian

Figure 5: An illustration of (a) the Darmois construction and (b) rotated Gaussian MPA that are counterexamples to blind-source separation of nonlinear ICA. The Darmois CDF transform maps the data to independent intervals of a uniform distribution. The Gaussian is invariant to rotations and thus represents the same observational distribution after rotation (the left circle and right circle are equivalent distributions).

on the interval $(0,1)^n$ by $p_u$. Then, the Darmois solution $(f^D, p_u)$ is a counterexample to blind source separation since we can always find another set of independent sources that explain the data equally as well. For instance, consider a two-variable case where we have independent random variables $x_1$ and $x_2$. Then, we have that $y_1 = g_1(x_1, x_2)$ and $y_2 = g_2(x_1, x_2)$ are independent from the original observed variables $x_1$ and $x_2$. Thus, we can construct infinitely many such independent random variables and can never recover the unique sources.

**Example 2 (Rotated Gaussian MPA - (Gresele et al., 2021))** Another class of counterexamples involves measure-preserving automorphisms (MPA), which are identity functions on the sources that preserve the distribution of the source variables. Let $R$ be an orthogonal matrix, and denote by $F_z(z) = (F_{z_1}(z_1), \ldots, F_{z_n}(z_n))$ and $\Phi(z) = (\phi(z_1), \ldots, \phi(z_n))$ the elementwise CDF of a smooth, factorized density $p_z$ and of a Gaussian, respectively. Then, the rotated Gaussian MPA $a^R(p_z)$ is

$$a^R(p_z) = F_z^{-1} \circ \Phi \circ R \circ \Phi^{-1} \circ F_z \tag{22}$$

$a^R(p_z)$ maps to a standard Gaussian (rotationally invariant), then applies a rotation, and finally maps back to the sources. In this case, there exists a different set of independent sources through $a^R(p_z)$ that explain the data equally as well. Interestingly, principal component analysis (PCA) is not identifiable precisely because it is a rotation of a Gaussian through the orthogonal rotation matrices $U$ and $V$ obtained through singular value decomposition of the data covariance matrix. Figure 5b shows the rotational indeterminacy of Gaussian i.i.d. models, where each point in the circle represents an equivalent solution. In other words, the joint distribution will always be equivalent no matter what rotations are performed on the Gaussian, so uniquely identifying the sources is hopeless.

Therefore, additional constraints and assumptions are necessary if we hope to uniquely identify the factors of variation. The goal is to get closer to recovering the true underlying factors up to some trivial transformation. Under certain conditions, we can often recover the underlying factors up to some affine transformation of the learned factors as formalized in Definition 5.

**Definition 5 (Affine equivalence)** *We say $\theta$ and $\tilde{\theta}$ are **affine equivalent** if*

$$(g, p_z) \sim_L (\tilde{g}, p_{\tilde{z}}) \iff \exists L \ s.t. \ (g, p_z) = (\tilde{g} \circ L^{-1}, p_{\tilde{z}}) \tag{23}$$

*where $L$ is a linear invertible transformation.*

However, to *disentangle* the sources, we must go beyond affine equivalent identifiability since it allows for any linear combination of learned factors to recover underlying factors, which leads to entangled solutions. For most settings, the best we can do is to recover the factors up to a simple reordering (permutation) and scaling (elementwise reparameterization) of the learned factors as formulated in Definition 6.

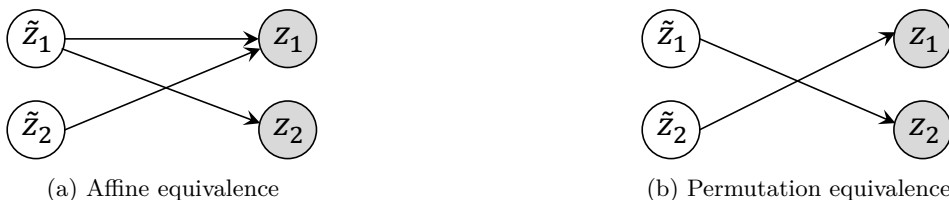

(a) Affine equivalence          (b) Permutation equivalence

Figure 6: A simple 2-variable example of (a) affine equivalence vs. (b) permutation equivalence, where $\tilde{z}_i$ denotes the learned latent factor and $z_i$ denotes the ground truth underlying factor. Note that the arrows in the diagram indicate mappings, not causal relations.

**Definition 6 (Permutation equivalence)** *We say $\theta$ and $\tilde{\theta}$ are **permutation equivalent** if*

$$(g, p_z) \sim_P (\tilde{g}, p_{\tilde{z}}) \iff \exists P, h \ s.t. \ (g, p_z) = (\tilde{g} \circ h^{-1} \circ P^{-1}, p_{\tilde{z}}) \tag{24}$$

*where $P$ is a permutation matrix and $h(z) = (h_1(z_1), \ldots, h_n(z_n))$ are invertible elementwise reparameterizations. A learned model $\tilde{\theta}$ is said to be **disentangled** if $\theta \sim_P \tilde{\theta}$.*

According to the above definition, it turns out that identifying latent factors up to permutation (and possibly elementwise reparameterization) of the ground-truth is equivalent to disentangling the factors of variation. **Componentwise equivalence** is a special case of permutation equivalence where $P = I$ (identity permutation). **Block identifiability** is a form of identifiability where a group of latents is uniquely identifiable, but latents within the group may not be. Since disentanglement is impossible to achieve without some form of inductive bias, we next briefly discuss works that propose additional weak supervision and other strategies to learn provably identifiable representations. An illustrative example of affine and permutation equivalence is shown in Figure 6.

### 2.6.2 Towards Identifiability in Disentangled Representation Learning

The natural question becomes: Can we utilize additional structure in data to attain identifiability? The following approaches have been the most promising strategies for dealing with identifiability challenges in representation learning. Many of the identifiability results in disentangled and causal representation learning utilize the general structure of the following proposals.

**Latent Conditioning.** Based on Hyvarinen et al. (2018), Khemakhem et al. (2020) propose a unified theory connecting VAEs to nonlinear ICA called identifiable VAE (iVAE). The authors propose conditioning the latent variable on some auxiliary information, such as time steps or class labels, with an exponential family prior such that the underlying factors are rendered conditionally independent:

$$p_{T,\lambda}(z|y) = \prod_i p_\theta(z_i|y) = \prod_i h_i(z_i) \exp\left[\sum_{j=1}^{k} T_{i,j}(z_i)\lambda_{i,j}(y) - \psi_i(y)\right] \tag{25}$$

where $h_i(z_i)$ is the base measure, $T_i : \mathcal{Z} \to \mathbb{R}^k$ and $T_i = (T_{i,1}, \ldots, T_{i,k})$ are the sufficient statistics, $\lambda_i(y) = (\lambda_{i,1}(y), \ldots, \lambda_{i,k}(y))$ are the corresponding natural parameters, $k$ is the dimension of each sufficient statistic, and the remaining term $\psi_i(y)$ acts as a normalizing constant. Consider the well-known Gaussian distribution, which is in the exponential family. We have that the sufficient statistic is $T_i(z) = (z, z^2)$ and the base measure is $h_i(z) = \frac{1}{\sqrt{2\pi}}$. The function $\lambda$ outputs the natural parameter vector for the conditional distribution, which is $(\lambda_1, \lambda_2) = (\frac{\mu}{\sigma^2}, -\frac{1}{\sigma^2})$ in the Gaussian case. Auxiliary label information (e.g., class label, domain label, time index, etc.) enables certain independence and distributional assumptions to be made on the underlying latent variables, which can be utilized to identify the latent factors.

**Multi-view Data and Contrastive Learning.** Multi-view nonlinear ICA (Gresele et al., 2019) posits that multiple views of observational data may have a shared latent representation. This paradigm has been utilized in several different ways for disentanglement and identifiability guarantees. Inspired, by this idea, Locatello et al. (2020) propose considering pairs of contrastive data $(x_1, x_2)$ consisting of before and after

changes (interventions) have occurred to a system as an inductive bias for representation learning. The generative model is defined as sampling two images from the causal generative process with an intervention on a random subset of factors of variation. The auxiliary information in this setting is the positive pair. Let $k$ be the number of factors in which the two observations differ, $S$ denote the subset of shared factors of size $n - k$, and $\bar{S} = [n] \setminus S$. Then, we have the following generative process:

$$p(z) = \prod_{i=1}^{n} p(z_i), \quad p(\tilde{z}) = \prod_{i=1}^{k} p(z_i), \quad S \sim p(S) \tag{26}$$

$$x_1 = g^*(z), \quad x_2 = g^*(f(z, \tilde{z}, S)) \tag{27}$$

$$f(z, \tilde{z}, S)_S = z_S \quad \text{and} \quad f(z, \tilde{z}, S)_{\bar{S}} = \tilde{z} \tag{28}$$

where $g^*$ is a function that maps the latent variable to observations and $f$ makes the relation between $x_1$ and $x_2$ explicit. Intuitively, when generating $x_2$, $f$ selects entries from $z$ with index in $S$ and replaces the remaining factors with $\tilde{z}$. Thus the two observations $x_1$ and $x_2$ have the same shared factors indexed by $S$. Contrastive learning has been recently shown to provably invert the data-generating process (Zimmermann et al., 2021). In the case of contrastive learning, the data from multiple views provides sufficient variability in the data to uniquely identify the factors of variation.

**Restricting Mixing Function.** Gresele et al. (2021) and Reizinger et al. (2022) propose a new theory called independent mechanism analysis (IMA) to learn identifiable representations based on restricting the function class of the mixing function and placing orthogonality constraints on the columns of the Jacobian of the mixing function. Inspired by the Independence of Causal Mechanisms (ICM) principle from causality (Scholkopf et al., 2021), the authors propose a new form of independence between the influences of sources on the observation rather than statistical independence. Specifically, the authors restrict the mixing function $f$ to satisfy mutual independence of $\frac{\partial f}{\partial z_i}$ for all $i = \{1, \ldots, n\}$. Formally, we have that the mechanisms by which each source $z_i$ influences the observed distribution are independent of each other in the sense that for all $z$:

$$\log |J_f(z)| = \sum_{i=1}^{n} \log \left\| \frac{\partial f}{\partial z_i}(z) \right\| \tag{29}$$

This implies that the absolute value of the determinant of the Jacobian should decompose as the product of the norms of $\frac{\partial f}{\partial z_i}$. Reizinger et al. (2022) show that the evidence lower bound in VAEs converges to a regularized version of the log-likelihood. The regularization term gives VAEs the ability to perform independent mechanism analysis and serves as an inductive bias for decoders with column-orthogonal Jacobians. The authors show that the regularized objective enables the identifiability of the ground-truth latent factors. Further, they show that VAEs satisfy the *self-consistency* property in near-deterministic regimes (i.e., the VAE encoder inverts the decoder even if the VAE is not exactly deterministic). Assumptions on the mixing function (linearity, polynomial, etc.) can also help to narrow down the space of solutions by assuming a simpler data-generating process.

**Sparsity.** Recently, sparsity principles have been proposed to promote disentanglement (Zheng et al., 2022; Lachapelle et al., 2022; Zheng & Zhang, 2023; Xu et al., 2023). Zheng et al. (2022) propose Structural Sparsity, a new theory of nonlinear ICA with unconditional priors by leveraging sparsity in the relations between sources and observations (i.e., mixing). The general intuition behind structural sparsity is that for each latent factor of variation $z_i$, there exists a set of observed variables in $x$ such that $z_i$ is the only latent source that is used to generate the set of observed variables. In addition to sparsity in the mixing, there are other applications of sparsity such as modeling sparse mechanisms between latent variables (Lachapelle et al., 2022; Lachapelle & Lacoste-Julien, 2022) and the use of a sparse number of latents for downstream prediction tasks to promote disentanglement (Lachapelle et al., 2023).

For a more rigorous treatment of identifiability and indeterminacies in generative models, we refer the reader to Xi & Bloem-Reddy (2023). Typically, in identifiability proofs, the goal is to first show linear-equivalent identifiability under a certain set of assumptions and then leverage independence or sparsity structure to show permutation-equivalent identifiability (disentanglement).

| Approach | Mixing Function | Causal Model | Key Assumptions | Identifiability Result |
|----------|-----------------|--------------|-----------------|------------------------|
| CausalVAE* (Yang et al., 2021) | nonparametric, injective | linear additive noise model | observed auxiliary labels | componentwise correspondence |
| DEAR* (Shen et al., 2022) | nonparametric, injective | nonlinear | observed auxiliary labels, causal prior | componentwise correspondence |
| SCM-VAE* (Komanduri et al., 2022) | nonparametric, injective | nonlinear | observed auxiliary labels, causal prior | componentwise correspondence |
| ICM-VAE* (Komanduri et al., 2023) | nonparametric, diffeomorphic | nonparametric | observed auxiliary labels, causal disentanglement prior | causal mechanism equivalence, permutation equivalence |
| Unpaired Multi-domain CRL (Sturma et al., 2023) | linear, injective | linear | data from $d'$ unpaired domains | permutation equivalence via linear ICA in each domain |
| G-CaRL* (Morioka & Hyvärinen, 2023) | nonparametric, diffeomorphic | nonparametric | $M$ disjoint groups of data | permutation equivalence |
| Multi-view CRL (Yao et al., 2023) | nonparametric, diffeomorphic, view-specific | nonparametric | multi-view data, subset of latents for each modality | block identifiability of shared latents |
| LEAP* (Yao et al., 2022b) | nonparametric, injective | nonparametric & parametric cases | temporal observational data, nonstationarity auxiliary labels | permutation equivalence of temporal latents |
| TDRL* (Yao et al., 2022a) | nonparametric, injective | nonparametric | temporal observational data, stationarity & observed nonstationarity | permutation equivalent of temporal latents |
| NCTRL* (Song et al., 2023) | nonparametric, injective | nonparametric | temporal observational data, unobserved nonstationarity | permutation equivalence of temporal latents |
| Interventional CRL (Ahuja et al., 2023) | polynomial, injective | nonparametric | 1 perfect intervention per node | permutation equivalence |
| Linear CRL (linear mixing) (Squires et al., 2023) | linear, injective | linear | 1 perfect intervention per node (unpaired), unknown interv. targets | permutation equivalence |
| Linear CRL (general mixing) (Buchholz et al., 2023) | nonparametric, diffeomorphic | linear | 1 perfect intervention per node (unpaired), unknown interv. targets | permutation equivalence |
| Discrepancy-VAE* (Zhang et al., 2023b) | polynomial, injective | nonparametric | 1 soft intervention per node (unpaired), unknown interv. targets | permutation equivalence of ancestral relations |
| Nonparametric CRL (von Kügelgen et al., 2023) | nonparametric, diffeomorphic | nonparametric | 2 perfect interventions per node (paired), unknown interv. targets | permutation equivalence |
| Parametric CRL (Score-based) (Varıcı et al., 2023a) | linear, injective | nonparametric | 1 hard intervention per node (unpaired)/ 1 soft intervention per node | permutation equivalence/ perfect DAG recovery |
| Nonparametric CRL (Score-based) (Varıcı et al., 2023b) | nonparametric, diffeomorphic | nonparametric | 2 hard interventions per node (unpaired), unknown interv. targets | permutation equivalence |
| CITRIS* (Lippe et al., 2022b) | nonparametric, bijective | nonparametric | temporal sequences, known interv. targets | recover minimal causal variables |
| iCITRIS* (Lippe et al., 2023a) | nonparametric, bijective | nonparametric | temporal sequences, known interv. targets, instantaneous effects | recover minimal causal variables and parents |
| BISCUIT* (Lippe et al., 2023b) | nonparametric, diffeomorphic | nonparametric | temporal sequences with action variable, unknown interv. targets | permutation equivalence given binary interaction pattern |
| Mechanism Sparsity Regularization (Lachapelle et al., 2022) | nonparametric, diffeomorphic | nonparametric | temporal sequences with action variable | permutation equivalence given sparse temporal action causal graph |
| SSL Data Augmentation CRL (von Kügelgen et al., 2021) | nonparametric, bijective | nonparametric | augmented counterfactual data | block identifiability of content latents |
| LCM* (Brehmer et al., 2022) | nonparametric, diffeomorphic | nonparametric | paired counterfactual data, perfect interventions, unknown interv. targets | permutation equivalence |

\* one of the main contributions is a learning framework.

Table 2: Summary of Causal Representation Learning Identifiability Results

# 3  Causal Representation Learning

Until recently, most representation learning methods assumed the underlying factors of variation to be independent (Khemakhem et al., 2020) and that the data is independently and identically distributed, which is often impractical in real-world scenarios. The world is full of dynamically changing systems consisting of complex interactions that induce correlations or causal dependencies between factors. Träuble et al. (2021) show that existing methods are not sufficient to disentangle the factors when the data is non-iid and factors are correlated. Thus, a more realistic endeavor would be to design models that assume the factors underlying a generative process are *causally related.* Combining causal modeling and representation learning allows one to develop machine learning models for high-dimensional inputs whose underlying factors are governed by an SCM. The goal of causal representation learning is to learn high-level causally related factors that describe meaningful semantics of the data and their causal structure (Scholkopf et al., 2021). Semantically meaningful can refer to several properties of a representation, such as robustness, fairness, transferability, interpretability, or explainability. Causal representation learning (CRL) is an amalgamation of multiple lines of work, including identifiable representation learning, causal structure learning, and latent causal DAG learning. In this survey, we do not cover causal discovery methods and refer the reader to other surveys that comprehensively cover this landscape (Vowels et al., 2021; Squires & Uhler, 2022). However, we note that since causal structure learning is a component of CRL, many methods incorporate existing heuristic structure learning methods from observational data (Zheng et al., 2018) or a combination of observational and interventional data (Lippe et al., 2022a; Ke et al., 2023). The key feature of causal variables is that they are variables on which interventions are defined and whose relations are of interest. It is worth noting that independent factors of variation are a special case of causal models where the causal graph is trivial. In the following sections, we build up causal representation learning from nonlinear ICA and discuss methods according to their assumptions on the data-generating process: observational, interventional, or counterfactual. Table 2 summarizes the identifiability results, key assumptions, and main contributions of the causal representation learning methods discussed. The key assumptions are generally listed as the type of data assumed (observational, interventional, or counterfactual), the parametric form of the latent SCM, and the type of mixing function, in addition to any method-specific assumptions.

## 3.1  Formulating Causal Representation Learning

### 3.1.1  Linking Causal Representations to Nonlinear Independent Component Analysis

Recall that in traditional ICA, the ground-truth factors are assumed to be statistically independent. For the setting of causal representation learning, we diverge from the independent factors of variation assumption and suppose that $z_i$ are causal variables that may be dependent. The joint distribution for $z$ can thus be expressed as the following Markov factorization

$$p(z_1, \ldots, z_n) = \prod_{i=1}^{n} p(z_i | z_{\mathbf{pa}_i}) \tag{30}$$

induced by an underlying acyclic SCM $\mathcal{C} = (\mathcal{Z}, \mathcal{E}, F, p_{\mathcal{E}})$ with jointly independent noise $\epsilon_i$ and a set of causal mechanisms

$$F = \{z_i = f_i(z_{\mathbf{pa}_i}, \epsilon_i)\} \tag{31}$$

The goal of causal representation learning is to (1) learn the causal representation $z = g^{-1}(x)$, (2) the corresponding causal DAG, and (3) the causal mechanisms $p(z_i | z_{\mathbf{pa}_i})$ (Parascandolo et al., 2018; Bengio et al., 2020). Assuming a *reduced form*[3] SCM where we recursively substitute the structural assignments in the topological order of the causal graph, we can write the latent variables $z$ directly as a function of the noise

$$z = f_{\mathrm{RF}}(\epsilon) \tag{32}$$

---

[3]A reduced form SCM is one where the mechanisms are abstracted away to obtain causal variables as a function of noise terms (i.e., $z = f(\epsilon)$).

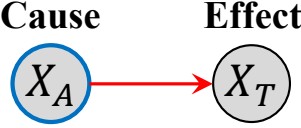

Figure 7: Relationship between altitude and temperature (Peters et al., 2017)

The mapping $f_{\mathrm{RF}} : \mathbb{R}^n \to \mathbb{R}^n$ has a lower triangular Jacobian. So, we can write the mapping from noise to data space as the following composition

$$x = g \circ f_{\mathrm{RF}}(\epsilon) \tag{33}$$

Learning (32) can be seen as a structured form of nonlinear ICA with an intermediate representation learned through $f_{\mathrm{RF}}$ (Schölkopf & von Kügelgen, 2022). There have also been recent efforts in generalizing ICA to a theory of causal component analysis (Liang et al., 2023) as a special case of causal representation learning where the causal graph is known.

### 3.1.2 Principles of Causal Representations

The following two principles describe the key properties of causal representations.

**Principle 1 (Independent Causal Mechanisms (ICM) - (Scholkopf et al., 2021))** *The causal generative process of a system's variables is composed of autonomous modules that do not inform or influence each other. In the probabilistic case, this means that the conditional distribution of each variable given its causes (i.e., its mechanism) does not inform or influence the other mechanisms.*

The independence of causal mechanisms is a statement about the algorithmic independence between mechanisms that generate causal variables. That is, the ICM principle implies the locality of interventions where changing a variable only affects the causal mechanism for that variable. In other words, the components, or mechanisms, are modular, autonomous, and reusable. Statistical dependencies are a result of introducing unexplained random variables into the system. However, a deterministic system consists of physical mechanisms. As a result, an algorithmic model of causal structure can be devised in terms of Kolmogorov complexity (Janzing & Schölkopf, 2010), which is the length of a bit string's shortest compression on a Turing machine and a measure of its information content. However, the Kolmogorov complexity is known not to be computable. Thus, we must resort to statistical and other information-theoretic notions of quantifying independence.

**A two-variable example of ICM Principle (Peters et al., 2017).** In the two-variable case, where one is referred to as the *cause* and the other as the *effect*, Principle 1 boils down to independence between the cause distribution and the mechanism producing the effect from the cause. The factors are independent in two senses: (i) *intervening* on one mechanism $p(z_i|z_{\mathbf{pa}_i})$ does not change the other mechanisms $p(z_j|z_{\mathbf{pa}_j})$ for $i \neq j$, and (ii) knowing information about some mechanism $p(z_i|z_{\mathbf{pa}_i})$ does not give us any information about a mechanism $p(z_j|z_{\mathbf{pa}_j})$ for $i \neq j$. Suppose we are interested in analyzing average annual temperature $X_T$ vs. altitude $X_A$ from different locations. Altitude and temperature are correlated variables. Intuitively, we can infer that this correlation must arise from the fact that changes in altitude cause changes in temperature. For example, at higher altitudes, the temperatures tend to be lower.

Figure 7 illustrates the relationship between the altitude and temperature variable at a given location. Suppose we collect two datasets from location $R$ and location $S$. Then, we can say that $p_R(X_A, X_T) \neq p_S(X_A, X_T)$ since the altitude marginal distributions likely differ, i.e., $p_R(X_A) \neq p_S(X_A)$. However, the conditionals (or mechanisms) that characterize the physical mechanism of generating temperature from altitude, $p(X_T|X_A)$, will likely be almost invariant to shifts in the marginal, i.e., $p_S(X_T|X_A) \approx p_R(X_T|X_A)$. Thus, if we assume that $X_A \to X_T$, in the joint factorization $p(X_A, X_T) = p(X_A)p(X_T|X_A)$, we can say that the mechanisms are independent and do not influence each other. However, if we assume an alternate factorization, where $X_T \to X_A$, namely $p(X_A, X_T) = p(X_T)p(X_A|X_T)$, we do not enjoy the same invariance property. That is, it would not make sense for the temperature to cause a physical change in the altitude and

we would not observe much change in altitude when intervening on temperature. Thus, the correct causal factorization will always yield independent causal mechanisms.

**Principle 2 (Sparse Mechanism Shift (SMS) - ([Scholkopf et al., 2021]))** *Small distribution changes tend to manifest themselves in a sparse way in the Markov/disentangled factorization in Eq. (30), i.e., they should usually not affect all factors simultaneously.*

Given that the ICM principle holds, the SMS hypothesis is a statement about distribution shifts. That is, distribution shifts (e.g., the training and testing set may come from different distributions) can often be attributable to sparse changes in causal conditionals of a causal disentangled factorization (or interventions on causal variables). To learn models that are generalizable to distribution shifts, it is desirable to learn causal representations due to their invariance to distribution shifts. There have been several works that use causal models to study distribution shift robustness ([Lu et al., 2022]) and how to mitigate spurious correlations ([Wang & Jordan, 2021]).

Guided by these principles, a causal representation should consist of independent mechanisms and must be robust to downstream tasks under distribution shifts.

### 3.1.3 Intuition behind Causal Disentanglement

The notion of *causal* disentanglement can seem counterintuitive or even contradictory. As mentioned before, disentanglement is not a completely well-defined concept. One approach could be to think about it in terms of simply enforcing statistical independence among all the learned factors in an unsupervised fashion, as done in $\beta$-VAE ([Higgins et al., 2017]). However, such an approach does not isolate the learned factors and semantically meaningful manipulations are difficult to perform; hence, the impossibility implication for arbitrary unsupervised disentanglement ([Locatello et al., 2019b]). Thus, Definition 3 aims to formulate a consistent definition through a non-overlapping condition on the information that each group of latent codes encodes. Causal representations adhere to the principle of independent causal mechanisms (Principle 1). Thus, causal disentanglement should be viewed through the lens of the ICM principle. In a setting where the factors of variation are causally related, the mechanisms that produce each causal variable as a function of its parents should be independent. Thus, we must ensure not only that each group of latent codes captures distinct information but also the recoverability of each independent causal mechanism ([Komanduri et al., 2023]) and the causal structure among factors (if learned). In a setting where the causal graph is learned jointly with the representation, the recoverability of the causal relations can be formulated as a graph isomorphism between the true causal model and the learned causal model ([Brehmer et al., 2022]).

### 3.1.4 Identifiability Problem in Causal Representation Learning

In causal representation learning, on top of identifying the latent representation, the causal graph encoding their relations must also be identifiable. The task of causal structure identification is already difficult in the purely observational setting since we can only recover a DAG up to Markov equivalence ([Spirtes et al., 2000]). However, it gets significantly more challenging when jointly learning latent causal variables. The increase in complexity from traditional disentangled representation learning to causal representation learning is primarily due to this joint identifiability of latent causal factors *and* the latent causal DAG. In traditional disentangled representation learning, one may be able to utilize strategies such as contrastive learning to disentangle factors ([Zimmermann et al., 2021]). However, in causal representation learning, we may need data generated according to some causal model (e.g., interventional or counterfactual data) to recover the causal mechanisms among causal factors. Additionally, discovering the causal structure and extracting causal variables becomes a chicken-and-egg problem, which makes CRL severely ill-posed. That is, how do we learn the causal structure without the causal variables, and how do we learn the causal variables without knowing the structural relationships among causal variables? This dilemma is what makes causal representation learning a challenging endeavor. In the following sections, we summarize work done towards achieving identifiable and disentangled causal representations from various weak-supervision signals and data-generating assumptions. We primarily focus our attention on VAE-based models used to learn disentangled causal representations, but some methods restrict the mixing function to be linear and thus

only use matrix decompositions. For motivation of the challenges we face in causal representation learning, consider the following simple example that illustrates unidentifiability.

**Example 3 (Unidentifiability in CRL Simple Linear Example)** *Suppose we have observations $x \in \mathbb{R}^2$ and latent variables $z \in \mathbb{R}^2$ related by $x = Gz$, where $G$ is a linear mixing function. For simplicity, let the causal variables be related by the following linear additive noise structural causal model*

$$z = A^T z + \epsilon = (I - A^T)^{-1}\epsilon, \qquad \epsilon \sim \mathcal{N}(0, I) \tag{34}$$

*where $A$ is the causal adjacency matrix. Suppose we have the following solution to the system*

$$A = \begin{bmatrix} 0 & 0 \\ -1 & 0 \end{bmatrix}, \qquad G = \begin{bmatrix} -2 & 3 \\ 1 & -4 \end{bmatrix} \implies \Sigma = \begin{bmatrix} 29 & -27 \\ -27 & 26 \end{bmatrix} \tag{35}$$

*where $\Sigma$ is the covariance of the observations $x$ derived as $\Sigma = \mathbb{E}[xx^T] = G(I - A^T)^{-1}(I - A^T)^{-T}G^T$. Now, consider the solution*

$$\hat{A} = \begin{bmatrix} 0 & 0 \\ 0 & 0 \end{bmatrix}, \qquad \hat{G} = \begin{bmatrix} -5 & 2 \\ 5 & -1 \end{bmatrix} \implies \hat{\Sigma} = \begin{bmatrix} 29 & -27 \\ -27 & 26 \end{bmatrix} \tag{36}$$

*We have that the covariance of the observation is identical for both solutions (i.e., $\Sigma = \hat{\Sigma}$). In the first solution, we assume the structure $z_2 \rightarrow z_1$; in the second solution, we assume independent factors of variation. However, we still obtain the same observational distribution. This implies that the independent factors of variation represent observations equally as well as the causally related factors, and we cannot identify a unique solution.*

The following sections outline causal representation learning methods that rely on certain data-generating assumptions (i.e., observational, interventional, counterfactual) and weak supervision signals to remedy the above unidentifiability issue and achieve disentanglement of causal factors. The majority of causal representation learning methods utilize VAE-based frameworks and maximize a suitable ELBO loss.

## 3.2 Learning from Static Observational Data

One approach to learning causal representations relies on purely observational data from $L_1$ of Pearl's Causal Hierarchy. This could be in the form of supervised labels or potentially from different sources. The following methods utilize such auxiliary information to provably disentangle the causal factors in this paradigm.

### 3.2.1 CausalVAE

**Learning Framework.** Yang et al. (2021) propose CausalVAE, a framework to learn causal representations and causal structure simultaneously given auxiliary information in the form of labels $y$ corresponding to causal factors of interest $z$. CausalVAE assumes a *linear* SCM with additive noise describing the structure between causal variables, parameterized by a causal adjacency matrix $A$, as follows:

$$z = A^T z + \epsilon = (I - A^T)^{-1}\epsilon \tag{37}$$

where $\epsilon \sim \mathcal{N}(0, I)$. CausalVAE proposes to encode high-dimensional data into low-dimensional noise variables $\epsilon$ and utilize an analytical mapping, $(I - A^T)$, to project the noise variables to causal variables $z$. The weighted adjacency matrix $A$ is learned via an acyclicity constraint and the augmented Lagrangian method (Yu et al., 2019). For identifiability, the labels $y$ are incorporated into a conditional prior, similar to Khemakhem et al. (2020), defined as follows:

$$p(z|y) = \prod_{i=1}^{n} p(z_i|y_i), \qquad p(z_i|y_i) = \mathcal{N}(\lambda_1(y_i), \lambda_2^2(y_i)) \tag{38}$$

where the latent variables $z$ are assumed to be mutually independent when conditioned on their corresponding labels. The conditional prior is used to regularize the posterior over the latent variables using the auxiliary

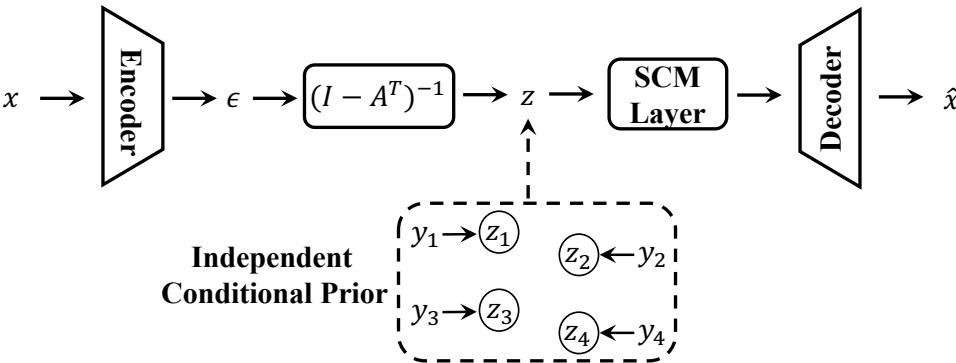

Figure 8: CausalVAE Framework (Yang et al., 2021)

labels as a supervision signal. During inference, counterfactual samples can be generated by encoding an image to an exogenous noise term, transforming it to causal variables, intervening on a dimension of the causal representation, propagating effects, and pushing it through the trained decoder to generate a counterfactual instance. Although CausalVAE is capable of learning causal representations and causal structure simultaneously, the linear SCM assumption is often unrealistic in practice. Further, continuous optimization through the acyclicity constraint (Zheng et al., 2018; Yu et al., 2019; Ng et al., 2022) does not necessarily guarantee an accurate learned DAG (recoverable only up to a supergraph). The quality of the representation learned is evaluated using the mutual information coefficient (MIC), a mutual information measure, which does not capture the disentanglement of causal factors. The overall design of the CausalVAE framework is shown in Figure 8.

**Identifiability Result.** The authors show that the labels enable the identifiability of causal factors up to component-wise reparameterization. They extend the identifiability result from Khemakhem et al. (2020) and show the identifiability of their framework when given the auxiliary label information.

### 3.2.2 DEAR

**Learning Framework.** Shen et al. (2022) propose disentangled generative causal representation learning (DEAR), a more general bidirectional generative model (BGM) to learn causal representations by adding a regularization term in the form of a GAN loss (Figure 9). Similar to other approaches, the DEAR framework requires access to all causal variables. They propose an SCM prior that can derive causal variables based on known causal orderings and a supergraph of the true causal graph. The authors propose the following cross-entropy loss-based supervised regularizer to learn to predict each generative factor, thereby taking the place of a Gaussian prior, and disentangle the causal factors

$$\mathcal{L}_{\text{sup}} = \sum_{i=1}^{n} \mathbb{E}_{x,z}[\text{CrossEntropyLoss}(y_i, z_i)] \tag{39}$$

Similar to Yang et al. (2021) and Komanduri et al. (2023), each latent $z_i$ is assumed to be supervised by its corresponding annotated label $y_i$ of each ground-truth factor, which they use to show the identifiability of latent factors up to component-wise correspondence. To learn *causal* representations, the authors propose to learn a nonlinear SCM prior, where causal orderings are given apriori (similar to Komanduri et al. (2023)). The SCM is assumed to follow a post-nonlinear additive noise model, with learnable parameters $\beta = (f, h, A)$, as follows:

$$z = F_\beta(\epsilon) = f((I - A^T)^{-1}h(\epsilon)) \tag{40}$$

where $f$ and $h$ are elementwise nonlinear transformations and $f$ is assumed to be invertible. The causal ordering of the supergraph is used as an initialization for causal structure learning using the acyclicity constraint. Unlike CausalVAE, DEAR does not learn intermediate noise encodings. Rather, noise is arbitrarily sampled from a standard Gaussian and mapped to generate causal variables through the learned structural assignments. The authors propose a generative model $p_\theta(x, z) = p_\theta(x|z)p_\beta(z)$ and a stochastic encoder

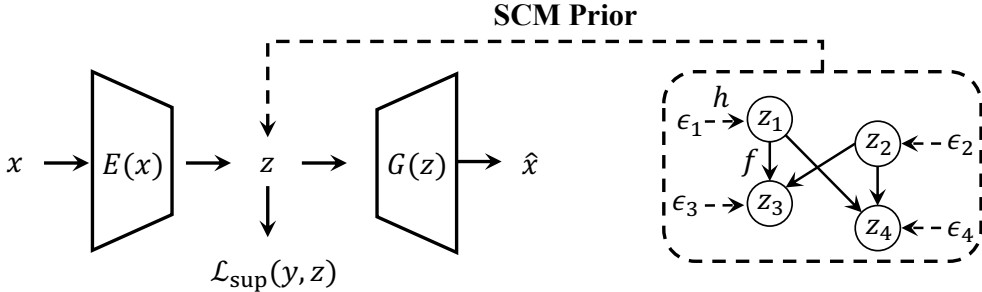

Figure 9: DEAR Framework ([Shen et al., 2022](#))

$q_\phi(z|x)$ to learn the posterior over the latent causal variables. The objective of the generative model is to minimize the following KL-divergence

$$\mathcal{L}_{gen} = \mathcal{D}_{KL}(q(x,z), p(x,z)) \tag{41}$$

with the following variational lower bound

$$\mathbb{E}_{x \sim q_x}\mathbb{E}_{q(z|x)}[\log p(x|z) + \mathcal{D}_{KL}(q(z|x)||p_\beta(z))] \tag{42}$$

where $p_\beta(z)$ is the learned SCM prior over causal variables $z$. DEAR uses a GAN-based method to adversarially estimate the gradients w.r.t the encoder, decoder (generator), and SCM prior with respective learnable parameters $\phi$, $\theta$, and $\beta$, through specified gradient formulas. The overall loss objective of the generative model and supervised regularizer can be formulated as follows:

$$\mathcal{L} = \mathcal{L}_{gen} + \lambda \mathcal{L}_{sup} \tag{43}$$

In order to tractably estimate the gradients of $\mathcal{L}_{gen}$, the authors adopt an adversarial approach using GANs. They train a discriminator $D$ using logistic regression and optimize the following objective

$$\min_{D'} \frac{1}{N_d}\left[\sum_i \log(1 + e^{-D'(x_i, z_i)}) + \sum_i \log(1 + e^{D'(x_i, z_i)})\right] \tag{44}$$

**Identifiability Result.** The authors provide theoretical analysis suggesting that learning from an independent prior will always lead to an encoder yielding an entangled solution. They also show the identifiability of DEAR up to component-wise correspondences since they assume access to labels as auxiliary information. Although representations learned using the DEAR method are shown to induce high distributional robustness, there is no quantitative evaluation of the disentanglement of causal factors.

### 3.2.3 SCM-VAE

**Learning Framework.** [Komanduri et al. (2022)](#) propose SCM-VAE, a framework for learning causal representations assuming access to the causal structure. SCM-VAE attempts to remedy the issues from CausalVAE by proposing to learn a *post-nonlinear* additive noise SCM describing the structure between causal variables

$$z_i = g(A_i \circ z) + \epsilon_i, \quad \forall i \in \{1, \ldots, n\} \tag{45}$$

where $g$ is some nonlinear function. Further, $A$ is assumed to be a binary unweighted adjacency matrix and $A_i$ is the $i$th column of $A$. SCM-VAE incorporates a structural causal prior that induces a causal-like factorization of labels

$$p_\theta(z|y) = \prod_{i=1}^{n} p_\theta(z_i|y_i, y_{\mathbf{pa}_i}), \quad p_\theta(z_i|y_i, y_{\mathbf{pa}_i}) = \mathcal{N}\big(\lambda_1((A + I_{n \times n})_i \odot y), \lambda_2^2((A + I_{n \times n})_i \odot y)\big) \tag{46}$$

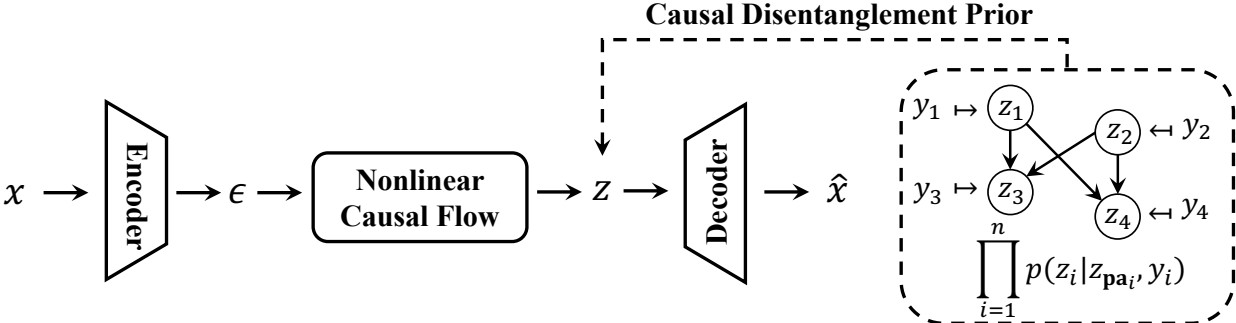

Figure 10: ICM-VAE Framework ([Komanduri et al., 2023](#))

By leveraging a causally factorized structural causal prior based on the known topological orderings of the causal graph, SCM-VAE addresses a concern of CausalVAE, where the conditional factorized prior simply assumes mutual independence among factors. One issue with SCM-VAE is that the learned mechanisms are not guaranteed to be bijective, which leads to issues in optimization and formulating the variational lower bound. Further, the structural causal prior only factorizes the labels, which may be redundant and can potentially induce entangled representations.

**Identifiability Result.** The authors show, similar to CausalVAE, that the supervision signal yields a one-to-one correspondence between learned factors and ground-truth factors. Thus, the causal factors are identifiable up to component-wise reparameterization.

### 3.2.4 ICM-VAE

**Learning Framework.** [Komanduri et al.](#) ([2023](#)) extend the work from [Khemakhem et al.](#) ([2020](#)) and [Yang et al.](#) ([2021](#)) and propose ICM-VAE, a framework of causal representation learning under supervision from labels. The ICM-VAE is based on learning a nonlinear *diffeomorphic* mapping $f_{RF} : \mathcal{E} \rightarrow \mathcal{Z}$ with causal mechanisms of the form

$$z_i = f_i(\epsilon_i; z_{\mathbf{pa}_i}) = \alpha(z_{\mathbf{pa}_i})\epsilon_i + \beta(z_{\mathbf{pa}_i}) \tag{47}$$

where $\alpha$ and $\beta$ are the neural network parameterized scale and shift parameters of a general nonlinear affine-form autoregressive flow, respectively. The diffeomorphic mechanisms are learned via a conditional autoregressive flow mapping a base noise distribution to a complex distribution over causal variables, similar to CAREFL ([Khemakhem et al.](#), [2021](#)). This choice is motivated by the efficient and exact likelihood estimation and the expressiveness of flow-based models in low-dimensional settings. The theory from [Khemakhem et al.](#) ([2020](#)) is extended to include causal mechanism equivalence towards a principled definition of causal disentanglement. That is, in a causal model, the standard notion of identifiability that yields marginal distribution equivalence is not sufficient to capture the equivalence of each individual mechanism between two models. To this end, the authors propose a causal disentanglement prior with a similar structure to the temporal prior from [Lippe et al.](#) ([2023a](#)) to *causally factorize* the latent space and learn causally disentangled mechanisms. The prior is formulated as follows:

$$p_\theta(z|y) = \prod_{i=1}^n p_\theta(z_i|z_{\mathbf{pa}_i}, y_i) = \prod_{i=1}^n p(y_i)\left|\frac{\partial \lambda_i(y_i; z_{\mathbf{pa}_i})}{\partial y_i}\right|^{-1} \tag{48}$$

$$p_\theta(z_i|z_{\mathbf{pa}_i}, y_i) = h_i(z_i)\exp(T_i(z_i|z_{\mathbf{pa}_i})\lambda_i(A_i \odot z, y_i) - \psi_i(z, y)) \tag{49}$$

where $\lambda$ is defined as an autoregressive causal flow that derives the natural parameters of $z_i$ as a function of the label $y_i$ and its parents $z_{\mathbf{pa}_i}$. Note that in this causally factorized prior, each factor $z_i$ can be identified due to the access to auxiliary information $y_i$. The ICM-VAE framework is shown in Figure 10.

**Identifiability Result.** The authors define a new notion of causal mechanism disentanglement and show that the causal factors are identifiable up to permutation and elementwise reparameterization. The intuition

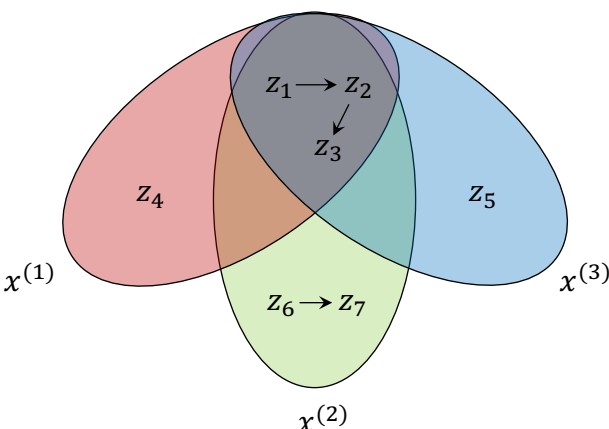

Figure 11: Unpaired Multi-domain Causal Representation Learning (Sturma et al., 2023)

is that identifiable models only guarantee the equivalence of the marginal distribution of the data and latent variables up to some tolerable ambiguity. However, the theory from Khemakhem et al. (2020) ignores the case where latent variables could be causally related and induces a Markov factorization. The authors in this work propose *causal mechanism identifiability* as a reformulation of permutation-equivalent identifiability to take into account causal conditionals in the Markov factorization. That is, causal mechanism identifiability requires the equivalence of causal mechanisms up to some tolerable ambiguity and is a *sufficient* condition for disentangling the causal factors. The mechanism equivalence definition proposed by Komanduri et al. (2023) is fundamentally different than the IMA theory introduced by Gresele et al. (2021) since IMA falls into the mixing function restriction class of methods rather than latent conditioning (Khemakhem et al., 2020).

### 3.2.5   Learning from Multi-Domain Observational Data

**Problem Setting.** Sturma et al. (2023) study the setting where one has access to observational data from multiple domains that may share an underlying causal representation (i.e., causal information should be invariant regardless of the view it is measured in). The observations from different domains are assumed to be *unpaired* (i.e., only the marginal distribution of each domain is observed but not the joint distribution). The setup is that the latent variables $z$ are sampled from some distribution $p(z)$, where $p(z)$ is determined by an unknown SCM among latent variables. In each domain $e \in \{1, \ldots, d'\}$, some domain-specific causal latents $z_{S_e}$ are mapped to the observed vector $x^{(e)}$ via a domain-specific (injective) mixing function $g_e$ such that

$$x^{(e)} = g_e(z_{S_e}) \tag{50}$$

where $S_e \subseteq \mathcal{H}$ is a subset of indices. The general idea is that shared latent variables capture the key causal relations, and different domains give combined information about the relations. That is, the multi-domain setup "completes the picture" of the causal model in some sense since latent variables captured in one domain may not be able to be captured in another (e.g., causal variables inferred from an imaging modality may not be inferred in sequence modality) and the relations can arise from different domains, as shown in Figure 11. Thus, when we put the domains together, we can identify the joint distribution of the latent variables.

**Identifiability Result.** The authors provide sufficient conditions for uniquely identifying the joint distribution of the causal factors and the causal graph, where the causal variables are described by a linear SCM. Assuming that (1) the marginal distributions are non-degenerate, non-symmetric, with unit variance, a genericity assumption that the errors must be pairwise different, and (2) the (linear) mixing function is full-rank, the authors show identifiability up to permutation. The first assumption restricts symmetric distributions and ensures that the distribution of errors is non-Gaussian. This allows for the application of the linear ICA identifiability theory from Comon (1994) in each domain individually. The second assumption requires that for each shared latent node there is at least a node in every domain such that the latent node

is a parent of the domain-specific node, as illustrated in Figure 11. However, this work does not address the more realistic situation where the mixing function is nonlinear. One direction to potentially enable identifiability in this setting is assuming access to some domain label for each domain and utilizing this augmented information to apply nonlinear ICA results (Hyvarinen et al., 2018) in each domain to identify latent factors.

Ahuja et al. (2024) study CRL in multi-domain settings from weak distributional invariances. This work is inspired by the invariance principle in Arjovsky et al. (2020), which requires that a fixed subset of latents is unintervened across domains while its distributional properties remain invariant. Such a property often holds when each domain comes from a multi-node imperfect intervention. Typically, the distributional properties of a representation can be partitioned into a set of stable (invariant) latents and unstable latents. To learn this invariance, the authors propose to train autoencoders with a class of invariance constraints to disentangle the stable latents from the unstable latents. Further, the authors in this work show block-affine identifiability of the underlying causal factors from a sufficiently large number of domains and specific invariance constraints without any interventional data.

### 3.2.6 Grouped Causal Representation Learning

**Problem Setting.** Morioka & Hyvärinen (2023) propose Grouped Causal Representation Learning (G-CaRL), a causal representation learning framework to disentangle causal variables from grouped high-dimensional observations without the need for additional supervision or interventions. The authors consider a data-generating process with data coming from $M$ groups (e.g., $M$ distinct sensor measurements), where the observational (diffeomorphic) mixing can be separated into $M > 1$ *non-overlapping* groups with separate (diffeomorphic) mixing functions and the number of variables can vary arbitrarily across groups. Thus, the observational model is formulated as follows

$$x = [x^1, \ldots, x^M] = \mathbf{g}([z^1, \ldots, z^M]) = [g^1(z^1), \ldots, g^M(z^M)] \tag{51}$$

where $g^m$ is a group-specific mixing function. In this setting, the latent variables $z^m$ in each group $m$ are multidimensional and can be causally related in two ways: (1) $z^m$ and $z^{m'}$ are causally related for $m \neq m'$ or (2) variables within $z^m$ are causally related. This strategy is employed so that the causal relations are complex enough to rule out the independence of latent variables. Notably, this formulation is not restricted to directed acyclic graphs and allows cycles to exist. However, the relations must be asymmetric. The graphical model for the G-CaRL setting is shown in Figure 12a.

**Identifiability Result.** The authors show that if (1) each variable in group $m$ has at least one neighbor in some other group and there is a sufficient dependency of the derivatives of the causal mechanisms on their inputs, then the latents in group $m$ (i.e., $z^m$) can be recovered up to permutation and elementwise invertible transformation. Given the identifiability of the latent variables, the authors also show that the learned causal graph, extracted from the latent variables via a post-processing causal discovery algorithm, is identifiable if (1) the intergroup causal relations are directed, and (2) each variable has a co-parent and co-child in the same group.

**Learning Framework.** The authors propose a self-supervised learning algorithm, G-CaRL, to learn the causal factors. The general idea is to use contrastive learning to discriminate between the original dataset $x^{(n)}$ and a shuffled version $x^{(n_*)}$, where $n$ is the sample index and $n_*$ is a shuffled index. The sample $x^{(n)}$ consists of $M$ groups known in advance. The elements of the sample $x^{(n_*)}$ are assigned to each of the $M$ groups by randomly selecting a sample index for each group $m$. The authors propose using a logistic regression classifier to distinguish between the two classes. The algorithm supports either joint learning of the causal graph or extracting it as a post-processing step.

### 3.2.7 Multi-view Causal Representation Learning with Partial Observability

**Problem Setting.** Extending work in multi-view nonlinear ICA (Gresele et al., 2019) and disentanglement, Yao et al. (2023) study the identifiability of representations learned from multiple views/modalities. We have observations from $K$ different views (e.g., measurements from different sensors) with $N$ underlying causal variables of interest. The authors consider a partially observed setting where each view constitutes a different nonlinear mixture of a subset of underlying latent variables (diffeomorphic mixing), as shown in Figure 12b.

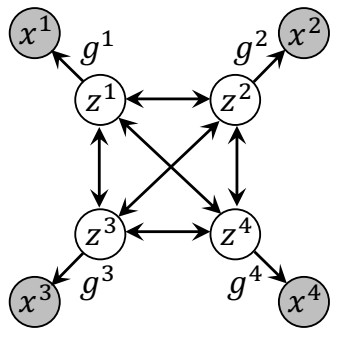

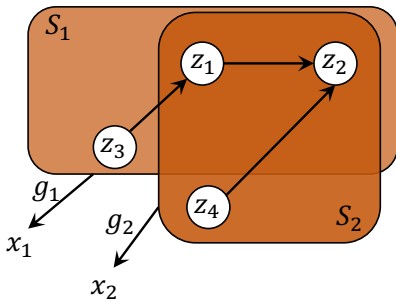

(a) G-CaRL setting with $M = 4$        (b) Multiview CRL: 2 views and 4 causal variables

Figure 12: Data-generating process of (a) G-CaRL (Morioka & Hyvärinen, 2023) and (b) Multiview CRL (Yao et al., 2023)

The key idea here is that a subset of latent variables corresponds to each modality. For example, some latent factors could capture information about text modalities while others capture information about image modalities.

**Identifiability Result.** The main identifiability result in this paper is that the factors shared across all subsets of *any number of* views are block-identifiable up to a smooth bijection by using a contrastive learning objective. For example, in Figure 12b, the intersection of views $S_1$ and $S_2$ enables block identifiability of latent variables $z_1$ and $z_2$. Further, as we add more views, the intersection of latents becomes smaller and thus we can identify the latent factors more precisely. In the naive case, we can train a pair of encoders and maximize content alignment between any pair of views via contrastive learning. This would allow one to achieve identifiability for the intersection of those views. Similarly, by induction, we can identify the intersection of a pair of blocks that have already been identified. However, the described approach requires one to train many encoders and thus is impractical. Ideally, we desire one encoder for each view. To achieve this, we can enforce content alignment only on the subset of latents shared between a pair of views. This can be done by learning an indicator function that effectively selects the latent dimension one cares about when performing contrastive learning between a pair of views. In a multimodal setup, this work generalizes work done by von Kügelgen et al. (2021) to the case where we can identify factors between any arbitrary intersection of subset of views. Xu et al. (2023) consider a similar partially observed setting where the observations are *unpaired*, but there exists instance-dependent partial observability pattern, and show permutation-equivalent identifiability of the partially observed latent variables under a linear mixing function regime.

### 3.3 Learning from Temporal Observational Data

Many real-world systems typically evolve through time, where certain events in the past may trigger events in the future following the arrow of time. This has inspired research in learning causal representations when we have access to time series data. The following sections explore observational identifiable causal representation learning where latent factors corresponding to high-dimensional temporal data can be temporally causally related. The general setting for observational temporal CRL methods is shown in Figure 13.

#### 3.3.1 Latent Temporal Causal Process

**Problem Setting.** Yao et al. (2022b) propose Latent tEmporal cAusal Processes (LEAP), a framework for disentangled representation learning in the temporal setting where latent variables are temporally causally-related. The authors consider a data-generating process where we have access to temporally measured observations $x_1, \ldots, x_T$. We assume that there exists a time-invariant mixing function $g$ that maps some latent variable $z_t$ to the observations via $x_t = g(z_t)$, for all $t = 1, \ldots, T$. Furthermore, we assume that the latent variables are causally related by some time-lagged causal mechanism $z_{it} = f_i(z_{\mathbf{pa}_{it}}, \epsilon_{it})$, where $\mathbf{pa}_{it}$ denotes the causal parents of latent variable $z_{it}$ from some previous timestep and $\epsilon_{it}$ is the noise term

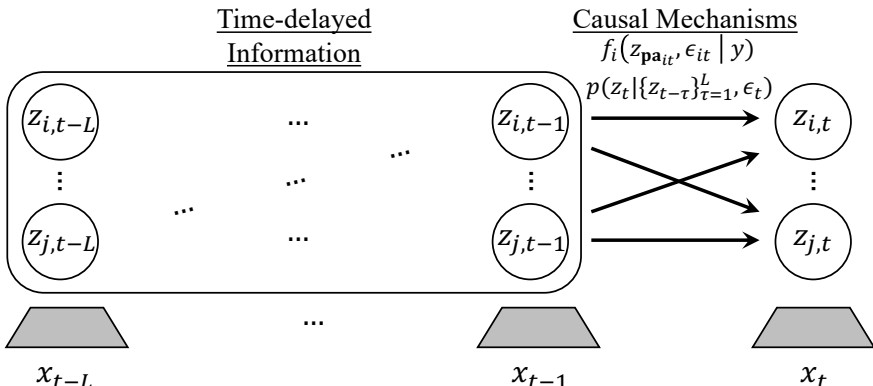

Figure 13: Observational Temporal CRL Setting

corresponding to latent variable $z_i$. In this work, the authors consider both parametric and nonparametric causal mechanisms in the nonstationary regime, where some side information $y$ (e.g., domain index) is observed for each time step.

**Identifiability Result.** The identifiability analysis extends work done in nonlinear ICA and the theoretical results from iVAE (Khemakhem et al., 2020) to the setting for temporally causally-related latent variables. The framework in iVAE shows that generative models can be rendered identifiable by introducing auxiliary information, such as class labels, time index, or domain index. However, the assumption in iVAE is that the latent factors of variation are independent. This work serves as an extension of iVAE in the temporal setting for causally related latent variables. The theoretical results presented utilize two central assumptions to show latent identifiability in the setting where latent variables are temporally causally-related: nonstationarity and functional constraints on temporal relations. The following are identifiability results under both nonparametric and parametric assumptions on the causal model.

Nonparametric. In the setting where the causal mechanisms are nonparametric, the nonstationarity assumption is necessary to enable identifiability. Specifically, the authors show that (1) if the noise distribution changes based on the observed auxiliary variable $y$ (i.e., $p_{\epsilon_i|y}$), (2) the noise $\epsilon_i$ are mutually independent in each regime $y$, and (3) there is sufficient variability of the latent transition function for different $y$, then the temporal latent causal factors are identifiable up to permutation and elementwise reparameterization.

Parametric. If the causal mechanisms are linear additive noise models (additive transitions), $\epsilon_{it}$ is from a generalized Laplacian distribution, and the state transitions are nonsingular (i.e., the state transition matrix is invertible for at least one time lag $\tau$), then the temporal latent causal factors are identifiable up to permutation and elementwise reparameterization.

**Learning Framework.** Based on the assumptions necessary for identifiability outlined above, the authors propose LEAP, a VAE-based framework to disentangle temporal causal factors. For $x_{1:T}$, LEAP learns an encoder that maps the high-dimensional observations to a low-dimensional latent variable $z_{1:T}$ via a bidirectional gated recurrent unit. The transition dynamics are modeled by a learned causal prior distribution. To enforce the independence of noise constraint in the framework, the authors propose learning the transition prior, $p(\hat{z}_t|\{\hat{z}_{t-\tau}\}_{\tau=1}^L)$, in terms of noise instead of latent causal variables by learning a flow-based mapping from causal variables to noise terms. That is, the function maps the latent causal variable at the current time step and its time-lagged causal parents to the corresponding noise term via $\hat{\epsilon}_{it} = r_i(\hat{z}_{it}, \hat{z}_{t-\tau})$ (parameterized by separate neural networks $r_i$). To estimate the noise distribution $p(\epsilon_i|y)$, the authors propose transforming standard Gaussian base distributions to noise terms via neural spline flows. To ensure that the estimated noises are mutually independent, the authors follow the strategy from FactorVAE (Kim & Mnih, 2018) and jointly train a discriminator to distinguish between estimated noise terms and randomly perturbed noise terms. The overall objective is to jointly train a VAE with (1) a suitable ELBO term, (2) regularize the latent posterior to follow the learned causal prior network, and (3) enforce independence among noise terms through time.

### 3.3.2 Temporally Disentangled Representation Learning

**Problem Setting.** Yao et al. (2022a) consider the same setting as LEAP (Yao et al., 2022b) but only focus on nonparametric transitions, stationary environments, and utilizing distribution shifts on stationary conditions to extend results to general nonstationary regimes. Under the stationary assumption, the mixing function $g$, transition functions $f_i$ (fixed causal dynamic), and noise distributions $p_{\epsilon_i}$ are considered to be invariant. Under a nonstationary regime, the authors consider each regime of data $y_r$ as a result of a distribution shift. The authors consider the following two violations of stationarity: (1) changing causal dynamics, and (2) changing global properties of the time series across domains (e.g., style of video). Finally, the authors consider a general nonstationary case by combining the fixed stationary regime with the two nonstationary regimes such that $z_t = (z_t^{\mathrm{fix}}, z_t^{\mathrm{chg}}, z_t^{\mathrm{obs}})$, where $z_t^{\mathrm{fix}}$ is the fixed dynamics, $z_t^{\mathrm{chg}}$ is changing dynamics, and $z_t^{\mathrm{obs}}$ is the global observation changes. Each of these latent partitions is captured by respective transition functions $[f_s, f_c, f_o]$.

**Identifiability Result.** In the stationary regime, where the causal dynamics are fixed, the authors show identifiability up to permutation and elementwise reparameterization if linear independence is satisfied among vectors of gradients of $\log p(z_{k,t}|z_{t-1})$ with respect to each latent $z_{i,t-1}$. In the nonstationary regime, where causal dynamics $p(z_{k,t}|z_{t-1})$ are changing across $m$ values of the context variable $y$ and $z_{k,t}$ are mutually independent conditioned on $z_{t-1}$, the authors show permutation-equivalent identifiability by showing linear independence of vector functions defined with respect to the gradients of the conditional distribution across the $m$ contexts. In the nonstationary regime where global observation changes occur (i.e., $p(z_{k,t}|y)$ changes across $m$ contexts), with the additional assumption that $z_t$ are mutually independent conditioned on $y$, the authors show permutation-equivalent identifiability. Finally, the authors show that under modular distribution shifts with stationary and the two nonstationary conditions, the partitioned latent factors $(z_t^{\mathrm{fix}}, z_t^{\mathrm{chg}}, z_t^{\mathrm{obs}})$ are jointly identifiable up to permutation and elementwise reparameterization.

**Learning Framework.** Based on the identifiability results, the authors propose TDRL, a VAE-based framework to recover the temporally causally-related factors of variation in the specified setting. The framework extends the sequential variational autoencoder with modules to model different distribution shifts. First, the authors embed the domain index $y_r$ into low-dimensional changing factors (dynamics and global properties) and use them as input to the inverse dynamics function $f_c^{-1}$ and observation function $f_o^{-1}$, respectively. Similar to LEAP, the inverse of the three transition functions (modeled as flows) map to random noise terms such that we can apply a change of variables to derive three transition priors $p(\hat{z}_{s,t}|\hat{z}_{\mathrm{time\text{-}lagged}})$, $p(\hat{z}_{c,t}|\hat{z}_{\mathrm{time\text{-}lagged}}, y_r)$, and $p(\hat{z}_{o,t}|\hat{z}_{\mathrm{time\text{-}lagged}}, y_r)$, where $\{s, c, o\}$ represents fixed, changing, and global observation dynamics, respectively. Similar to other VAE-based methods, the TDRL objective is to optimize a suitable ELBO using the joint transition prior $p(\hat{z}_t|\hat{z}_{\mathrm{time\text{-}lagged}}, y_r)$ to regularize the latent posterior.

### 3.3.3 Temporally Disentangled Representation Learning with Unknown Nonstationarity

**Problem Setting.** Song et al. (2023) consider the same setting as LEAP, but assume that the latent transition function is also a function of *unobserved* side-information (nonstationarity) $c$ such that $z_{it} = f_i(z_{\mathbf{pa}_{it}}, c_t, \epsilon_{it})$. Further, $c_1, \ldots, c_T$ are assumed to be sampled from a Markov chain based on the transition matrix.

**Identifiability Result.** First, the authors show that the nonstationarity $c$ is recoverable up to permutation and elementwise reparameterization assuming that the number of latent states for nonstationarity is unknown and the transition matrix is full rank. That is, the parameters of the Markov process are identifiable. Based on this condition, if we additionally assume that (1) the mixing function is bijective, (2) the latent components at the current timestep $z_t$ are conditionally independent given the latent variables at the previous time step $z_{t-1}$ and the latent nonstationary $c_t$, (3) and the latent transition function is sufficiently different for different $c_t$, then the authors show that the latent causal factors are identifiable up to permutation and elementwise reparameterization.

**Learning Framework.** The authors propose NCTRL, a VAE-based framework to learn disentangled latent temporally causally related factors. The framework consists of three modules. (1) The Prior Network Module is similar to the causal prior network from LEAP and estimates the latent transition prior distribution via inverse flow-based mapping to noise variables. (2) An autoregressive Hidden Markov Model (HMM) learns

to estimate the transition functions, the transition matrix, and utilizes the Viterbi algorithm (Forney, 1973) to obtain the optimal nonstationary states $\{\hat{c}_1, \ldots, \hat{c}_T\}$. (3) An encoder-decoder architecture is used to learn the latent causal factors.

## 3.4 Learning from Static Interventional Data

Instead of assuming access to observed labels or only observational data, several approaches leverage interventional data from $L_2$ of Pearl's hierarchy to learn identifiable causal representations. The following methods assume access to interventional data under perfect or imperfect interventions. This can often be a realistic assumption in practice since several domains, such as robotics, medicine, and genomics, have an abundance of interventional data available. The general intuition around using interventional data is as follows. Latent causal variables are generated by causal mechanisms as defined in a structural causal model. According to the ICM principle, the causal mechanisms must be independent and modular components. This implies that upon an intervention, only the manipulated causal mechanisms relating to the intervention actually change. Such *sparse* changes suggest that interventions are indeed critical to preserving the modularity property of causal mechanisms. Interventional data can thus be used to achieve sufficient variability in the observations for identifiability guarantees. The data generating process for the interventional paradigm is similar to Figure 15, where there may be observations sampled from several different environments (or contexts) that are induced by perfect or imperfect interventions on underlying causal variables.

Before we describe interventional CRL methods, we define the important notion of paired vs. unpaired interventional datasets. *Paired* interventional datasets are those where we couple environments (or datasets) that intervene on the same node. *Unpaired* interventional datasets are those where we do not know whether the same node was intervened in a pair of environments. In other words, in the unpaired setting, we have multiple interventional datasets, but we do not know which environments may have intervened on the same node. An example of unpaired interventional data is in biology, where for each cell, we can only obtain the measurement under a single intervention (Zhang et al., 2023b).

### 3.4.1 Interventional Causal Representation Learning

**Problem Setting.** Ahuja et al. (2023) explore to what extent access to interventional data can facilitate causal representation learning. The main contribution of this work is identifiability results in the specified observational and interventional data setting.

*Observational Setting.* The observational data-generating process they consider is as follows:

$$z \sim p(z), \qquad x = g(z) \tag{52}$$

where $x$ is an observational data point rendered from the underlying latent $z$ via an injective decoder $g$. The authors show that in the purely observational setting, we can identify the latent factors up to affine transformation (affine equivalence) if the mixing function is a finite-degree polynomial. Further, if we assume that the latents have independent support (Wang & Jordan, 2021), we can identify the factors up to permutation, shift, and scaling.

*Interventional Setting.* The interventional data-generating process is given by

$$z \sim \prod_{i \neq j} p(z_i | z_{\mathbf{pa}_i}) \delta_{Z_j = z_j}, \qquad x = g(z) \tag{53}$$

where $z$ is sampled from the distribution under intervention on causal factor $z_j$. Access to arbitrary interventional data is a more realistic scenario since interventions often act sparsely in the real world, and changes in causal conditionals lead to distribution shifts. Modeling distribution shifts through access to data from interventional distributions should intuitively be a sufficient weak supervision signal to learn accurate causal representations.

**Identifiability Result.** If we also assume access to interventional data from perfect *do-interventions*, the authors show identifiability of the *intervened* factors up to permutation, shift, and scaling and potentially up to componentwise correspondence if the mixing function is assumed to be diffeomorphic. The result

suggests that intervened-upon latents are identifiable up to permutation and the remaining latents up to affine transformation. This can be formulated as optimizing the following constrained autoencoder objective that performs reconstruction under the constraint that an arbitrary latent has been fixed to a certain unknown value

$$\min_{f,h} \ \mathbb{E}\left[\|h \circ f(x) - x\|^2\right], \ \text{s.t.} \ f_i(x) = \tilde{z} \tag{54}$$

where $(f, h)$ is an encoder-decoder pair and $f_i(x)$ is the $i$th latent factor with intervened value $\tilde{z}$. However, if we only assume *soft* interventions (i.e., imperfect interventions), the authors claim identifiability of the factors up to *block* affine transformation.

### 3.4.2 Learning Linear Causal Representations with Linear Mixing Function

**Problem Setting.** Squires et al. (2023) study identifiability and causal disentanglement when one has access to unpaired interventional data with single node interventions. The authors consider a setting with a linear injective mixing function $G$ (i.e., $x = Gz$) with a pseudoinverse $H$ and a linear Gaussian additive noise structural causal model. The data-generating setup in this work considers $n$ latent variables that are generated according to a linear SCM. Additionally, following the common assumption in identifiability that variables are observed across multiple contexts, the authors define several contexts (i.e., environments) that are a result of an intervention on a causal variable. These contexts are indexed by a variable $k \in \{0, \ldots, K\}$, where $k = 0$ is the observational context and $k > 0$ refers to an interventional context. In this setting, the mixing function is assumed to be invariant across contexts. The use-case considered in this work is modeling the internal state of a cell. There are complex interactions between the concentration of proteins, location of organelles, etc, and each context is an exposure to a different small molecule. Each molecule has a highly influential effect, changing only one cellular mechanism.

**Identifiability Result.** The main identifiability result in this work shows that having access to an intervened context for each of the $n$ causal variables is a sufficient condition for identifiability. Furthermore, in the worst case, they show that at least one intervention per latent causal variable is, in fact, a necessary condition for identifiability. The intuition behind CRL unidentifiability is shown in Example 3. If we extend this example to multiple contexts by defining a context-specific noise scaling such that the SCM is $z = A_k^T z + \Omega_k \epsilon$, where $A_k$ is the weighted adjacency matrix in context $k$ and $\Omega_k$ is diagonal with positive entries, it is clear that without interventional contexts for each latent variable, we collapse to a spurious ICA solution. That is, two different solutions $(\{A_k\}_{k=0}^{<n+1}, G, \{\Omega_k\})$ and $(\{\hat{A}_k\}, \hat{G}, \{\hat{\Omega}_k\}_{k=0}^{<n+1})$ can produce the same observational distribution (covariance of the observed data $X$) in all contexts. Thus, the authors show that any setting with access to less than $n + 1$ observed contexts renders the latent factors unidentifiable. In Example 3, this means that we must observe at least 3 contexts: one for the observational, one for an intervention on $z_1$, and one for an intervention on $z_2$ to identify the latent factors. In the general case, where the covariance of the data $x$ is rank-deficient (i.e., $n < d$), the authors propose to decompose the pseudoinverse of the covariance matrix of $x$ in each context via RQ decomposition to recover the mixing function. With the aforementioned assumptions, the authors show that the latent factors and invariant mixing function can be recovered up to permutation and scaling.

### 3.4.3 Learning Linear Causal Representations with General Mixing Function

**Problem Setting.** Buchholz et al. (2023) study causal representation learning where the mixing function is assumed to be general (e.g., deep neural networks) instead of restricting it to be linear (Squires et al., 2023) or polynomial (Ahuja et al., 2023). Consider a data-generating process with $n$ causal variables, where one has access to an observational distribution and $n$ unpaired interventional distributions (i.e., for each sample, we only obtain data under a single intervention), each a result of an intervention (unknown target) on exactly one causal variable. The latent variables are assumed to be described by a linear Gaussian SCM and the mixing function is injective. The setting considered in this work is quite similar to Squires et al. (2023) and generalizes the results to arbitrary mixing functions.

**Identifiability Result.** The main identifiability result in this work shows that the latent factors are identifiable up to permutation and scaling. Three main assumptions are required to keep the soundness of

the proposed identifiability theory. First, the number of interventions must be at least the dimension of the number of causal factors. A central assumption is that there must be *at least one intervention on each causal factor.* This assumption has been made in several other works such as Ahuja et al. (2023); Brehmer et al. (2022); Squires et al. (2023). The authors show that this is a necessary condition since a violation of this property renders even the weakest form of identifiability to break down. Second, the interventions must not be pure shift interventions. The authors show that if interventions are pure shift (relation to parents stays the same and only mean is changed), any causal graph is compatible with the observations and will induce an indistinguishable model (i.e., spurious solution). Lastly, similar to Yang et al. (2021), the latent variables are assumed to follow a linear SCM with Gaussian noise. Although post-nonlinear additive noise models have good approximability, an interesting direction is to explore identifiability when assuming nonlinear and non-additive noise models (Brehmer et al., 2022; Komanduri et al., 2023). However, it is not clear that the same theory would hold in the nonlinear setting.

**Learning Framework.** To implement interventional causal representation learning, the authors propose a novel contrastive learning approach for interventional learning. The goal is to train a deep neural network to distinguish observational samples from interventional samples. Although the identifiability results suggest that a model learning from the aforementioned data-generating process can properly disentangle the factors of variation, practical models are still lacking. The authors design a variational autoencoder-based approach that still relies on paired counterfactual data to tractably estimate the log-likelihood of the data.

### 3.4.4 Causal Disentanglement under Soft Interventions

**Problem Setting.** Zhang et al. (2023b) consider a setting where unpaired observational and interventional data is available with soft interventions on the mechanism of latent variables. In the limit of infinite data, the authors show affine and permutation equivalent identifiability under a certain set of conditions and a generalized notion of faithfulness. This work relaxes the assumptions of Squires et al. (2023) and considers polynomial mixing functions with a nonparametric latent causal model. The authors assume that the dataset includes samples from both observational and interventional distributions, where at least one intervention is required per latent node for identifiability. In other words, there must be a distribution (or dataset) induced as a result of an intervention of a latent node, for all latent nodes. Furthermore, the interventions are assumed to be unpaired, which means for each observation, one only obtains a measurement under a single intervention. Thus, this work serves as an extension of Squires et al. (2023) to *arbitrary* soft interventions. The authors run experiments on a biological genomics dataset, Perturb-seq (Norman et al., 2019), and utilize the framework to predict the effect of unseen combinations of genetic perturbations.

**Identifiability Result.** Zhang et al. (2023b) define an equivalence class, CD-equivalence, as the recovery of the latent variables, causal graph, and intervention targets up to some permutation. In addition to identifying the causal factors, we must also identify ancestral relations. In general, the interventional faithfulness assumption, which suggests that interventions on nodes will always change the marginals of all of its descendants, can be used to identify the descendants of an intervention target simply by testing if the marginal has changed. However, if we assume a linear mixing, the authors show that the interventional faithfulness assumption is not sufficient to detect the marginal distribution change in descendants when intervening on causal variables. Thus, they use a more general notion of faithfulness to avoid downstream effects being nullified by linear combinations of factors. The authors show that in the best case, only the ancestral relations and intervention targets are identifiable up to CD-equivalence.

**Learning Framework.** The authors propose Discrepancy-VAE, an autoencoding variational Bayes framework to disentangle causal factors using interventional data. The proposed framework utilizes two parallel encoders and a joint training objective.

Observational encoding. The observational encoder maps the high-dimensional observations $x$ to latent exogenous noise variables $\epsilon$. Then, the noise encodings are mapped to latent causal variables $z$ via some deep structural causal model $s$ that parameterizes the causal mechanisms between causal variables in the topological order of the causal graph. Finally, the mixing function, which is assumed to be a polynomial class, maps the causal latent variables back to observational space.

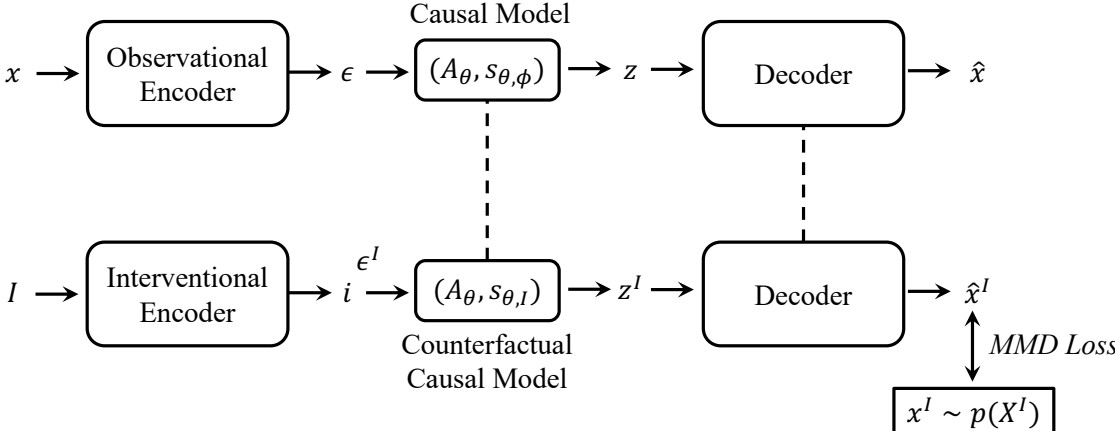

Figure 14: Discrepancy-VAE Framework (Zhang et al., 2023b)

Interventional encoding. The interventional encoder learns to map the intervention label $I$ to an intervention target $i$ via softmax normalization to approximate one-hot encoding of the intervention target (i.e., Gumbel-Softmax). Given the intervention target, an interventional deep SCM $s_I$ is learned to map the intervened noise encoding $\epsilon^I$ to the intervened causal variable $z^I$. Since the interventions act on the noise terms here, they are considered to be soft interventions (e.g., additive shifts). Accordingly, we get a "virtual" counterfactual $x^I$ once the (shared) mixing function decodes the intervened causal variables. To supervise the generated virtual counterfactual samples, the authors enforce a Maximum Mean Discrepancy (MMD) (Gretton et al., 2012) loss between the generated counterfactual and a sample from the interventional distribution $p(x^I)$. In this framework, the causal DAG is assumed to be unknown, so the authors place a sparsity constraint on the causal graph adjacency matrix to ensure that the causal relations are sparse. This is based on the intuition that the latent causal graph can be identified by using the sparsest possible model that fits the data.

### 3.4.5 Learning from Heterogeneous Multi-Environment Data with Unknown Interventions

Causal Component Analysis. Liang et al. (2023) study an intermediate problem between independent component analysis (ICA) and causal representation learning (CRL), known as causal component analysis (CauCA). CauCA is a generalization of ICA to the setting where the latent variables may have causal dependencies. A rigorous identifiability theory is formulated in a multi-environment setting, where data is collected from several (perfect or imperfect) interventional environments. In CauCA, both the causal graph and intervention targets are assumed to be known. Thus, any results from CauCA apply as a special case to ICA. However, any negative results from CauCA apply to CRL. That is, if certain tasks cannot be accomplished in CauCA when the causal graph is known, they certainly cannot be done in CRL when the graph is unknown. A key observation of this work is that if any pair of observational and interventional environments overlap, we can apply a measure-preserving transformation (e.g., Gaussian rotation) to the overlapping region to induce a different solution that is observationally equivalent, which implies unidentifiability of the latent factors. To rule out spurious solutions, the following *interventional discrepancy* requirement is proposed.

**Definition 7 (Interventional Discrepancy (Liang et al., 2023))** *Any pair of observational and interventional densities* $(p_i, \tilde{p}_i)$ *must be different almost everywhere:*

$$\frac{\partial}{\partial z_i}\left(\frac{\tilde{p}_i(z_i|z_{\tilde{\boldsymbol{pa}}_i})}{\tilde{p}_i(z_i|z_{\boldsymbol{pa}_i})}\right) \neq 0 \tag{55}$$

Nonparametric CRL. von Kügelgen et al. (2023) extend the CauCA framework to study CRL in a completely nonparametric setting, where the causal graph and intervention targets are both *unknown*.

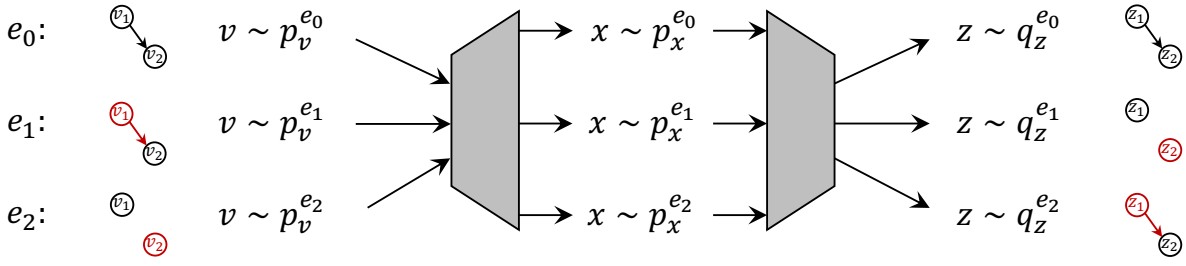

Figure 15: Multi-environment setup with single node perfect interventions and shared mixing function, where $z$ are the learned factors and $v$ are the underlying ground-truth (von Kügelgen et al., 2023). The red color indicates the intervened causal variable. We can see that the learned model (right) identifies the factors up to a permutation of the ground-truth factors (left).

**Problem Setting.** von Kügelgen et al. (2023) consider learning representations from heterogeneous data from multiple related distributions that arise from interventions in a shared underlying causal model. Since only a dataset of i.i.d observations cannot yield identifiability, the authors propose learning from multiple environments given access to heterogeneous data from multiple distinct distributions. This multi-environment data satisfies the assumption that certain parts of the causal generative process are shared across environments so that the environments do not vary arbitrarily. All environments share the same *invariant* diffeomorphic mixing function and underlying SCM, and any distribution shift occurs as a result of interventions on some causal mechanisms. Such an assumption is also consistent with the SMS hypothesis from Principle 2. The shifts across environments from this principle can be formulated as follows:

**Definition 8 (von Kügelgen et al. (2023))** *Each environment $e$ shares the same mixing function $g$ and each $p^e$ results from the same SCM through an intervention on a subset of mechanisms $\mathcal{I}^e \subseteq [n]$:*

$$p^e(z) = p^e(z_1, \ldots, z_n) = \prod_{i \in \mathcal{I}^e} p^e(z_i | z_{\boldsymbol{pa}_i}) \prod_{j \notin \mathcal{I}^e} p(z_j | z_{\boldsymbol{pa}_j}) \tag{56}$$

*where intervention targets $\mathcal{I}^e$ are unknown.*

**Identifiability Result.** The authors highlight that imperfect interventions, which change the mechanism but preserve parental dependence, are insufficient for full identifiability, as shown by Brehmer et al. (2022). This is because arbitrary imperfect interventions that preserve parents and only change the noise term produce a spurious ICA solution, which would be indistinguishable from the ground truth. Thus, they consider only perfect interventions (i.e., $\forall e$ and $i \in \mathcal{I}^e, p^e(z_i | z_{\mathbf{pa}_i}) = p^e(z_i)$). In the two-variable case, they show that environments from 1 perfect intervention per node are sufficient to identify the causal factors up to permutation and elementwise reparameterization. Figure 15 shows a simple 2-variable causal graph $v_1 \to v_2$, where $v_1$ and $v_2$ are the ground-truth causal variables. In this case, if one has an environment as a result of an intervention on $v_1$ and an environment as a result of an intervention on $v_2$, we can learn latent factors $z_1$ and $z_2$ that are identifiable up to permutation of the ground truth factors $v_1$ and $v_2$ and their causal structure. For a general number of latent causal variables, access to pairs[4] of environments corresponding to two distinct perfect interventions on each node is enough to guarantee identifiability up to permutation and elementwise reparameterization.

**Example 4 (2-variable environment setup)** *In the $n = 2$ case, we have 3 possible environments $\{p_x^{e_i}\}_{i=0}^2$, where $e_0$ is the observational environment and for $i \neq 0$, $e_i$ is the environment induced by a*

---

[4]Note that paired interventional datasets are fundamentally different than paired counterfactuals as studied in (Brehmer et al., 2022)

*perfect intervention on one of the two variables. That is, it could be $v_1$ or $v_2$ since the intervention target is unknown.*

**Example 5 ($n$-variable environment setup)** *In the general $n > 2$ variable case, we need at least one pair of distinct single-node perfect interventions per node to fully identify the causal factors. If $n = 4$, we have $2n$ environment pairs from two perfect interventions. That is, we would need the following 8 pairs: $e' = \{(e_i, e_i')\}_{i=1}^n$ and $e'' = \{(e_i, e_i'')\}_{i=1}^n$, where $e_i'$ is the result of an intervention on one of the $n$ variables and $e_i''$ is the result of another distinct intervention on the same variable.*

**Learning Framework.** The authors propose potential approaches for learning causal representations from finite interventional datasets sampled from $\{p_x^e\}_{e \in E}$, where $E$ is the collection of all environments. For instance, a VAE could be used to learn an encoder, causal graph, and intervention targets, similar to Brehmer et al. (2022), with $z_i \perp E \mid z_{\mathbf{pa}_i}$ for unintervened $i$. Another approach is to use normalizing flows to learn the mixing function and causal mechanisms.

### 3.4.6  Score-based Causal Representation Learning with Interventions

**Problem Setting.** Varıcı et al. (2023a) propose a score-based approach to causal representation learning. The general intuition is that parameterizing causal mechanisms with sufficiently complex neural networks is enough to capture the effect of interventions from changes in the score function, which is simply defined as the gradient of the log of the probability density. Thus, a valid transformation (or unmixing) renders the score function essentially invariant across different interventional environments. Such a property can be used to *perfectly* recover the latent DAG structure with soft interventions. The main motivation behind using score functions is that they often have sparse changes in coordinates upon a single-node intervention, which is consistent with the ICM principle.

**Nonparametric Identifiability.** Varıcı et al. (2023b) consider learning causal representations in a completely nonparametric setup where there are no parametric assumptions on the latent causal model nor the mixing function. The general idea is to obtain sufficient interventional diversity by having access to 2 interventional distributions for each latent causal variable. That is, for $n$ latent causal variables, we have $2n$ observational distributions for sufficient coverage.

Minimizing the score difference in the latent space gives us the correct encoder and latent causal variables. However, in practice, we only have access to high-dimensional observations. The authors derive the following relation of score difference between observation and latent space to show that minimizing score differences in the observational space will solve the same objective as that in the latent space up to the linear transformation defined by the Jacobian of the decoder.

$$s_{\hat{Z}}(\hat{z}) - s_{\hat{Z}}^m(\hat{z}) = [J_{\text{dec}}(\hat{z})]^T (s_X(x) - s_X^m(x)) \tag{57}$$

where $J_{\text{dec}}$ is the Jacobian of the decoder and $s_X^m(x) = \nabla \log p^{\text{Int}_m}(x)$ is the score of the $m$th interventional distribution. The authors then utilize score functions to solve the identifiability problem by minimizing

$$\|\mathbb{E}[|J_{\text{dec}}(x)(s^1(x) - s^2(x))|] + \cdots + \mathbb{E}[|J_{\text{dec}}(x)(s^{2n-1}(x) - s^{2n}(x))|]\|_0 + \text{Reconstruction Loss} \tag{58}$$

If we minimize the score difference for each pair of distributions, which is implemented via sliced score matching (Song et al., 2019b), then we can recover the true latent causal variables and the latent DAG up to permutation and elementwise reparameterization. In contrast to von Kügelgen et al. (2023), the authors assume access to $2n$ environments but do not assume knowledge of the correct coupling of observations corresponding to intervention on a target latent variable (i.e., unpaired interventional data).

**Parametric Identifiability.** As a special case of this result, Varıcı et al. (2023a) show that under (1) only single-node stochastic hard interventions, (2) linear mixing function, and (3) sufficiently nonlinear latent causal model, the latent variables and DAG are identifiable up to permutation and elementwise reparameterization. Furthermore, if we have access to one soft intervention instead of hard intervention per node, the latent DAG can be perfectly recovered and the recovered latent causal variables satisfy the Markov property. This result is similar to the result from Squires et al. (2023), which considers identifiability

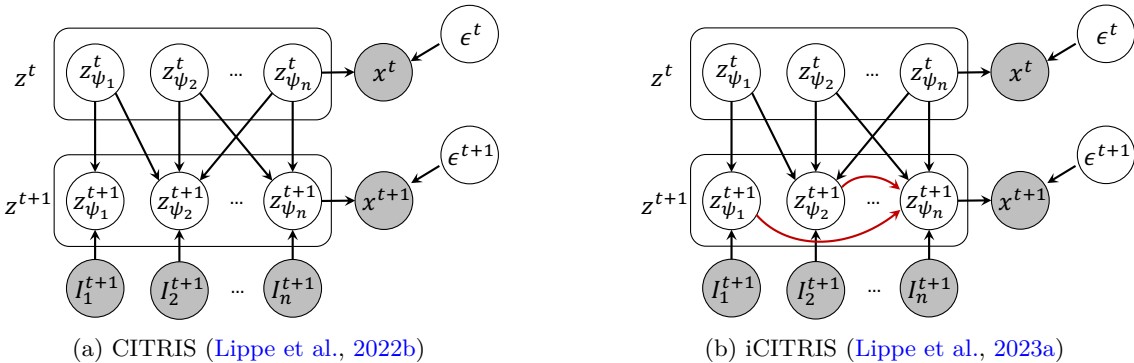

(a) CITRIS (Lippe et al., 2022b)          (b) iCITRIS (Lippe et al., 2023a)

Figure 16: Data-generating process of Causal Identifiability from Temporal Intervened Sequences

when both the mixing function and the latent causal model are linear and extends the result of recovering only ancestral relations as best case (Zhang et al., 2023b) to perfect DAG recovery by injecting sufficient nonlinearity in the latent causal model.

### 3.4.7 Extensions for Interventional Causal Representation Learning

Although most work in interventional CRL focuses on leveraging data from single-node interventions for identifiability guarantees, there has been some recent work that has generalized this idea to allow for multi-node interventions under a certain set of sparsity or invariance assumptions (Bing et al., 2024; Ahuja et al., 2024). Some work has also studied the extrapolation of results from identifiable CRL to downstream tasks to predict how interventions affect outcome variables, especially under novel interventions that are outside the support of the training data (Saengkyongam et al., 2023).

### 3.5 Learning from Temporally Intervened Data

Sometimes, causality can have a temporal interpretation. That is, causal factors in a dynamic system may evolve through time and be influenced by their values at previous time steps. Unlike methods in Section 3.3, which utilize only unintervened observational temporal data ($L_1$ data), the following methods consider learning causal representations in the temporal setting by leveraging interventional data in the form of intervention targets from $L_2$ of Pearl's hierarchy (i.e., an indicator of the factor being intervened on).

### 3.5.1 Causal Identifiability for Temporally Intervened Sequences

**Problem Setting.** Lippe et al. (2022b) present causal iden-tifiability for temporally intervened sequences (CITRIS), a temporal setting for causal representation learning. The authors consider a data-generating process with access to a sequence of images $\{x_t\}_{t=1}^T$ that represent temporal changes (interventions) to a system. The key assumption is that the causal variables at time step $z^{t+1}$ depend on some subset of causal variables at the previous time step $z^t$ along with an intervention target $I^{t+1}$ (specifying which latent is inter-vened on). Assuming $n$ *multidimensional* causal variables $(z_{\psi_1}^t, \ldots, z_{\psi_n}^t)_{t=1}^T$ following a dynamic Bayesian network with structural assignments of the form $z_{\psi_i}^t = f(z_{\psi_i^{\mathbf{pa}}}^t, \epsilon)$, each la-tent factor $z_i^t$ is assigned to one of the $n$ causal variable sets

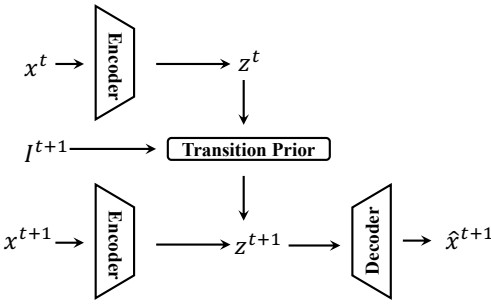

Figure 17: CITRIS/iCITRIS VAE Framework

via an assignment function $\psi$ learned via Gumbel-Softmax per latent variable (sampling latent-to-causal variable assignment). A set $z_{\psi_0}^t$ is also reserved for unintervened factors that do not affect anything (noise). The causal factors are assumed to satisfy causal sufficiency and the induced distribution satisfies faithfulness. The data-generating process of CITRIS is shown in Figure 16a.

**Learning Framework.** CITRIS is a VAE-based method that utilizes a prior over latent variables conditioned on the corresponding intervention target to learn to disentangle the causal factors. This ensures that intervened variables are no longer dependent on parents while unintervened variables preserve their parental dependencies. The observed data is of the form $\{x^t, x^{t+1}, I^{t+1}\}$. Notably, we have that the intervention targets are assumed to be known.

The temporal causal structure is time-invariant, so the parents are the same between all time steps. The authors also propose the notion of "minimal causal variables", which are latents in each causal set that split into invariable (independent of interventions) and variable (sensitive to interventions) parts. In other words, we are trying to find the minimum factors that are sensitive to interventions. Further, the following factorized prior distribution, called the *transition prior*, consisting of the current latent conditioned on the latent at the previous time step and intervention target, is proposed to disentangle the causal factors

$$p_\phi(z^{t+1}|z^t, I^{t+1}) = \prod_{i=0}^{n} p_\phi(z_{\psi_i}^{t+1}|z^t, I_i^{t+1}) \tag{59}$$

To include instantaneous effects, the authors extend CITRIS and propose iCITRIS (Lippe et al., 2023a), which can handle instantaneous causal relations within a timestep when given perfect interventions with known intervention targets (e.g., flicking a light switch and the bulb turning on can happen instantaneously). In this setting, both temporal and instantaneous causal relations exist, as shown in Figure 16b. To disentangle and identify the causal factors, the authors propose the following causally factorized transition prior

$$p_{\phi,A}(z^{t+1}|z^t, I^{t+1}) = p_\phi(z_{\psi_0}^{t+1}|z^t) \prod_{i=1}^{n} p_\phi(z_{\psi_i}^{t+1}|z^t, z_{\psi_i^{\mathbf{pa}}}^{t+1}, I_i^{t+1}) \tag{60}$$

where $A$ denotes the instantaneous causal graph. The VAE framework for CITRIS and iCITRIS is shown in Figure 17.

**Identifiability Result.** The identifiability analysis shows that although all causal factors are not identifiable in this setting, the minimal causal variables (intervention-sensitive components) are uniquely identifiable if we condition each latent variable on exactly one intervention target (in addition to instantaneous parents for iCITRIS). In this setting, the authors show that for every pair of causal variables, exactly one must be intervened on to achieve identifiability. If both causal variables are simultaneously intervened on or neither are intervened on, we lose the ability to identify the factors.

### 3.5.2 Binary Interactions for Causal Identifiability (BISCUIT)

**Problem Setting.** Lippe et al. (2023b) propose Binary Interactions for Causal Identifiability (BISCUIT), a method for causal representation learning from videos of dynamic interactive systems. The setup for BISCUIT is quite similar to CITRIS, where we have $n$ causal variables $(z_{\psi_1}^t, \ldots, z_{\psi_n}^t)_{t=1}^T$ in a dynamic Bayesian network, with an assignment function $\psi$ mapping latent factors to causal groups, as shown in Figure 18a. The authors assume that an agent's interaction with a causal variable can be described by an unknown binary interaction variable similar to the action variable in Lachapelle et al. (2022). Under this assumption, the causal variables of an environment are shown to be identifiable. BISCUIT is an extension of the CITRIS (Lippe et al., 2022b) framework to the setting where intervention targets are *unknown*.

**Learning Framework.** Given a temporal setting, BISCUIT is a VAE that encodes $x^{t+1}$ and $x^t$ to the latent space where each causal variable is encoded in a separate set of dimensions. The prior over the latent variables is conditioned on the previous frame and a known regime variable $R^t$ consisting of action information. Given the regime variable at each time step, a binary interaction target for each latent causal variable is predicted via a multilayer perceptron (MLP) that takes as input the action information and representation from the previous frame. This can be viewed as each causal variable having an observational ($I_i = 0$) and interventional ($I_i = 1$) mechanism. The transition prior for the model is defined as follows:

$$p_\omega(z^{t+1}|z^t, R^{t+1}) = \prod_{i=1}^{m} p_\phi(z_{\psi_i}^{t+1}|z^t, \mathrm{MLP}_\omega^{I_i}(R^{t+1}, z^t)) \tag{61}$$

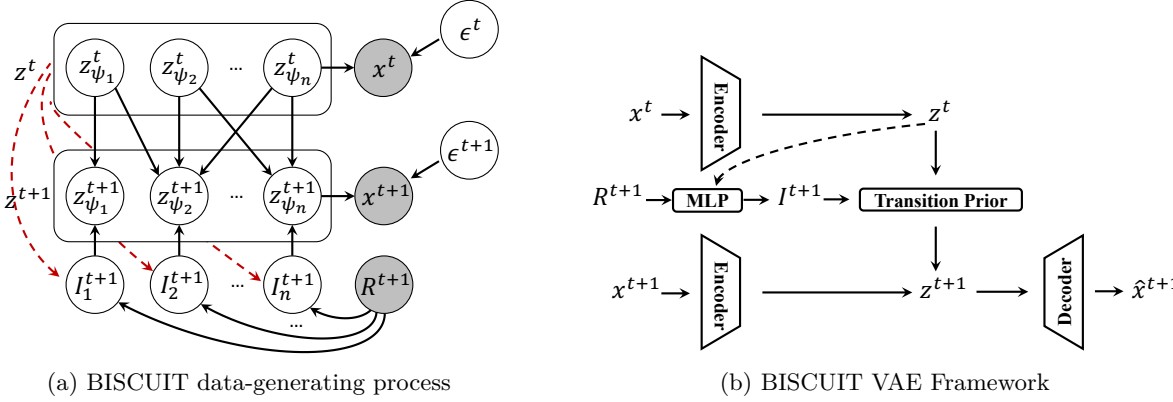

(a) BISCUIT data-generating process  (b) BISCUIT VAE Framework

Figure 18: Binary Interactions for Causal Identifiability (Lippe et al., 2023b)

where $m$ is the dimension of the latent space such that $m \gg n$ and $\omega$ denotes the parameters of the MLP. The "regime" variable and representation of the previous time step can be considered the auxiliary information that we condition on to achieve identifiability, consistent with Khemakhem et al. (2020). Thus, BISCUIT disentangles the causal factors. Their identifiability result mainly relies on the binary interaction variable $I_i^t$, which gives a distinct interaction pattern for a causal variable and a minimum of $\lfloor \log_2 n \rfloor + 2$ actions. This means that an agent interacts with each causal variable in a distinct pattern and does not always interact with any two causal variables at the same time. If $I_i^t = 0$, they claim the unidentifiability result of nonlinear ICA (Hyvärinen & Pajunen, 1999). The overall VAE framework for BISCUIT is shown in Figure 18b. Similar to other causal representation learning models, during inference, BISCUIT supports the manipulation of the latent codes to generate counterfactual instances.

**Identifiability Result.** In short, the authors show that if two models have the same likelihood, each causal variable has a distinct interaction pattern, and sufficient dynamics and temporal variability conditions are met (similar to Lachapelle et al. (2022); Komanduri et al. (2023)), then a learned model identifies the true model up to permutation and componentwise invertible transformation.

### 3.5.3 Mechanism-Sparsity Regularization

**Problem Setting.** Lachapelle et al. (2022) propose a new theory of identifiability from the perspective of temporal dynamics. They explore a similar setting to Lippe et al. (2022b), where they consider a sequence of observations $\{x^t\}_{t=1}^T$ that can be explained by semantically meaningful latent variables $\{z^t\}_{t=1}^T$ and a sequence of corresponding auxiliary variables in the form of actions $\{a^t\}_{t=1}^T$ taken by an agent (i.e., an action that changes the state of the latents). Following the form of the conditionally factorial prior from Eq. (25) (Khemakhem et al., 2020), the authors propose a conditional prior where latent factors at a given time step are conditionally independent given their state at the previous time step and the action variable as follows:

$$p(z^t | z^{<t}, a^{<t}) = \prod_{i=1}^n p(z_i^t | z^{<t}, a^{<t}) \tag{62}$$

$$p(z_i^t | z^{<t}, a^{<t}) = h_i(z_i^t) \exp\{T_i(z_i^t)^T \lambda_i(A_i^z \odot z^{<t}, A_i^a \odot a^{<t}) - \psi_i(z^{<t}, a^{<t})\} \tag{63}$$

where $A^a$ denotes the action graph specifying the action variables that are the causal parents of $z$, $A^z$ specifies the causal parents of $z^t$ from the latents $z^{t-1}$ at the previous time step, $\psi_i$ is the normalizing constant, and $h_i, T_i, \lambda_i$ are defined as in Eq. (25). The authors propose a VAE-based method with the defined conditional prior to regularize latent mechanisms to be sparse via $A^z$ and $A^a$, which act as masks to select the direct parents of $z^t$. The key assumption behind their approach is that the relations between causal variables across time and the effect of action variables on the latent variables are both *sparse*. One limitation of their exploration is ignoring the case where there could be instantaneous causal effects within a time step, as considered by Lippe et al. (2023a).

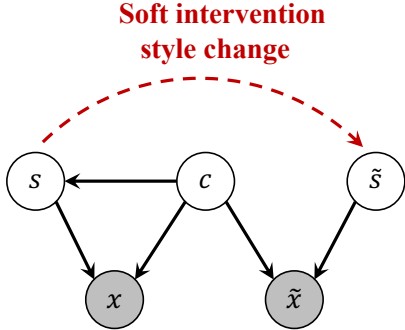

Figure 19: Problem formulation of self-supervised learning for causal representation learning

**Identifiability Result.** Under sufficiently varying natural parameter functions with respect to both temporal latent factors and actions, the authors show permutation-equivalent identifiability (disentanglement) of latent factors when only the mean is learned, the decoder is diffeomorphic onto its image, and the learned connectivity graphs are sufficiently sparse, which can be enforced by a sparsity constraint in causal discovery. Follow-up works also consider partial disentanglement by generalizing the specific graph criteria on the ground-truth causal graph (Lachapelle & Lacoste-Julien, 2022).

### 3.6   Learning from Paired Counterfactual Data

Another class of methods assumes access to counterfactual data from $L_3$ of Pearl's hierarchy to aid in learning identifiable causal representations. The following methods assume counterfactual data augmentations or contrastively paired counterfactual data as weak supervision to learn causal representations. Works such as Gresele et al. (2019) and Locatello et al. (2020) have taken a multi-view approach to nonlinear ICA and have indicated that learning a representation that represents multiple different views of the same underlying factors is a good inductive bias to learn disentangled representations.

#### 3.6.1   Self-supervised Learning with Data Augmentations

**Problem Setting.** von Kügelgen et al. (2021) study the role of data augmentations in self-supervised learning (SSL), specifically in the context of identifiable representation learning. SSL with data augmentations is closely related to identifiability in the multi-view setting (Gresele et al., 2019; Locatello et al., 2020), where we are given a second view $\tilde{x}$ of some observation $x$, resulting from a modified version $\tilde{z}$ of the underlying factors $z$. In this work, instead of applying transformations at the observation level (i.e., on $x$), as typically done in self-supervised learning, the authors propose a process of applying transformations in the representation space on $z$, where $z = g^{-1}(x)$, to obtain a modified $\tilde{z}$ which is decoded to obtain the augmented image $\tilde{x}$.

The authors propose a VAE-based objective. First, we train an encoder $g^{-1}$ using neural networks to obtain the representations $(z, \tilde{z})$ for the pair $(x, \tilde{x})$ and use a negative L2 loss as a similarity measure. The authors propose splitting the latent variable into a *content* part $c$, which would be invariant to augmentations and thus preserve semantic information, and a *style* part $s$, which captures the stochastic variation from the augmentations. Diverging from the typical assumption of independent factors of variation, the authors assume that there can exist any nontrivial statistical or causal dependencies in the latent space. Thus, a causal view of the data augmentation process is counterfactuals generated under soft style interventions, as shown in Figure 19. Based on this assumption, we can define an underlying SCM on the content and style latents, with causal graph $c \rightarrow s$, as follows:

$$c = f_c(u_c), \qquad s = f_s(c, u_s), \qquad (u_c, u_s) \sim p_{u_c} \times p_{u_s} \qquad (64)$$

where $u_c, u_s$ are independent exogenous noise variables, and $f_c, f_s$ are deterministic functions. Then, the causal latents can be decoded into observations $x = g(c, s)$. Given the factual observation $x^F = g(c^F, s^F)$,

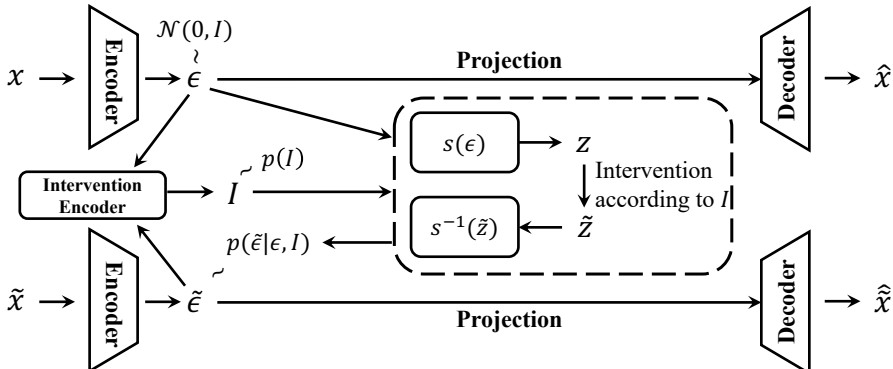

Figure 20: Implicit Latent Causal Model (ILCM) (Brehmer et al., 2022)

which resulted from $(u_c^F, u_s^F)$, we may ask the counterfactual question: "What would have happened if the style variables had been randomly perturbed without changing anything else?" Thus, we can consider a soft (imperfect) intervention on the style variable $s$, which changes the mechanism $f_s$ as follows:

$$\mathbf{do}(s = \tilde{f}_s(c, u_s, u_A)) \tag{65}$$

where $u_A$ is an additional source to account for the randomness of the augmentation. After decoding this intervened latent variable, we obtain a distribution over counterfactual observations $x^{CF} = g(c^F, s^{CF})$ that can be computed from the modified SCM by fixing the exogenous noise variables to their factual values and performing the soft intervention as follows:

$$c^{CF} = c^F, \qquad s^{CF} = \tilde{f}_s(c^F, u_s^F, u_A), \qquad u_A \sim p_{u_A} \tag{66}$$

**Identifiability Result.** In this setting, the authors prove that we can identify the invariant content partition $c$ under a weaker set of assumptions compared to existing identifiability results from nonlinear ICA. In this work, rather than disentangling each factor of variation, the focus is to disentangle the content from style (i.e., block identifiability). This theoretical guarantee has seen empirical successes in downstream tasks such as domain generalization and out-of-distribution applications (Chen et al., 2021; Mahajan et al., 2021).

### 3.6.2 Latent Causal Models

**Problem Setting.** Brehmer et al. (2022) consider contrastively paired *counterfactual* data and extend the work by Locatello et al. (2020) to the causal setting where the data pairs $(x, \tilde{x})$ represent a system before and after a random unknown, perfect, and sparse intervention on a causal variable, respectively. They propose Latent Causal Models (LCMs) and consider the following weakly supervised data-generating process.

**Definition 9 (Weakly-supervised counterfactually paired generative process)** *The weakly supervised generative process of counterfactual paired data $(x, \tilde{x})$ is defined as follows:*

$$\epsilon \sim p_{\mathcal{E}}, \qquad z = s(\epsilon), \qquad x \leftarrow g(z)$$
$$I \sim p_{\mathcal{I}}, \qquad \forall i \in I, \tilde{\epsilon}_i \sim p_{\tilde{\mathcal{E}}_i}, \qquad \forall i \notin I, \tilde{\epsilon}_i = \epsilon_i, \qquad \tilde{z} = \tilde{s}_I(\tilde{\epsilon}), \qquad \tilde{z} = s(\tilde{\epsilon}), \qquad \tilde{x} \leftarrow g(\tilde{z}) \tag{67}$$

*where $\epsilon$ is the pre-intervention noise, $\tilde{\epsilon}$ is the post-intervention noise, $z$ is the pre-intervention causal variables, $\tilde{z}$ is the post-intervention causal variables, $p_I$ is the distribution over intervention targets, $g$ is a diffeomorphic mixing function, and $s$ is a diffeomorphic solution function that maps the noise variables to causal variables.*

**Learning Framework.** *Implicit LCM (ILCM).* For the implicit, noise-based formulation, the pair $(x, \tilde{x})$ is mapped to a pair of noise encodings $(\epsilon, \tilde{\epsilon})$ using a projective noise encoder to learn a posterior $p_\theta(\epsilon, \tilde{\epsilon}|x, \tilde{x})$. In this formulation, $\tilde{\epsilon}$ is equivalent to the SCM noise term that would have generated the post-intervention

causal variable $\tilde{z}$ under the *unintervened* causal mechanisms. They propose learning an intervention encoder $p_\theta(I|\epsilon, \tilde{\epsilon})$ to predict the (single) intervention target based on the dimension-pair with the minimum KL-divergence. The pair of noise encodings are then restricted to differ only in the dimensions of the intervention target. The remaining dimensions are forced to be the same using different averaging schemes (Locatello et al., 2020; Hosoya, 2019; Bouchacourt et al., 2018). Then, the pre-intervention noise is mapped to the causal variable $z$ via a conditional flow $s$. The causal structure is inferred based on conditional independence testing to simulate causal parents. The causal structure is extracted from the learned causal variables after training the LCM. The noise-centric formulation aims to infer the noise distribution of the post-intervention data by performing a random intervention on the causal variable $z$ (based on the predicted intervention target) to yield $\tilde{z}$ and learning an inverse flow prior to then infer the noise variable $\tilde{\epsilon}$ for intervened dimensions. The post-intervention noise prior is defined as follows:

$$p(\tilde{\epsilon}|\epsilon, I) = \prod_{i \notin I} \delta(\tilde{\epsilon}_i - \epsilon_i) \prod_{i \in I} \tilde{p}(\bar{z}_i) \left| \frac{\partial \bar{z}_i}{\partial \tilde{\epsilon}_i} \right| \tag{68}$$

where $\bar{z}_i = s(\tilde{\epsilon}_i; \epsilon_{\mathbf{pa}_i})$ with base density $\tilde{p}(\bar{z}_i) = \mathcal{N}(0, 1)$ and $\delta$ is the Dirac delta function with a delta peak where the pre and post-intervention noise variables are equivalent. The overall architecture of the ILCM is shown in Figure 20.

*Explicit LCM (ELCM).* The authors also propose an explicit latent causal model that does not encode an intermediate noise variable and parameterizes all components of the SCM. In this formulation, the authors consider only the causal variables as the latents. The idea is to map the pair $(x, \tilde{x})$ directly to causal latents $(z, \tilde{z})$ via a stochastic encoder $q(z|x)$ and decoder $p(x|z)$. Further, the authors propose a prior $p(z, \tilde{z})$ that consists of a parameterized causal graph and causal mechanisms. The model is called as an explicit LCM as it directly parameterizes all components of an LCM and contains an explicit representation of the causal graph and causal mechanisms.

For both formulations, the authors utilize interventional causal discovery algorithms, such as ENCO (Lippe et al., 2022a), as a post-processing step to extract the latent causal structure from the learned causal variables. The authors note although optimally trained ELCMs correctly identify the causal structure and disentangle causal variables on simple dataset, jointly learning explicit graph and representations presents a challenging optimization problem.

**Identifiability Result.** The main identifiability result shows that LCMs are identifiable up to relabeling and elementwise reparameterizations of the causal variables under weak supervision from contrastively paired data samples. The authors prove this by leveraging graph isomorphisms between the true and learned causal models and showing that the entailed weakly supervised distributions must be equivalent. The main assumption behind the identifiability result is that paired data counterfactual data is available and interventions are perfect.

### 3.7 Other Works

Related to the methods discussed in this section, there have been alternate formulations of causal disentanglement in representation learning. Suter et al. (2019) consider a causal perspective of disentangled representation learning and define the notion of a *disentangled causal process* as shown in Figure 21. In this setting, the generative factors of variation $z_1, \ldots, z_n$ are independent when conditioned on a set of confounders $C$.

Reddy et al. (2022b) extend the disentangled causal process from Suter et al. (2019) to encoders (to study interventions) and generators (to study counterfactuals) of latent variable models. Further, the authors propose concrete metrics to account for confounding in the generative process. Induced spurious correlations among generative factors in observational data are often due to the existence of confounders. The authors propose two metrics to evaluate causal disentanglement in their setting: *unconfoundedness* and *counterfactual generativeness*. The unconfoundedness metric evaluates if the model is able to map the generative factor to a unique set of latent codes. The counterfactual generativeness metric is formulated as an average causal effect between counterfactuals generated by intervened factors and those generated by unintervened factors.

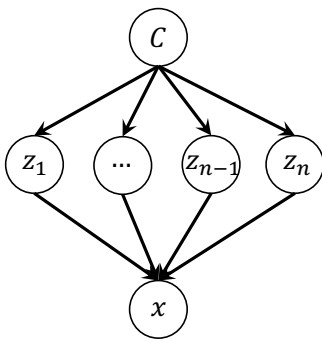

Figure 21: Disentangled Causal Process (Suter et al., 2019)

That is, this metric ensures that only interventions on the desired group of latent variables influence the generated image.

Based on the disentangled causal process definition, Wang & Jordan (2021) lay out desiderata for unsupervised disentanglement in representation learning. They propose an independence of support score (IOSS) as a metric to disentangle representations learned from observational data. The IOSS is defined as the Hausdorff distance, which measures the distance between two subsets of a metric space, between the joint support of the representation $\{z_1, \ldots, z_n\}$ and the product of each individual latent's support[5] (i.e., $d_H(\text{supp}(z_1, \ldots, z_n), \text{supp}(z_1) \times \text{supp}(z_2) \times \cdots \times \text{supp}(z_n)))$. This condition essentially enforces that each factor must take on a distinct set of values (i.e., the set of values that a factor can take does not depend on the values that a different factor can take). Enforcing IOSS can be seen as ensuring orthogonality between groups of latent factors and thus satisfying the disentanglement definition in Definition 3. The authors utilize the IOSS to prove the permutation-equivalent identifiability of representations under certain conditions.

### 3.8 Discussion

**Advantages.** Although representation learning has been used in some of the most impactful applications, its causal blindness prevents generalizability and interpretability. The primary advantage of learning a causal representation is its robust application to downstream tasks. A causal perspective of representation learning enables generalizability to distribution shifts. For instance, take the canonical example of an image of a cow on grass (Beery et al., 2018). A prediction task that utilizes representations learned from a dataset of images can easily fall into the trap of associating grass with cows (spurious correlation) and fail to perform well when it sees a cow on different terrain. However, *causal* representations are invariant to distribution shifts (Arjovsky et al., 2020). The intuition comes from the Sparse Mechanism Shift hypothesis (Principle 2), where distribution shifts can be attributed to sparse changes in causal conditionals in the Markov factorization. For instance, the causal representation could capture information about latent factors corresponding to the shape of the cow. Such information is invariant to what environment the image comes from. One could also learn more robust downstream models by utilizing causal representations to generate counterfactuals for data augmentation (von Kügelgen et al., 2021). Causal representations are also important in fairness applications (Locatello et al., 2019a; Zuo et al., 2023). Disentangled causal factors enable one to model dependencies between factors and perform interventions on sensitive factors of variation to observe their effect on other relevant factors. Thus, causal representations would have high utility in causal fairness settings when dealing with high-dimensional data such as images.

**Limitations.** As emphasized in previous sections, the identifiability of causal representations remains to be a great challenge. Most methods rely upon strong assumptions such as linear and additive noise latent SCMs, no unobserved confounding, and no misspecification of SCMs. Such assumptions make it quite difficult to use causal representation learning (CRL) methods in practice. Further, many CRL methods require access to interventional data, which may be infeasible to obtain in many cases. We detail more limitations of CRL in Section 8. Depending on the situation, methods from the observational, interventional, and counterfactual

---

[5]Note that support refers to the set of possible values a representation can take.

| Approach | Architecture | Causal Mechanism | Counterfactual Inference* | Key Features |
|---|---|---|---|---|
| CausalGAN (Kocaoglu et al., 2018) | GAN | causal generators | heuristic | interventional distribution sampling |
| CGN (Sauer & Geiger, 2021) | GAN | analytic composition | heuristic | models invariant (stable) features |
| CONIC (Reddy et al., 2022a) | GAN | data augmentation | heuristic | models confounding |
| DSCM (Pawlowski et al., 2020) | Flow | conditional flows | analytical | mechanism learning |
| CausalHVAE (De Sousa Ribeiro et al., 2023) | VAE | hierarchical VAE | analytical | models mediators |
| Diff-SCM (Sanchez & Tsaftaris, 2022) | Diffusion | stochastic differential equation | analytical | anti-causal predictor guidance |

\* Counterfactuals computed heuristically or analytically using Pearl's abduction, action, prediction steps (Pearl, 2009).

Table 3: Summary of Controllable Counterfactual Generation Methods

layers can be employed. For example, if one has access to data from multiple domains and domain labels, observational methods can be naturally applied. However, learning the causal graph jointly with the causal representation in this paradigm can prove to be quite challenging as we need to make restricting parametric assumptions on the causal model or assume knowledge of the causal ordering. In domains such as genomics or robotics, where perturbed data is feasible to obtain, interventional methods can be useful. Interventional CRL methods can often have nonparametric assumptions on the causal model (Zhang et al., 2023b; Varıcı et al., 2023b; von Kügelgen et al., 2023) and interventional data can be used to identify both the latent representation and causal relationships. However, for most domains, collecting interventional data can be infeasible due to ethical concerns or lack of interventional facilities. Thus, developing more accurate CRL models in the observational paradigm would be highly beneficial. Although some methods utilize counterfactual data to learn causal representations, for most real-world applications, obtaining counterfactual data is simply impossible. However, getting data that may be close to counterfactual can potentially be utilized to learn both latent factors and their causal dependencies. Overall, recent research efforts seem to suggest that methods utilizing a combination of observational and interventional data are quite effective.

## 4   Controllable Counterfactual Generation

The objective of *controllable generation* is to produce novel samples based on specified attributes rather than simply mimicking the characteristics of the training data in non-controllable generation. However, their ability to extrapolate beyond the training distribution is constrained because of their reliance on sampling from observational distribution. Controllable counterfactual generation incorporates encoded causal knowledge that can further generate samples in previously unobserved contexts and enable users to edit specific samples by intervening on targeted attributes, with downstream causal effects automatically incorporated. Although causal representation learning methods are also capable of controllable generation through latent space manipulations and decoding, the focus is primarily on learning robust and interpretable representations for downstream tasks. Counterfactual generation methods usually assume access to ground-truth causal information and thus have the advantage of not being plagued by unidentifiability issues. In contrast to representation learning, controllable counterfactual generation models often utilize GANs or diffusion models to generate realistic counterfactual images. Thus, such models are often more suited for real-world applications, where domain knowledge can be exploited to describe the causal dependencies of a system. We cover the main controllable counterfactual generation approaches proposed in recent years. A summary of the main methods is shown in Table 3.

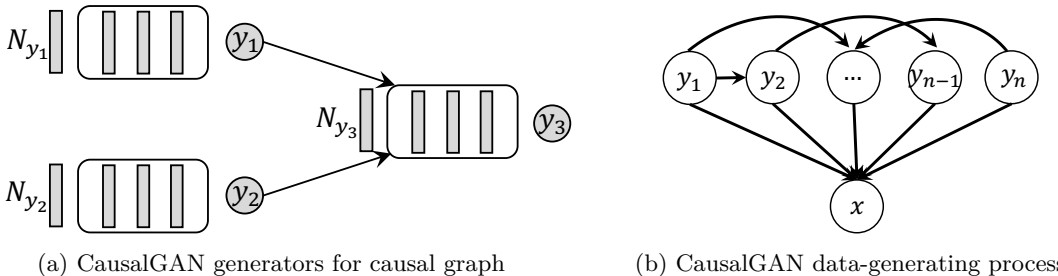

(a) CausalGAN generators for causal graph       (b) CausalGAN data-generating process

Figure 22: CausalGAN (Kocaoglu et al., 2018)

## 4.1 GAN-based

GANs have shown good performance in tasks involving sampling from data distributions. Although the standard GAN does not encode any causal information, it turns out we can use generators to learn mechanisms between causal variables and formulate mechanism learning as an adversarial problem. The following methods focus on utilizing GAN components to learn causal mechanisms from label space to image space and enable counterfactual generation.

### 4.1.1 CausalGAN

**Problem Setting.** As a modification of the ConditionalGAN (Mirza & Osindero, 2014) to support sampling from interventional distributions, Kocaoglu et al. (2018) propose CausalGAN. Traditional GANs have shown impressive performance in generating samples from a given data distribution. GANs are an example of implicit generative models, which can sample from data distributions but cannot estimate likelihoods for data points. However, GANs cannot sample from *interventional* distributions. Assuming that we have a dataset with labels, mutually independent labels can be thought of as a trivial causal graph. In this case, the ability to intervene on a variable and get novel samples is desirable. We assume the labels cause the image, as shown in Figure 22b. For instance, if color and species are independent labels, then `color=blue` should give us blue eagles and blue jays. However, if labels are causally related, such as `gender → mustache`, conditioning on the mustache will only give us images of males. However, *intervening* on mustache will give us males and females with mustaches. Interventions preserve ancestral distributions, whereas conditioning doesn't.

The main idea of CausalGAN is to model each label using a generator neural network that uses the causal parents of the label as its inputs. In this setting, the authors assume access to the complete causal graph among the labels. Since jointly training the labels and image generators is infeasible, the authors propose a two-stage framework: (1) train a generative model over only labels $y$ and (2) use the node labels in the causal graph to train a conditional generative model for images.

**Learning Framework.**
*Stage 1.* First, the authors propose to learn the Causal Controller, a causal implicit generative model between labels $y$ using a feedforward neural network for variables such that each variable is conditioned on the outputs of its parents' neural network satisfying structural equations of the form $v = f(v_{\mathrm{pa}}, \epsilon)$. The arrangement of the causal variable neural networks for the graph $y_1 \to y_3 \leftarrow y_2$ is shown in Figure 22a. This process can be split into two subtasks. First, we learn a causal implicit generative model over a small set of variables and then learn the remaining set of variables conditioned on the first set using a conditional generative network. Each node in the first set should come before any node in the second set. The Causal Controller module decides what distribution the image will be sampled from (interventional or observational) and generates labels according to the causal graph. For example, if we want to sample from the observational distribution, we can specify the values of each label. If we want to sample from the interventional distribution, we can fix the value of a specific label and propagate causal effects via the feedforward neural networks. This module is pre-trained using a Wasserstein GAN (Arjovsky et al., 2017) to produce binary class labels with a specific probability.

*Stage 2.* Let $p_{data}^l$ be the real data distribution between the image $x$ and the binary label $l \in \{0, 1\}$. Similarly, let $p_g^l$ denote the joint distribution between the labels given to the generator and the generated images. We also assume a perfect Causal Controller such that $p_{data}^1 = p_g^1 = \rho$, where $\rho$ is the probability of class label $l$. Conditioned on the label distribution from the Causal Controller, the CausalGAN learns an implicit class conditional generative model for images. There are three main components in the CausalGAN.

1. **Labeler.** The Labeler discriminator $D_{LR}$ is trained to estimate the labels of real images in the training dataset. The Labeler discriminator module solves the following optimization objective

$$\max_{D_{LR}} \rho \mathbb{E}_{x \sim p_{data}^1(x)}[\log(D_{LR}(x))] + (1 - \rho)\mathbb{E}_{x \sim p_{data}^0(x)}[\log(1 - D_{LR}(x))] \qquad (69)$$

2. **Anti-Labeler.** The Anti-Labeler discriminator $D_{LG}$ is trained to estimate the labels of the images sampled from the generator (labels produced by Causal Controller). The Anti-Labeler is especially used to prevent label-conditioned mode collapse. The Anti-Labeler module solves the following optimization objective

$$\max_{D_{LG}} \rho \mathbb{E}_{x \sim p_g^1(x)}[\log(D_{LG}(x))] + (1 - \rho)\mathbb{E}_{x \sim p_g^0(x)}[\log(1 - D_{LG}(x))] \qquad (70)$$

3. **Generator.** The generator produces realistic images conditioned on labels from the Causal Controller by minimizing the Labeler loss and avoids unrealistic image distributions that are easily labeled by maximizing the Anti-Labeler loss, thereby distinguishing between classes and images conditioned on those classes.

For a fixed generator, the discriminator $D$ solves the following:

$$\max_D \mathbb{E}_{x \sim p_{data}(x)}[\log(D(x))] + \mathbb{E}_{x \sim p_g(x)}\left[\log\left(\frac{1 - D(x)}{D(x)}\right)\right] \qquad (71)$$

For a fixed discriminator, Labeler, and Anti-Labeler, the generator solves the following:

$$\begin{aligned}
\min_G \mathbb{E}_{x \sim p_{data}(x)}[\log(D(x))] + \mathbb{E}_{x \sim p_g(x)}\left[\log\left(\frac{1 - D(x)}{D(x)}\right)\right] \\
-\rho \mathbb{E}_{x \sim p_g^1(x)}[\log(D_{LR}(X))] - (1 - \rho)\mathbb{E}_{x \sim p_g^0(x)}[\log(1 - D_{LR}(X))] \\
+\rho \mathbb{E}_{x \sim p_g^1(x)}[\log(D_{LG}(X))] - (1 - \rho)\mathbb{E}_{x \sim p_g^0(x)}[\log(1 - D_{LG}(X))]
\end{aligned} \qquad (72)$$

During inference time, one can sample from the interventional label distribution produced by the Causal Controller and generate novel counterfactual images unseen during training.

### 4.1.2 Counterfactual Generative Networks (CGN)

**Problem Setting.** Sauer & Geiger (2021) propose counterfactual generative networks (CGN), a generative model aimed at modeling images as a realization of shape, texture, and background variables, where spurious signals are strong predictors of the class label. In contrast to CausalGAN, CGN aims to obtain classifiers that are interpretable and robust to spurious correlations that arise as a result of some confounding variable $C$. The authors design a model that learns independent causal mechanisms $m = f_{shape}(\cdot)$, $f = f_{texture}(\cdot)$, $b = f_{background}(\cdot)$, and an analytical composition mechanism $g$ that combines the three mechanisms as follows:

$$\hat{x} = g(m, f, b) = m \odot f + (1 - m) \odot b \qquad (73)$$

Given some Gaussian noise and labels $y$ sampled uniformly as weak supervision (labels for high-level concepts like shape and background are hard to obtain), the architecture consists of BigGAN (Brock et al., 2019) backbone to model each mechanism as a generator. The overall data-generating process of CGN is illustrated in Figure 23a.

**Learning Framework.** A loss is specified for each of the mechanisms, which are then combined together by a composite function (analytical, not learned) to generate the image as a combination of shape, texture,

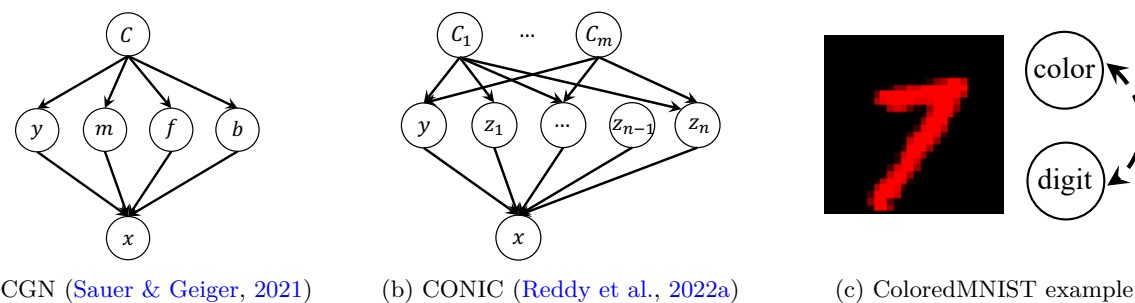

(a) CGN (Sauer & Geiger, 2021)  (b) CONIC (Reddy et al., 2022a)  (c) ColoredMNIST example

Figure 23: Data-generating process of (a) counterfactual generative networks (CGN) and (b) counterfactual generation under confounding (CONIC), and (c) an example of spurious correlation between digit and color from ColoredMNIST dataset

and background. To incorporate some form of a supervisory signal, the architecture proposes simultaneously learning a conditionalGAN to generate ground-truth images. A corresponding reconstruction loss is also defined between generated images from mechanisms and those from the conditionalGAN. After training the CGN, the authors propose learning an invariant classifier that is invariant of any spurious features such as texture and background and only relies on shape changes for prediction tasks by using counterfactually generated data to predict mechanism-specific labels. Take the canonical example of images of cows standing on a field. Through counterfactually generated data from the trained CGN model, the invariant classifier is effectively trained to use only shape attributes to predict labels. This is consistent with preventing the occurrence of spurious associations in distribution shifts, such as predicting 'cow' based on background spurious features. The authors empirically evaluate their framework on variants of MNIST and ImageNet, which are benchmarks, and their results seem to show that they are able to randomly sample noise and labels to controllably generate realistic counterfactuals. Since CGN utilizes a BigGAN (Brock et al., 2019) architecture to train each of the generative mechanisms, it can be computationally demanding, inefficient, and not generalizable. Further, it does not take into account confounding effects.

### 4.1.3 Counterfactual Generation under Confounding (CONIC)

**Problem Setting.** Reddy et al. (2022a) study the adversarial effect of confounding in observational data on a classifier's performance and how data augmentation using counterfactual data can be used to mitigate these confounding effects. The authors assume a causal generative process, based on Suter et al. (2019) and Reddy et al. (2022b), consisting of generative factors $z_1, \ldots, z_n$ (e.g., background, shape, texture, etc.) and some label $y$, that determine an observation $x$ through some unknown mechanism $f$. These variables are considered to be confounded by a set of confounders $C_1, \ldots, C_m$. The generative factors here are arbitrary and can be viewed as a generalization of the shape, background, and texture features from Counterfactual Generative Networks (Sauer & Geiger, 2021), and perhaps have more potential to robustly capture important information about the generative process. The authors propose Counterfactual Generation under Confounding (CONIC) to remove the effect of confounding, which can lead to spurious correlations, by performing counterfactual data augmentation.

**Learning Framework.** To quantify uncertainty, a connection can be made between the amount of confounding and the correlation between any pair of generative factors in the causal process from Figure 23b. The authors propose a KL-divergence metric between the conditional and post-interventional distribution of a pair of generative factors (i.e., $\forall (z_i, z_j), D_{KL}(p(z_i|z_j)\|p(z_i|\mathrm{do}(z_j))))$. In an ideal scenario with no confounding, we have that $p(z_i|\mathrm{do}(z_j)) = p(z_i|z_j)$. Thus, the variables are unconfounded if the KL divergence is close to 0. In general, the existence of confounding can be viewed through backdoor paths between factors of the form $z_l \leftarrow C_i \rightarrow z_j$, where $l \neq j$. To break this confounding, the authors lay out a counterfactual augmentation procedure using conditional CycleGANs (Zhu et al., 2017). The CycleGAN is a GAN model that is used predominantly in domain shift scenarios. The idea of CycleGAN is to use unpaired training data samples to learn a mapping between two domains $A$ and $B$, while preserving semantic information and satisfying the cycle consistency property (invertible mapping). In CONIC, the authors assume $A = X$ and

$B = X'$, where $X'$ is the counterfactual sample space generated as a result of intervening on a generative factor and producing an image.

Consider the ColoredMNIST benchmark dataset (Arjovsky et al., 2020), which adds colors to the digits and alters the strength of association between the color and digit label between the training data and test data, thereby inducing spurious correlations. We can enforce an extremely high correlation coefficient between the *color* and *digit* factors, thereby inducing confounding. For example, assume that if a digit is red, it is more likely to be the digit 7, as shown in Figure 23c. Suppose $z_j$ represents the color and $z_l$ is the digit class. The authors consider two subsets, $T_1$ and $T_2$, to remove this confounding effect. $T_1$ consists of samples where exactly one factor is different from the original image. For example, if the original image was a red 7, then $T_1$ would consist of the digit 7 of any other color except red. $T_2$ consists of the original samples. The mapping $M$ is learned between $T_1$ and $T_2$. For any given sample that differs in a factor, $M$ will map $X$ to a counterfactual instance $X'$, where $z_j$ is changed to the original value. These counterfactual instances can be augmented to the original data to break any correlation between $z_j$ (color) and $z_l$ (digit). We can repeat the process for each confounding edge from Figure 23b to generate a set of counterfactual samples that remove the spurious correlations. These augmented data points differ from the original in only one feature. In order to satisfy this condition, the authors introduce a contrastive learning objective with a pre-trained discriminator that takes $X$ and $X'$ and ensures that the representation $z_j$ is different between the two samples. Similarly, a discriminator is introduced to ensure that the representation $z_l$ is similar between the samples. That is, color can change, but the digit class is invariant. Finally, to evaluate the counterfactual generation under confounding, the authors consider a downstream classification task where they use the counterfactual augmented data to train a classifier to predict class labels. The empirical results show that CONIC outperforms empirical risk minimization, invariant risk minimization, and other baselines on test set accuracy.

## 4.2 Flow & Variational Inference-based

Although GANs can enable mechanism learning, they are notorious for their unstable training. Furthermore, we are often interested in estimating exact densities of distributions in a tractable way, which GANs are incapable of doing. Flow and VAE-based models are a flexible alternative for learning causal mechanisms. Further, they have several rigorous theoretical properties that make them stable and versatile. The following approaches leverage these models to enable tractable counterfactual inference.

### 4.2.1 Deep Structural Causal Models (DSCM) for Counterfactual Inference

**Problem Setting.** Pawlowski et al. (2020) propose a framework for building SCMs using normalizing flows and amortized variational inference to tractably estimate exogenous noise of the SCM for counterfactual inference. This work studies the setting where we have access to observed data (images) with known labels that may be causally related. The general setup is a structural causal model defined by the interactions between known labels and the observed data $x$. The observed data could be an image generated from high-level labels containing semantic information relevant to the image. For example, Figure 24c shows an MRI scan of a brain from the UK Biobank Imaging Study (Sudlow et al., 2015) and the causal structure of the associated data-generating process. In the causal graph, the brain volume (b) and ventricle volume (v) directly influence the generation of the image. However, other factors such as age (a) and sex (s) indirectly influence the image but directly affect the volume variables. Unlike CausalGAN (Kocaoglu et al., 2018), only a subset of labels directly influence the image.

**Learning Framework.** The authors propose using deep learning components to parameterize each mapping between variables in the SCM. The authors propose deep structural causal models (DSCM), an approach that is capable of performing counterfactual inference with fully specified causal models with no unobserved confounding. A central assumption in this work is that the mapping between observed data and its noise variable is bijective with a differentiable inverse. This is to ensure when given a set of observations, we can estimate the noise terms for counterfactual inference. Several works have emphasized the importance of reversibility for counterfactual inference models (Monteiro et al., 2023; Nasr-Esfahany et al., 2023). The authors leverage normalizing flows, an expressive class of generative models that offer flexibility in modeling

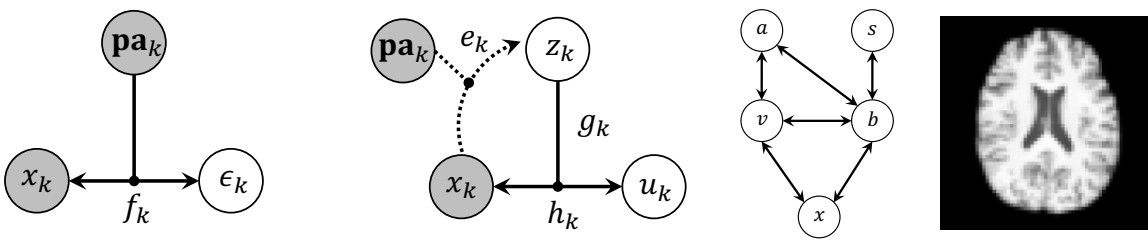

(a) Invertible explicit likelihood     (b) Amortized explicit likelihood     (c) Brain MRI example

Figure 24: DSCM modeling (Pawlowski et al., 2020)

complex probability distributions. The general idea of DSCM is to learn a mapping between a causal variable and its exogenous noise. This can be done by assuming a base distribution (typically a Gaussian) and using a flow to model the distribution over the causal variable. To model causal relations, the authors use conditional flows to additionally use parent variables to model the causal mechanisms. Thus, an invertible and explicit parameterization of the SCM, as shown in Figure 24a, would yield mechanisms of the form

$$x = f(\epsilon_x; \mathbf{pa}_x) \tag{74}$$

where the exogenous noise can be inferred as $\epsilon_x = f^{-1}(x; \mathbf{pa}_x)$.

When introducing complex nonlinear functions in high dimensional settings, inference of counterfactual queries can quickly become intractable. To overcome this, the authors propose using variational inference in addition to normalizing flows to model the noise term of the observed data, as shown in Figure 24b. This approach involves using amortized variational inference to learn a noise representation of the observed data. The amortized, explicit approach is a VAE-based method that separates a structural assignment $f$ into a low-level (invertible) component $h$ and a high-level (non-invertible) component $g$ such that

$$x_k = f_k(\epsilon_k; \mathbf{pa}_k) = h_k(u_k; g_k(z_k; \mathbf{pa}_k), \mathbf{pa}_k) \tag{75}$$

where $\mathbf{pa}_k$ denotes the parent variables of the observation $x_k$ (labels), the noise $\epsilon_k$ is decomposed into $(u_k, z_k)$ corresponding to low-level and high-level components of $(h_k, g_k)$, $g$ is a neural network that outputs mean and variance of $x_k$, and $h$ is a reparameterization.

**Counterfactual Inference.** During inference, we can compute answers to counterfactual questions. Following Pearl's counterfactual analysis, the three steps can be achieved as follows.

- **Abduction.** Each noise variable is assumed to affect only the respective observed variable. Thus, the noise $\epsilon_1, \ldots, \epsilon_K$ are conditionally independent given the observed data $x$. The following product factorization can approximate the noise posterior

$$p(\epsilon_k | x_k, \mathbf{pa}_k) = p(z_k | x_k, \mathbf{pa}_k) p(u_k | z_k, x_k, \mathbf{pa}_k) \approx q(z_k | e_k(x_k; \mathbf{pa}_k)) \delta_{h_k^{-1}(x_k; g_k(z_k; \mathbf{pa}_k), \mathbf{pa}_k)}(u_k) \tag{76}$$

  where $h^{-1}$ inverts the reparameterization to yield the noise term $u_k$.

- **Action.** To perform an intervention on the SCM $\mathcal{G}$, the structural assignment of the desired intervened variable $x_k$ is replaced by a constant, $x_k = \tilde{x}_k$, making it independent of the parents $\mathbf{pa}_k$ and exogenous noise $\epsilon_k$. This induces a counterfactual SCM $\tilde{\mathcal{G}} = (\tilde{\mathcal{S}}, p_{\mathcal{G}}(\epsilon|x))$.

- **Prediction.** To sample from the counterfactual SCM $\tilde{\mathcal{G}}$, the inferred noise can be plugged into the forward mechanism. The counterfactual distribution cannot be characterized explicitly in the explicit, amortized scenario. However, for each sample $s$, we can approximately sample from it via Monte Carlo as follows:

$$z_k^s \sim q(z_k | e_k(x_k; \mathbf{pa}_k)) \tag{77}$$

$$u_k^s = h^{-1}(x_k; g_k(z_k^s; \mathbf{pa}_k), \mathbf{pa}_k) \tag{78}$$

$$\tilde{x}_k^s = \tilde{h}_k(u_k^s; \tilde{g}_k(z_k^s; \tilde{\mathbf{pa}}_k), \tilde{\mathbf{pa}}_k) \tag{79}$$

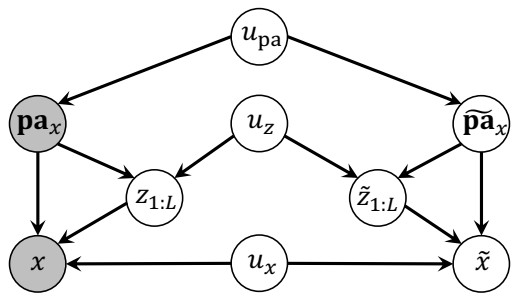
(a) Causal HVAE Latent Mediator Causal Graph

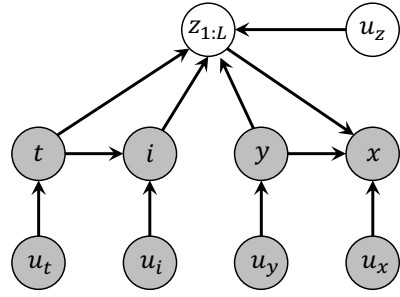
(b) MorphoMNIST Latent Mediator Graph

Figure 25: Causal Hierarchical VAE (a) mediator graph and (b) example (De Sousa Ribeiro et al., 2023)

**Example 6** *Given a high level parameterization $g_k(z_k; \boldsymbol{pa}_k) = (\mu(z_k, \boldsymbol{pa}_k), \sigma^2(z_k, \boldsymbol{pa}_k))$ and a low-level parameterization $h_k(u_k; (\mu, \sigma^2), \boldsymbol{pa}_k) = \mu + \sigma^2 \odot u_k$.*

$$u_k^s = (x_k - \mu(z_k^s; \boldsymbol{pa}_k))/\sigma(z_k^s; \boldsymbol{pa}_k), \qquad \tilde{x}_k^s = \mu(z_k^s; \boldsymbol{p\tilde{a}}_k) + \sigma(z_k^s; \boldsymbol{p\tilde{a}}_k) \odot u_k^s \tag{80}$$

### 4.2.2 High Fidelity Counterfactual Inference via Probabilistic Causal Models

**Problem Setting.** De Sousa Ribeiro et al. (2023) extend DSCM (Pawlowski et al., 2020) and propose a causal generative modeling framework for image counterfactual generation with an accurate estimation of interventional and counterfactual queries including direct and mediated causal effects. The problem setup consists of a high-dimensional image $x$ generated by a set of $n$ factors. The factors are a set of labels $\mathbf{pa}_x$ that may be causally related. This work aims to use VAE-based models to model the mechanisms between the labels and the image $x$. Then, one can infer exogenous noise, perform interventions on the set of labels to obtain $\mathbf{p\tilde{a}}$, and generate counterfactual instances using the decoder. The authors introduce a new framework that includes latent mediators between $\mathbf{pa}_x$ and $x$, which allow one to compute direct, indirect, and total effects of interventions from the learned mean and variance of $x$. The latent mediator model is shown in Figure 25a. Since directly estimating the exogenous noise can often be impractical and expensive, one can alternatively learn a latent mediator model. Consider the MorphoMNIST dataset (Castro et al., 2019) causal graph in Figure 25b. In this system, thickness, intensity, and the label all causally influence the generated image. Thus, inferring the latent mediator can enable one to compute indirect effects on the generated image.

The authors point out the fundamental notion of self-consistency of deterministic VAEs, which posits that the VAE encoder inverts the decoder. Thus, deterministic VAEs effectively function as normalizing flows. Reizinger et al. (2022) prove that VAEs actually satisfy self-consistency even in the near-deterministic regime. Pawlowski et al. (2020) model the SCM noise variable as a latent variable learned through variational inference. In this setup, the VAE is near-deterministic but proposes to abduct the noise of the observations through deterministic mapping, which leads to poor sample quality in DSCMs. To remedy this issue, the authors propose utilizing a more complex generative causal model for accurate noise abduction and high-fidelity counterfactual generation.

**Learning Framework.** The authors leverage hierarchical latent variable models (HLVM), which define priors over $L$ layers of hierarchical latent variables. Thus, HLVM can be viewed as a generalization of latent variable models where the latent variables have a hierarchical structure. In a hierarchical variational autoencoder (HVAE) (Sønderby et al., 2016), the latent variable $z_{1:L}$ is assumed to be hierarchical. That is, each latent $z_i$ is determined by latent $z_{i+1}$ in the $L$-layers. In HVAEs, the prior is learned from data instead of assumed to be a standard Gaussian. The authors propose a conditional HVAE model that decouples the prior from conditioning on the causal parents $\mathbf{pa}_x$. The generative model is defined as follows:

$$p(x, z_{1:L}|\mathbf{pa}_x) = p(x|z_{1:L}, \mathbf{pa}_x)p(z_L)\prod_{i=1}^{L-1} p(z_i|z_{>i}) \tag{81}$$

The conditional prior forces the prior $p(z_{1:L})$ to be independent of the causal parents $\mathbf{pa}_x$, while preserving the likelihood's dependence on $\mathbf{pa}_x$. In this formulation, $z$ is the exogenous noise of the observed data $x$.

The authors point out that the preceding formulation suffers from unidentifiability due to the unconditional prior. Based on the identifiability result from Khemakhem et al. (2020), the authors propose a conditional prior for the hierarchical latent variable $p(z_{1:L})$, where the auxiliary conditioning information is the set of parents $\mathbf{pa}_x$. The conditional prior takes the following form

$$p(z_{1:L}|\mathbf{pa}_x) = p(z_L|\mathbf{pa}_x) \prod_{i=1}^{L-1} p(z_i|z_{>i}, \mathbf{pa}_x) \tag{82}$$

where the prior is from an exponential family (Gaussian). In this formulation, the hierarchical latent variables depend on the causal parents and can be viewed as the latent *mediators* to be inferred.

The authors point out that conditioning the generative model on the counterfactual parents $\tilde{\mathbf{pa}}_x$ can often be ignored with a fixed abducted noise. To remedy this issue, the authors propose a constrained counterfactual training procedure optimized via the Lagrangian to maximize the mutual information between a counterfactual image $\tilde{x}$ and its counterfactual parents $\tilde{\mathbf{pa}}_x$.

One of the main contributions of this approach is that the mediator-based model enables computing direct, indirect, and total effects for generated images.

### 4.3 Diffusion-based

Diffusion probabilistic models have become the state-of-the-art generative model for high-quality data generation. In order to enable high-quality counterfactual generation, recent work has focused on modeling causal processes using diffusion models. The following approach elaborates on this idea and presents a method to perform tractable counterfactual generation using diffusion models and the theory of causality.

#### 4.3.1 Diffusion Causal Models for Counterfactual Estimation

**Problem Setting.** Sanchez & Tsaftaris (2022) propose a unifying framework between diffusion models and causal models that is capable of tractable counterfactual estimation. The authors take inspiration from the connection between diffusion models and score-based generative models (Song et al., 2021b) and propose a forward diffusion process formulated as the *weakening* of causal relations among variables of an SCM via a set of stochastic differential equations (SDEs).

$$dx^{(k)} = -\frac{1}{2}\beta_t x^{(k)} dt + \sqrt{\beta_t} dw, \qquad \forall k \in [1, K] \tag{83}$$

where $\beta_t$ is the variance parameter and $w$ denotes Brownian motion, $k$ is the index of the causal variable, and $K$ is the total number of causal variables. The distribution over causal variables induces the following causal Markov factorization

$$p(x_0^{(k)}) = \prod_{j=k}^{K} p(x^{(j)}|\mathbf{pa}^{(j)}) \tag{84}$$

where $x_0^{(k)}$ is observed causal variable $k$. The generative process is the solution to the following reverse-time SDE

$$dx^{(k)} = \left[ -\frac{1}{2}\beta_t + \beta_t \nabla_{x_t^{(k)}} \log p(x_t^{(k)}) \right] dt + \sqrt{\beta_t}\bar{w} \tag{85}$$

where $\bar{w}$ denotes Brownian motion (noise) in the reverse SDE. The reverse SDE represents the process of *strengthening* causal relations between variables.

**Learning Framework.** The authors propose Diff-SCM, a diffusion-based causal model to enable counterfactual estimation. The model is trained, as described above, with the reverse diffusion gradually strengthening the causal relationships between variables. In order to enable interventions and counterfactual estimation,

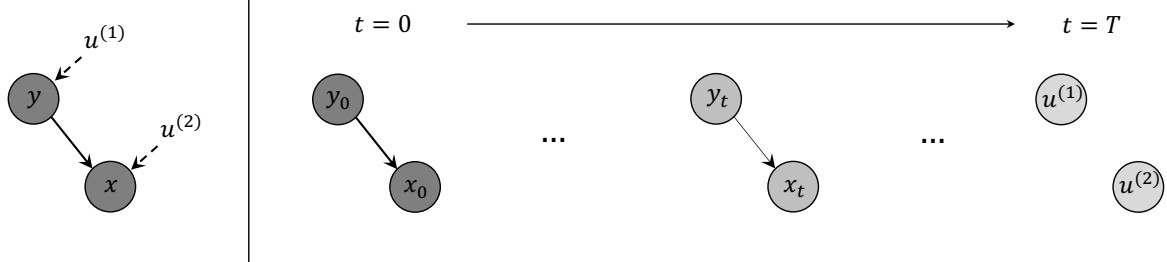

Figure 26: Diffusion Causal Models (Sanchez & Tsaftaris, 2022)

the authors propose an anti-causal predictor that is trained for each edge in the causal graph. To intervene on node $x^{(j)}$ and see its effects on node $x^{(k)}$, we need to estimate $p(x^{(k)}|\mathbf{do}(x^{(j)} = \tilde{x}^{(j)}))$. However, this distribution can be sensitive to changes in the cause distribution $p(x^{(k)})$. Thus, as proposed in Schölkopf et al. (2012), it would be easier to estimate $p(x^{(j)}|x^{(k)})$ and use Bayes' rule to estimate the original distribution. This objective aims to help guide the intervention toward the desired counterfactual class, similar to how classifier guidance is used to generate samples from a specified class label (Dhariwal & Nichol, 2021). The sampling process is guided by the anti-causal gradient $\nabla_{x_t^{(k)}} p(x^{(j)}|x^{(k)})$. The anti-causal predictor is trained separately from the diffusion model.

**Counterfactual Inference.** After training both the anti-causal predictor and diffusion model, the authors outline a three-step procedure to perform counterfactual inference using their framework. First, *abduction* is performed via forward diffusion using the SDE from Eq. (83) to infer the exogenous noise terms for the factual observation. Then, *action* is performed by intervening on the desired variable and removing edges between the intervened variable and its parents. Finally, *prediction* is performed via reverse diffusion controlled by the anti-causal gradients to obtain the counterfactual. First, we perform the intervention as follows:

$$\epsilon \leftarrow \epsilon_\theta(x_t^{(k)}, t) - s\sqrt{1-\alpha_t}\nabla_{x_t^{(k)}} \log p_\phi(x_{0,CF}^{(j)}|x_t^{(k)}) \tag{86}$$

where $\epsilon_\theta$ is the UNet of the diffusion model, $s$ is a hyperparameter that controls the scale of the anti-causal gradient, and $\alpha_t$ is the variance parameter, $\nabla_{x_t^{(k)}} \log p_\phi(x_{0,CF}^{(j)}|x_t^{(k)})$ is the anti-causal gradient of the predictor that estimates the counterfactual class of causal variable $j$, $x_{0,CF}^{(j)}$, given the causal variable $x_t^{(k)}$.

Then, we sample the counterfactual

$$x_{t-1}^{(k)} \leftarrow \sqrt{\alpha_{t-1}}\left(\frac{x_t^{(k)} - \sqrt{1-\alpha_t}\epsilon}{\sqrt{\alpha_t}}\right) + \sqrt{\alpha_{t-1}}\epsilon \tag{87}$$

The authors model the causal variables in the diffusion process as a causal relationship between class label $(x^{(0)} = y)$ and image sample $(x^{(1)} = x)$, as shown in Figure 26. For example, we can consider the relationship between a digit class and the corresponding image of the digit from the MNIST dataset. Thus, we have the causal graph $y \rightarrow x$.

**Example 7** *Given two variables x (image) and y (label), where $y \rightarrow x$, we can model the interventional distribution $p(x|\mathbf{do}(Y = y))$ using anti-causal gradients. Specifically, we have the following*

$$\nabla_x \log p(Y = y|x) \tag{88}$$

One drawback of Diff-SCM is that it is limited to the bivariate setting. It is not clear that the method would be able to scale to larger causal graphs. However, this approach could be a feasible solution for the specific use case of labels and images. There have been other lines of work utilizing the diffusion framework to study counterfactual explanations (Goyal et al., 2019; Augustin et al., 2022; Jeanneret et al., 2022), which is a related objective.

### 4.4 Other works

In this survey, we only cover counterfactual image generation methods that utilize the SCM formalism. However, there are other lines of work that utilize the potential outcomes framework (Rubin, 2005) or counterfactual inference methods that do not focus on controllable generation.

**Potential Outcomes.** Bose et al. (2022) propose causal probing of deep generative models (CAGE), a framework for controllable counterfactual generation in latent variable models based on causal reasoning. Given some data $x$ explained by attributes, perhaps in the form of annotated labels, CAGE proposes to estimate the average treatment effect of attributes considered to be effect variables. Unlike existing work such as CausalGAN and CausalVAE, which assume apriori that certain variables are causally related, CAGE aims to *infer* the causal relationships that are inherently captured by an arbitrary generative model capable of generalizing outside the training dataset. Given a pre-trained model, such as a VAE, CAGE studies the treatment effects of deep generative models. In the context of the potential outcomes (PO) framework (Rubin, 2005), the fundamental problem in causal inference is the inability to measure the effect of two different treatments simultaneously. To circumvent this problem, the authors leverage generative models to explicitly generate counterfactuals via some latent space manipulation strategy. The authors propose the notion of generative average treatment effect (GATE), which is an extension of the average treatment effect (ATE) from the potential outcomes framework to quantify the difference in outcomes for factual and counterfactual generations.

**Counterfactual Inference.** Rather than counterfactual generation, other related works focus on tractable counterfactual inference from non-image data (i.e., tabular data).

- **CAREFL.** Khemakhem et al. (2021) propose causal autoregressive flows (CAREFL) to model causal mechanisms in bivariate SCMs. In addition to approximating interventional distributions, the authors show that the invertible nature of flows allows for tractable counterfactual inference. The authors show that a generalized form of the additive noise model is also identifiable if the noise is from a Gaussian distribution. Based on this result, the authors develop a causal discovery method for bivariate causal models. The approach utilizes the tractable likelihood estimation capability of flows to check which causal direction produces the highest likelihood. However, CAREFL is limited to interventions on root nodes and does not evaluate non-root node interventions.

- **VACA.** Sánchez-Martin et al. (2022) propose variational causal graph encoders to generalize CAREFL to arbitrary causal graphs without parametric assumptions on the causal model. The general idea here is to use a variational graph autoencoder to learn the causal relationships among causal variables given a causal graph adjacency matrix $A$. First, a GNN encoder, with no hidden layers, maps the original data $x$ to a corresponding latent variable $z$ with the graph structure specified by $A$. Then, a GNN decoder, with the number of hidden layers at least the length of the longest shortest path between any two causal variables, models the structural mechanisms between causal variables via message passing. In this context, if $i \rightarrow j$, then $z_i$ must be used to reconstruct $x_j$. Interventions are performed by modifying the desired variable and the causal adjacency matrix. Further, counterfactual inference is performed by replacing the desired variable from the factual GNN encoder output with the variable from the intervened GNN encoder output and decoding.

- **DCM.** Chao et al. (2023) propose a diffusion-based causal model (DCM), a framework to learn causal mechanisms to enable sampling under interventions and counterfactual inference. Given $n$ causal variables $\{x_1, \ldots, x_n\}$ (in the form of observed data) and a causal graph, DCM proposes to learn a separate diffusion model for each causal variable. The authors formulate a forward and reverse diffusion process as a modification of the DDIM (Song et al., 2021a) objective. The forward diffusion process adds noise to the causal variable $x_i$ until at time step $t = T$, the variable $x_i^T$ is the SCM noise term for causal variable $x_i$. Then, in the reverse diffusion process, the parameterized noise predictor $\epsilon_\theta$ is additionally conditioned on causal variable $x_i$'s denoised parents $x_{\mathbf{pa}_i}$. That is, for the $\epsilon_\theta$ parameterization, we have $\epsilon_\theta(x_i^t, x_{\mathbf{pa}_i}, t)$. Counterfactual inference is facilitated by performing abduction via forward diffusion, action by intervening on a causal variable, and prediction by performing reverse diffusion (in topological order of the known causal graph).

### 4.5 Discussion

**Advantages.** Using traditional controllable generation approaches, such as GANs, to conditionally generate image samples has been an important direction of research for diverse data generation. However, such methods are simply conditional and do not encode any causal information. Thus, one cannot generate counterfactual images relative to factual observations. Controllable counterfactual generation (CCG) methods primarily work by modeling causal relationships between labels and high-dimensional observations. Thus, one could interpret this class of methods as a form of structural mechanism learning. So, we can ask the question, if the image labels were different, what would the image look like? The generative component of causal representation learning methods is also capable of counterfactual generation upon interventions on latent variables. However, in CCG, the main task is to generate high-fidelity image counterfactuals when given access to observed labels. CCG can be utilized to generate high-quality counterfactuals that can be used to augment datasets for more robust training. Further, the ability to generate novel scenarios never before observed through counterfactual inference can be crucial for applications in medicine, where patient data is scarce and often difficult to obtain due to privacy concerns. Finally, although the identifiability of counterfactuals can pose a challenge (Nasr-Esfahany & Kiciman, 2023), unlike representation learning, we do not have to worry about representation identifiability.

**Limitations.** The main disadvantage of causal controllable generation methods is the strong assumptions that are often required to be made, such as no unobserved confounding. Implicit generative models, such as CausalGAN, can often be quite difficult to optimize due to mode collapse. However, GAN-based methods can be useful for data augmentations to account for confounders (e.g., CONIC). Thus, explicit density estimating frameworks such as VAE-based, flow-based, and diffusion-based approaches are promising in controllable counterfactual generation tasks. A combination of VAE and flow-based methods, such as DSCM, can be used to tractably estimate distributions over causal variables and counterfactuals in a principled fashion using Pearl's counterfactual analysis (i.e., abduction, action, prediction). Diffusion models have shown superior generated image sample quality compared to GANs and thus can be used to generate high-quality counterfactuals. Lastly, most of the CCG methods are not scalable to the scenario where there are dozens of causal variables.

## 5 Evaluation Metrics

In this section, we list several important metrics to evaluate learned causal representations and generated counterfactuals. While some metrics are quite heuristic, others have strong ties to the theoretical properties of generative models.

### 5.1 Metrics for Causal Representation Learning

**Disentanglement, Completeness, and Informativeness (DCI).** The DCI metric (Eastwood & Williams, 2018) quantifies the degree to which ground-truth factors and learned latents are in one-to-one correspondence. The DCI disentanglement ($D$), completeness ($C$), and informativeness ($I$) scores are computed based on a feature importance matrix quantifying the degree to which each latent code is important for predicting each ground truth causal factor. The feature importance matrix is computed using gradient-boosted trees. The informativeness ($I$) score is the prediction error in the latent factors predicting the ground-truth generative factors. We train a generative model on a dataset with underlying true generative factors $y \in \mathbb{R}^K$ and obtain the latent representation $z \in \mathbb{R}^N$. Given $z$, we can learn a mapping $y = f(z)$ (often a regressor). We can train $K$ such predictors, one for each of the $K$ generative factors (i.e., $\forall i \in \{1, \ldots, K\}, y_i = f_i(z)$). Thus, we get a matrix of relative importances $R$, where $R_{ij}$ is the relative importance of $z_i$ in predicting $y_j$.

- **Disentanglement (D).** The degree to which a representation disentangles the underlying factors of variation with each variable capturing *at most one* generative factor. Let $P_{ij} = R_{ij} / \sum_{k=0}^{K-1} R_{ik}$ be the probability of $z_i$ being a strong predictor of $y_j$. Then, the disentanglement score is defined as

$$D_i = (1 + \sum_{k=0}^{K-1} P_{ik} \log_k P_{ik}) \tag{89}$$

If $z_i$ is a strong predictor for only a single generative factor, $D_i = 1$. If $z_i$ is equally important in predicting all generative factors, $D_i = 0$. Let $\rho_i = \sum_j R_{ij} / \sum_{ij} R_{ij}$ be the relative latent code importances. The total disentanglement score is simply a weighted average of the individual $D_i$

$$D = \sum_i \rho_i D_i \tag{90}$$

- **Completeness (C).** The degree to which a single latent code captures each underlying factor is defined as the completeness score

$$C_j = (1 + \sum_{n=0}^{N-1} \tilde{P}_{nj} \log_N \tilde{P}_{ij}) \tag{91}$$

where $\tilde{P_{nj}}$ is obtained in the same manner as $P_{ij}$. If a single latent code contributes to the prediction of $y_j$, the completeness score $C_i$ is 1. If all latent codes contribute equally to the prediction of $y_j$, the completeness score is 0.

- **Informativeness (I).** The amount of information that a representation captures about the underlying factors of variation. The informativeness score is quantified by the prediction error, averaged over the dataset, between the true factor and the predicted factor. Informativeness requires perturbations in a single latent factor to be systematic and informative.

Eastwood et al. (2023) extend the DCI framework and incorporate explicitness (E), which captures the ease-of-use of the representation, and size (S), which captures the amount of information that could be represented for evaluating representations. Further, they show that the DCI is a robust metric that has strong connections to linear and permutation-equivalent identifiability.

**Interventional Robustness Score (IRS).** It is often important that we achieve the robustness of groups of features in the latent variable with respect to interventions on groups of generative factors. To evaluate how changes in the generative factors affect the latent factors, one can compute the interventional robustness score (IRS) (Suter et al., 2019), which is similar to an $R^2$ value. IRS involves computing a post-interventional discrepancy that quantifies how much the learned latent factor shifts when intervening on extrinsic generative factors while keeping the generative factors we are interested in capturing at their predefined values. IRS has ties to the DCI metric and subsumes disentanglement as a special case. An IRS score of 1 implies perfect robustness and 0 implies no robustness.

**Mean Correlation Coefficient (MCC).** The mean correlation coefficient is obtained by first computing the Pearson correlation matrix $C \in \mathbb{R}^{n \times n}$ between the learned representation and ground-truth latent variables. Then, the MCC is computed as MCC $= \max_{\pi \in \text{permutations}} \frac{1}{n} \sum_{i=1}^{n} |C_{i,\pi(i)}|$. This metric has been used to assess permutation-identifiability (disentanglement) in Khemakhem et al. (2020) and Lachapelle et al. (2022).

**Mutual Information Gap (MIG).** Chen et al. (2018) propose the Mutual Information Gap, which is a classifier-free metric to measure the disentanglement of latent factors. The empirical mutual information between a latent variable $z_j$ and the ground-truth factor $y_k$ can be estimated using the joint distribution $q(z_j, y_k) = \sum_{i=1}^{n} p(y_k) p(i|y_k) q(z_j|i)$. We can then define mutual information as

$$I_i(z_j; y_k) = \mathbb{E}_{q(z_j, y_k)} \left[ \log \sum_{i \in \mathcal{X}} q(z_j|i) p(i|y_k) \right] + H(z_j) \tag{92}$$

A high mutual information $I_i$ implies that $z_j$ consists of a lot of information about $y_k$ and is maximal if there exists a bijection between $z_j$ and $y_k$. To cover discrete cases, we can additionally normalize $I_i$ by the entropy $H(y_k)$ (i.e., $I_i(z_j; y_k)/H(y_k)$). Disentanglement, according to Definition 3, requires each learned factor to be axis-aligned with the ground truth. Thus, the following Mutual Information Gap (MIG) metric enforces axis-alignment by measuring the difference between the top two latent variables with the highest

mutual information

$$\frac{1}{K}\sum_{i=1}^{K}\frac{1}{H(y_k)}\left(I_i(z_j^{(k)};y_k) - \max_{j\neq j^{(k)}} I_i(z_j;y_k)\right) \tag{93}$$

where $j^{(k)} = \arg\max_j I_i(z_j;y_k)$ and $K$ is the number of known factors. The MIG significantly penalizes unaligned variables, which are a sign of entanglement.

**Causal Disentanglement Score (CDS).** The DCI metric sometimes cannot distinguish between statistical correlation and causal link. Leeb et al. (2022) propose the causal disentanglement score (CDS), a metric based on decoding and reencoding latent samples through a *latent response function*, which helps to extract semantic information from the latent space without knowing how the information is encoded. The problem with evaluating disentanglement from only an encoder is that some latent variables may be spuriously correlated with the true factors without having a causal effect on the factors when generating new samples. The latent response function is defined as $h = f \circ g$, where $f$ is some encoder and $g$ is some decoder. First, we compute the following latent response matrix, which quantifies the degree to which an intervention on latent dimension $j$ causes a response in latent dimension $k$ and can be interpreted as a weighted causal graph

$$M_{jk}^2 = \frac{1}{2}\mathbb{E}_{z\sim p(Z);\tilde{z}_j\sim p(Z_j)}\left[|h_k(\Delta^{(z_j\leftarrow\tilde{z}_j)}(z)) - h_k(z)|^2\right] \tag{94}$$

where $\Delta^{(z_j\leftarrow\tilde{z}_j)}(z)$ refers to the interventional sample where the $j$th latent variable is sampled from the marginal of the aggregate posterior $\tilde{z}_j \sim q_\phi(Z_j)$.

If we have access to ground-truth variables $Y$, a variant of the latent response function, called the *conditioned response matrix*, can be used to quantify how much the learned factors correspond to the true ones. Let $Y_c$ denote a causal variable of interest and $Y_{-c}$ denote all other causal variables except $Y_c$. In order to condition the set of interventions on a specific factor $Y_c$, one can choose a subset of observations that are semantically the same except for the factor $Y_c$ (i.e., $Y_{-c}$ is the same for the observations). In other words, we sample latent variables where all causal labels are identical except $Y_c$. This guarantees that the sampled latent can be used as a valid intervention to invoke a response in the ground truth factor $Y_c$. The *conditioned response matrix* is defined as follows

$$M_{cj}^2 = \frac{1}{2}\mathbb{E}_{z\sim p(Z);\tilde{z}_j\sim p(Z_j|Y_{-c})p(Y_{-c})}\left[|h_k(\Delta^{(z_j\leftarrow\tilde{z}_j)}(z)) - h_k(z)|^2\right] \tag{95}$$

where $M_{cj}^2$ quantifies how much an intervention on each latent variable $z_j$ affects the ground-truth factors of variation $y_c$. The causal disentanglement score (CDS) is an extension of the DCI metric to take into account the learned generative process and is computed as follows:

$$CDS = \frac{\sum_j \max_c M_{cj}}{\sum_{cj} M_{cj}} \tag{96}$$

Similar to the DCI metric, CDS is a single-value score that quantifies the degree to which a learned representation can disentangle the factors of variation.

## 5.2 Metrics for Controllable Counterfactual Generation

**Counterfactual Latent Divergence (CLD).** Sanchez & Tsaftaris (2022) propose the counterfactual latent divergence (CLD) metric, which is used to measure the proximity and minimality of counterfactual samples. The CLD metric involves utilizing the Kullback-Leibler (KL) divergence to measure the divergence between latent distributions. The authors propose a relative distance metric to ensure that the minimality property of counterfactuals is satisfied (i.e., proximity to factual). Based on the relative distance metric, the counterfactual latent divergence (CLD) is the LogSumExpo of two probability measures, one enforcing minimality and one enforcing larger distances from factual.

That is, for a pair $(x_{CF}^{(1)}, x_F^{(1)})$, a VAE is used to project the samples to a low-dimensional latent representation $\mu_i, \sigma_i = E(x_i^{(1)})$. Then, we can measure the divergence as $\mathcal{D}(x_{CF}^{(1)}, x_F^{(1)}) = \mathcal{D}_{KL}(\mathcal{N}(\mu_i,\sigma_i)||\mathcal{N}(\mu_j,\sigma_j))$. The

authors define two sets $S_F$ and $S_{CF}$. The set $S_F$ consists of all relative distances comparing the factual $x_F^{(1)}$ to all possible samples with the same factual class. A low $P(S_F \geq \mathcal{D}(x_{CF}^{(1)}, x_F^{(1)}))$ enforces larger distances from the factual class. The set $S_{CF}$ consists of all relative distances comparing the factual $x_{CF}^{(1)}$ to all possible samples with the same counterfactual class. In this case, a low $P(S_{CF} \leq \mathcal{D}(x_{CF}^{(1)}, x_F^{(1)}))$ indicates that the counterfactuals satisfy minimality. The counterfactual latent divergence is simply defined as

$$\text{CLD} = \log(\exp P(\mathcal{S}_{CF} \leq \mathcal{D}(x_{CF}^{(1)}, x_F^{(1)})) + \exp P(\mathcal{S}_{CF} \geq \mathcal{D}(x_{CF}^{(1)}, x_F^{(1)}))) \tag{97}$$

**Axiomatic Evaluation.** Monteiro et al. (2023) propose evaluation metrics based on axiomatic principles of counterfactuals. Specifically, the soundness of counterfactuals can be formulated in terms of *effectiveness* (interventions properly affect the desired variables), *composition* (an identity intervention renders all variables unaffected), and *reversibility* (mapping between observations and counterfactuals is deterministic and invertible) (Pearl, 2009; Galles & Pearl, 1998). Let $x = g(\epsilon, \mathbf{pa})$ be the causal mechanism generating $x$ from its noise term $\epsilon$ and parents $\mathbf{pa}$. Then, for counterfactual analysis, we have $x^* = g(\text{abduction}(x, \mathbf{pa}), \mathbf{pa}^*)$, where $*$ represents the counterfactual. We can then generalize this counterfactual function and define a function $x^* = f(x, \mathbf{pa}, \mathbf{pa}^*)$ with the same parameters. Suppose we have an approximate counterfactual function $\hat{f}$ in place of the ideal counterfactual function $f$. The three axiomatic metrics to evaluate the soundness of counterfactuals are formally defined as follows

- **Composition.** We can measure how much the approximate model deviates from the true model by computing the distance between the original observation and the null transformation (trivial intervention) applied to the observation $m$ times. This is done to capture any sources of corruption from the approximate model. Formally, we have that the following should be low

$$\text{composition}^{(m)}(x, \mathbf{pa}) = d_X(x, \hat{f}^{(m)}(x, \mathbf{pa}, \mathbf{pa})) \tag{98}$$

  where $d_X$ is an arbitrary distance metric.

- **Reversibility.** In order for the mechanism to be invertible, the cycle-consistency property must hold. So, we can compute the distance between the original observation and the output after applying the transformation $m$ times. First, we apply $\hat{f}$ to perform abduction and map to a counterfactual $x^*$ via counterfactual parents $\mathbf{pa}^*$. Then, we apply $\hat{f}$ again to map back to $x$ by performing abduction with the counterfactual $x^*$ via the original parents $\mathbf{pa}$. That is, we have the composite function $p(x, \mathbf{pa}, \mathbf{pa}^*) = \hat{f}(\hat{f}(x, \mathbf{pa}, \mathbf{pa}^*), \mathbf{pa}^*, \mathbf{pa})$. Formally, reversibility is defined as follows

$$\text{reversibility}^{(m)}(x, \mathbf{pa}, \mathbf{pa}^*) = d_X(x, p^{(m)}(x, \mathbf{pa}, \mathbf{pa}^*)) \tag{99}$$

- **Effectiveness.** It is more difficult to measure effectiveness objectively. Thus, effectiveness can be measured individually for each parent of the observation by constructing an oracle function $\hat{\text{Pa}}_k(\cdot)$, implemented as a regressor or classifier, that returns the value of the parent $\text{pa}_k$ given the observed data. Formally, the effectiveness of each parent is defined as follows

$$\text{effectiveness}_k(x, \mathbf{pa}, \mathbf{pa}^*) = d_k(\hat{\text{Pa}}_k(\hat{f}_k(x, \text{pa}_k, \text{pa}_k^*)), pa_k^*) \tag{100}$$

  where $d_k(\cdot, \cdot)$ is a distance metric.

## 5.3 Discussion

In the domain of causal representation learning, especially causal disentanglement, the two most utilized metrics are DCI and MCC since they are theoretically grounded and have strong connections to identifiability theory (Eastwood et al., 2023). However, we have listed several other metrics (e.g., IRS, CDS, etc.) that are also promising in evaluating disentanglement and causal representations. We note that most controllable counterfactual generation methods rely on qualitative visual analysis to evaluate performance. However, several metrics such as CLD, Compositionality, Reversibility, and Effectiveness were recently proposed as principled metrics to evaluate counterfactuals. We believe these metrics and others that build from the foundational theory of counterfactual analysis should be used for evaluating generated counterfactuals.

| Dataset | Synthetic Image | | | | Real-World Image | | | | | Temporal Video | | | | | | | | | | | Biological & Medical | | |
|---|---|---|---|---|---|---|---|---|---|---|---|---|---|---|---|---|---|---|---|---|---|---|---|
| | Pendulum | Water Flow | CausalCircuit | Causal3DIdent | CelebA | MPI3D | MNIST | Morpho-MNIST | ImageNet | KiTTiMask | Mass-Spring | CMU MoCap | Cartpole | MoSeq | Causal Pinball | Interv. Pong | Ball-in-Boxes | Voronoi | CausalWorld | iTHOR | Perturb-Seq | UK Biobank | MIMIC-CXR |
| CausalVAE | ✓ | ✓ | | | ✓ | | | | | | | | | | | | | | | | | | |
| SCM-VAE | ✓ | ✓ | | | ✓ | | | | | | | | | | | | | | | | | | |
| DEAR | ✓ | | | | ✓ | | | | | | | | | | | | | | | | | | |
| ICM-VAE | ✓ | ✓ | ✓ | | | | | | | | | | | | | | | | | | | | |
| LEAP | | | | | | | | | | ✓ | ✓ | ✓ | | | | | | | | | | | |
| TDRL | | | | | | | | | | | | | ✓ | | | | | | | | | | |
| NCTRL | | | | | | | | | | | | | ✓ | ✓ | | | | | | | | | |
| Discr. VAE | | | | | | | | | | | | | | | | | | | | | ✓ | | |
| CITRIS | | | | ✓ | | | | | | | | | | | ✓ | ✓ | ✓ | ✓ | | | | | |
| iCITRIS | | | | ✓ | | | | | | | | | | | ✓ | ✓ | ✓ | ✓ | | | | | |
| BISCUIT | | | | | | | | | | | | | | | | | | ✓ | ✓ | ✓ | | | |
| SSL | | | | ✓ | | ✓ | | | | | | | | | | | | | | | | | |
| LCM | | | ✓ | ✓ | | | | | | | | | | | | | | | | | | | |
| CausalGAN | | | | | ✓ | | | | | | | | | | | | | | | | | | |
| CGN | | | | | | | ✓ | | ✓ | | | | | | | | | | | | | | |
| CONIC | | | | | ✓ | | ✓ | | | | | | | | | | | | | | | | |
| DSCM | | | | | | | | ✓ | | | | | | | | | | | | | | ✓ | |
| CausalHVAE | | | | | | | | ✓ | | | | | | | | | | | | | | ✓ | ✓ |
| Diff-SCM | | | | | | | ✓ | | ✓ | | | | | | | | | | | | | | |

Table 4: Datasets used in Causal Representation Learning and Controllable Counterfactual Generation

# 6 Datasets

In this section, we enumerate datasets that are often utilized in causal generative modeling tasks. We group them into four main categories: synthetic image datasets, real-world image datasets, temporal video datasets, and biological/medical datasets. We note that many CRL methods, especially those with more theoretical contributions, are also evaluated on a common non-image synthetic benchmark that involves generating random DAGs and transforming them via normalizing flows to high-dimensional observational data. The Erdős-Rényi random graph model (Erdős & Rényi, 1960) is primarily used to generate synthetic data for experiments in Unpaired Multi-domain CRL (Sturma et al., 2023), Score-based CRL (Varıcı et al., 2023b), Nonparametric CRL (von Kügelgen et al., 2023), Interventional CRL (Ahuja et al., 2023), Linear CRL (linear mixing) (Squires et al., 2023), and Linear CRL (general mixing) (Buchholz et al., 2023). We show examples of some datasets and their causal graphs in Figure 27 to illustrate causal modeling. Table 4 shows CRL and CCG methods and the visual datasets used for experiments.

## 6.1 Synthetic Image Datasets

**Pendulum.** The Pendulum dataset (Yang et al., 2021), consisting of roughly $7,000$ images with $96 \times 96$ resolution, describes the interaction between four causal variables: pendulum angle, light position, shadow length, and shadow position. The pendulum system shows a light source on the pendulum object that casts a shadow with a specific position and length. An example image and the causal structure of this system are shown in Figure 27a.

**Water Flow.** The Water Flow dataset (Yang et al., 2021), consisting of roughly $8,000$ images with $96 \times 96$ resolution, describes the interaction between four causal variables: ball size, water height, hole position, and water flow. The flow system shows a ball submerged in a glass of water, which has a hole in the side.

**CausalCircuit.** The CausalCircuit dataset (Brehmer et al., 2022) consists of $512 \times 512 \times 3$ resolution images generated by 4 ground-truth latent causal variables: robot arm position, red light intensity, green light intensity, and blue light intensity. The images show a robot arm interacting with a system of buttons and lights. The data is rendered using an open-source physics engine. An example image and the causal structure of this system are shown in Figure 27b.

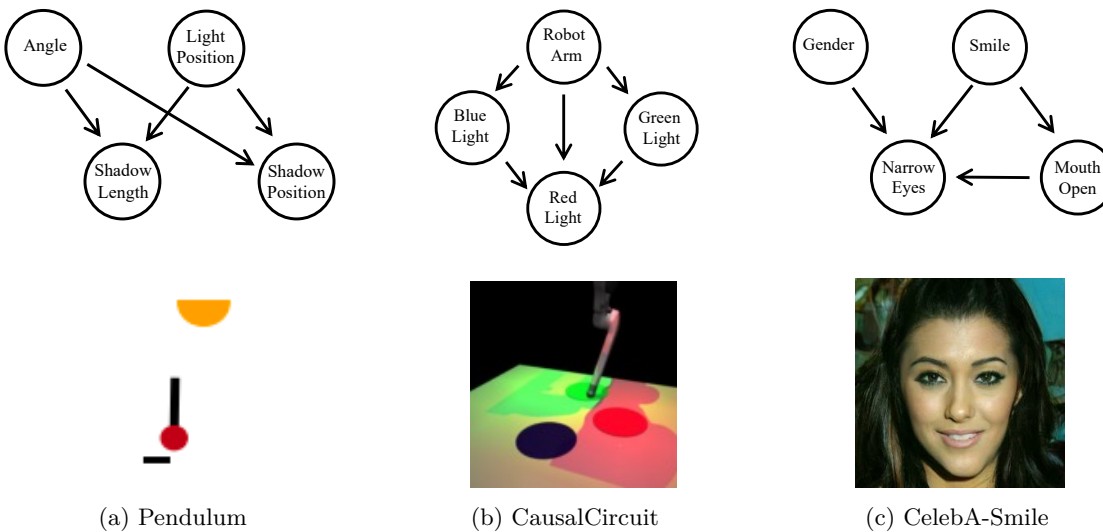

(a) Pendulum      (b) CausalCircuit      (c) CelebA-Smile

Figure 27: Causal Graphs of Selected Example Datasets used in Causal Representation Learning

**Causal3DIdent.** The 3DIdent dataset (Zimmermann et al., 2021) is a benchmark created for evaluating the identifiability of latent variable models. The dataset consists of $224 \times 224$ resolution images containing hallmarks of natural environments such as shadows, lighting conditions, and 3D objects. The Causal3DIdent dataset (von Kügelgen et al., 2021) consists of seven classes: Teapot, Hare, Dragon, Cow, Armadillo, Horse, and Head with a causal graph over the latent variables. The latent factors capture the spotlight position, object class, background color, spotlight color, object position, object rotation, and object color.

## 6.2 Real-World Image Datasets

**CelebA.** The CelebA dataset (Liu et al., 2015) is a large-scale face attributes dataset with more than 200K images of celebrity faces, each with 40 attribute annotations. We can impose a causal structure among any subset of the 40 attributes, such as CelebA-Smile, as shown in Figure 27c.

**MPI3D.** The MPI3D dataset (Gondal et al., 2019) consists of over one million images of physical 3D objects with seven factors of variation, such as object color, shape, size, and position.

**MNIST.** The MNIST dataset (Deng, 2012) consists of 70K, grayscale images of handwritten digits (zero to nine) with $28 \times 28$ resolution. Several methods utilize the MNIST dataset to generate counterfactual samples of different digits by modeling the causal mechanism between class labels and images.

**MorphoMNIST.** The MorphoMNIST dataset (Castro et al., 2019) is produced by applying morphological transformations on the original MNIST handwritten digit dataset. The digits can be described by measurable shape attributes such as stroke thickness, stroke length, width, height, and slant of digit. Pawlowski et al. (2020) impose a 3-variable SCM to generate the morphological transformations. The authors define stroke thickness as a cause of the brightness of each digit. That is, thicker digits are often brighter, whereas thinner digits are dimmer.

**ImageNet.** The ImageNet dataset (Deng et al., 2009) consists of over 14 million annotated images with 1000 object classes. Image Net has been utilized primarily in counterfactual generation methods such as those proposed by Sauer & Geiger (2021) and Sanchez & Tsaftaris (2022).

## 6.3 Temporal Video Datasets

**KiTTiMask.** The KiTTiMask video dataset (Klindt et al., 2021) includes pedestrian segmentation masks sampled from KiTTiMOS, an autonomous driving benchmark. In each time frame, the position and scale of pedestrian masks are measured.

**Mass-Spring system.** The Mass-Spring video dataset (Li et al., 2020) consists of ball movement rendered in color and invisible springs. There are ten causal relations, where six are set connected and four are disconnected.

**CMU MoCap.** The MoCap dataset (Yao et al., 2022b) consists of 3D point clouds of skeleton-based signals. It is a human motion capture dataset with various motion capture recordings (e.g., walking, jumping, etc.) performed by over 140 subjects. Each activity consists of several trials of recordings. The skeleton-based signal measurements have 62 observed variables that correspond to the locations of joints in the human body at each time step.

**Cartpole.** The Cartpole video dataset (Huang et al., 2022) consists of a cart and a vertical pendulum attached to the cart using a pivot joint. The cart is allowed to move laterally left or right. The ultimate goal is to prevent the pendulum from falling off the cart by applying force on the cart to move it, where the action space is defined by moving left or right. The original dataset also consists of two change factors: varying gravity/mass of the cart and the noise level of the image. These factors correspond to different domains. The two causal variables in this system are the cart position and the pole angle.

**MoSeq.** The MoSeq dataset (Wiltschko et al., 2015) is a video dataset of mice exploring an open field. The temporal observations are taken to be the first 10 principal components of the video data.

**Causal Pinball.** The Causal Pinball dataset (Lippe et al., 2023a) implements the idea of the popular game Pinball. In Pinball, the user controls two fixed paddles on the bottom of a field and tries to hit the ball such that it collides with various objects in the playground to score points. In Causal Pinball, there are 5 main components: the ball, left paddle position, right paddle position, bumpers, and the score.

**Interventional Pong.** The Interventional Pong dataset (Lippe et al., 2022b) models the dynamics of the game Pong. The underlying causal factors in this system are ball x-position, ball y-position, ball velocity, left paddle position, right paddle position, and the score.

**Ball-in-Boxes.** The Ball-in-Boxes dataset (Lippe et al., 2022b) is a synthetic dataset consisting of $32 \times 32$ images of a ball in 2D space that can be in one of two boxes. The ball can move within the box it is in but requires an intervention to move boxes. The dynamics of the system are governed by two causal variables: ball x-position and ball y-position, which both follow a truncated Gaussian distribution. The dataset is generated by using a sequence of 100K frames and sampling intervention targets from a Bernoulli distribution.

**Voronoi Benchmark.** The Voronoi Benchmark dataset (Lippe et al., 2022b) is a synthetic one created by generating one sequence with $150K$ timesteps, during which single-target interventions may have been performed. The interventions are sampled with $1/(n + 2)$ for each variable and with $2/(n + 2)$ for the observational regime, where $n$ is the number of patches that a $32 \times 32$ image is partitioned into (also representing the causal variables). The $n$ patches are transformed via a normalizing flow to obtain hues that are mapped to RGB space to render an image.

**CausalWorld.** The CausalWorld (Ahmed et al., 2021) is a benchmark for causal structure and representation learning in a robotic manipulation environment that is a simulation of an open-source robotic platform. The dataset consists of $128 \times 128$ resolution images from many tasks such as the robot assembling 3D complex shapes given a set of blocks, similar to how children reason in the world. The dataset includes underlying causal factors such as the robot and object masses, sizes, and colors.

**iTHOR.** The iTHOR dataset is one of the datasets from The House Of inteRactions (THOR) framework (Kolve et al., 2022). It consists of photo-realistic 3D door scenes, where AI agents can navigate the scenes and interact with objects to perform different tasks. iTHOR is the original set of scenes that includes 120 room-sized scenes, covering bedrooms, bathrooms, kitchens, and living rooms. The causal variables in this system are represented by objects in the scene and actions taken upon them.

### 6.4 Biological/Medical Datasets

**Perturb-Seq.** The Perturb-Seq (Norman et al., 2019) is a biological dataset created to study genetic interactions and phenotypic outcomes. Perturb-Seq is a high-dimensional dataset that measures the tran-

scriptional response of cells under different genetic perturbations. This dataset has been used to study interventional causal representation learning (Zhang et al., 2023b). Their approach utilizes both perturbed and unperturbed data samples to predict the effect of unseen combinations of interventions.

**UK Biobank.** The UK Biobank dataset is from the UK Biobank Imaging Study (Sudlow et al., 2015) that uses the FSL neuroimaging toolkit to preprocess 3D T1-weighed brain MRI scans. The dataset also provides age, biological sex, and precomputed brain and ventricle volumes for each brain MRI sample.

**MIMIC-CXR.** The MIMIC-CXR dataset (Johnson et al., 2019) consists of chest X-ray images with resolution $192 \times 192$ and several corresponding attributes such as sex, race, age, disease, etc.

## 7 Applications

In this section, we first focus on three important aspects of AI trustworthiness: fairness, privacy preservation, and robustness against distribution shift, in the context of causal generative modeling. The incorporation of these AI trustworthiness properties is commonly required in various applications. We then delve into the application of causal generative modeling in precision medicine due to the necessity of explainable generative models in the medical domain. Finally, we discuss an application of causal representation learning in biological sciences, where one often has access to an abundance of multi-domain data.

**Achieving Fairness in Causal Generative Models.** Achieving fairness is an imperative task in generative model learning. Zemel et al. (2013) first study fair representation learning and formulate fairness as an optimization problem of finding a good representation of the data that encodes the data as much as possible while simultaneously obfuscating the membership information. Louizos et al. (2015) extend the VAE objective with a Maximum Mean Discrepancy to ensure independence between the latent representation and the sensitive factors. Song et al. (2019a) further propose an objective for learning maximally expressive representations subject to fairness constraints and allow the user to control the fairness of the representations by specifying limits on unfairness. Locatello et al. (2019a) study the downstream usefulness of disentangled representations through the lens of fairness.

For data generation via GAN models, Xu et al. (2019) develop causal fairness-aware generative adversarial networks (CFGAN), which ensures various causal fairness criteria based on a given causal graph. CFGAN extends the FairGAN model (Xu et al., 2018) that achieves fair data generation in terms of disparate treatment and disparate impact. CFGAN adopts two generators whose structures are purposefully designed to reflect the structures of causal graphs and interventional graphs. The two generators can respectively simulate the underlying causal model that generates the real data and the causal model after the intervention. On the other hand, two discriminators are used for producing a close-to-real distribution and achieving various fairness criteria based on causal quantities simulated by generators.

**Preserving Privacy in Causal Generative Models.** Representation learning and generative modeling enable domain users to use representations or generated data for downstream tasks, which, to some extent, improves the privacy protection of the original sensitive data. However, releasing models or generated data could still be detrimental since it may unintentionally leak sensitive information to adversaries. Differential privacy (DP) (Dwork et al., 2006) is a popular mechanism for training machine learning models with bounded leakage about the presence of specific individuals in the training data. There have been a few works (Xie et al., 2018; Zhang et al., 2018; Chen et al., 2020; Torkzadehmahani et al., 2019) on training GANs in a differentially private way, and some recent works (Weggenmann et al., 2022; Yang et al., 2022) on training VAEs in a differentially private way. However, there has been no study on achieving differential privacy in causal generative models.

Most works on privacy-preserving GANs adopt the idea of tampering with the training process of the target model based on the differentially private stochastic gradient descent (DP-SGD) algorithm (Abadi et al., 2016). The DP-SGD algorithm involves clipping the gradients to bound sensitivity, adding calibrated noise to the sum of clipped gradients, and using the moment accountant technique to keep track of the privacy budget during backpropagation. The dp-GAN algorithm (Zhang et al., 2018) achieves DP by injecting random noise in the SGD training of the discriminator. Similarly, the DPGAN algorithm (Xie et al., 2018) adds noise to the gradient of the Wasserstein distance with respect to the training data. The GS-WGAN

algorithm (Chen et al., 2020) selectively applies sanitization to a necessary and sufficient subset of gradients, thus enabling the exploitation of the theoretical property of Wasserstein GANs. Torkzadehmahani et al. (2019) develop a differentially private conditional GAN (DP-CGAN) training framework that can generate not only synthetic data but also corresponding labels. It further clips the gradients of discriminator loss on real and fake data separately, allowing the user to better control the sensitivity of the model to real (sensitive) data.

Weggenmann et al. (2022) propose an end-to-end differentially private obfuscation mechanism called DP-VAE for generating private synthetic text data. The idea is to constrain the probabilistic encoder to facilitate differentially private latent sampling. Yang et al. (2022) propose a method that first maps the source data to the latent space through the VAE model to get the latent code, then performs a noise process satisfying metric privacy on the latent code, and finally uses the VAE model to reconstruct the synthetic data.

**Causal Representation Learning for Distribution Shift Modeling.** Distribution shift refers to changes in data distribution between the training and testing phases, leading to a decline in model performance. In real-world scenarios, the data encountered during the testing or deployment phase often comes from a slightly different distribution due to various factors, such as temporal changes, geographical changes, or different data collection processes. This discrepancy can lead to a model that performs well on the training data but poorly on new, unseen data. Causal representation learning addresses distribution shifts by focusing on the stable and consistent causal structures underlying the data invariant across distribution shifts. The ability of a model to generalize well across distribution shifts or domains is termed out-of-distribution (OOD) generalization. Sun et al. (2021) propose a variational-Bayesian-based framework that learns the underlying causal structure of the data and the source of distribution shifts from multiple training domains based on conditionally factorized priors about the latent factors. It specifies latent causal invariance models (LaCIM) that consider latent factors separable into latent semantic factors (e.g., the shape of an object) and variation factors (e.g., background, object detection), and a domain variable, as a selection mechanism to generate spurious correlations between the semantic and variation factors across distribution shifts. The latent semantic factors are direct causes of predictive variables (e.g., object class) and invariant to distribution shifts or domains. Lu et al. (2022) consider a causal generative model similar to LaCIM but assumes a more flexible conditionally non-factorized prior based on general exponential family distributions. The authors propose invariant Causal Representation Learning (iCaRL), which enables out-of-distribution (OOD) generalization based on learning from multiple training domains. Liu et al. (2021) propose a causal semantic generative model (CSG) for OOD generalization from a single training domain. CSG is similar to LaCIM but does not contain a domain variable to generate spurious correlations between the semantic and variation factors. The authors develop methods with a novel design in variational Bayes for efficient learning and OOD prediction.

**Causal Generative Modeling in Precision Medicine.** Clinical decision support systems are powered by machine learning models but tend to learn only associations between variables in the data (Sanchez et al., 2022b). Thus, healthcare systems are prone to spurious correlations. In a field as high stakes as medicine, it is paramount to use explainable and interpretable models to achieve robust knowledge about a system. Causal machine learning offers several potential directions to learn more robust models, including causal representation, discovery, and reasoning. When treating patients, healthcare professionals are often concerned with actionable insights from data. In the context of causal inference, this means treatment (interventional) decisions based on a comparison between a treated and a control group. Before reasoning about the effects of causal variables, we must be able to represent them. In the domain of medicine, the data can be highly complex, high-dimensional, and multimodal. Thus, it is critical to learn meaningful representations of the data before using it for decision-making.

Consider a healthcare application in Alzheimer's disease (AD) (Sanchez et al., 2022b). To understand the root causes of AD, many have recently proposed understanding it through a causal lens that takes into account several biomarkers and uses domain knowledge for deriving ground-truth causal relationships. For instance, we can consider causal variables such as chronological age, brain MRI image, and AD diagnosis. If we impose an SCM over these variables, guided by domain knowledge, we can properly model distribution shift scenarios. Further, we can learn semantically meaningful variables from MR images that may be causally related. Accurately modeling causal dependencies leads to the ability to directly generate counterfactual scenarios by sampling from a counterfactual SCM. One can also model domain/distribution shifts in the

data, such as population shifts. For example, if we train a classifier on data from younger individuals with a predisposition for AD, our model would likely perform poorly when we try to extrapolate to older populations. However, one can intervene on the age variable in causal models to generate new instances of older individuals. Due to this invariance property, causal models prove to be robust to distribution shifts. Recently, there has been work in using causal modeling in the medical imaging domain to synthetically generate counterfactuals using diffusion probabilistic models to analyze the health state of brain MRIs of patients and predict lesions from healthy and unhealthy patient data (Sanchez et al., 2022a).

**Causal Representation Learning from Biological and Genomics Data.** In biological sciences, one is often interested in discovering the underlying mechanisms governing biological systems. For example, if we want to study a specific disease and develop a suite of drugs to combat the disease, we would often want to know how proteins interact and the downstream effects of perturbations. Such a problem necessitates the use of causal modeling to explain complex biological processes. Unlike many other fields, biological sciences often utilize several tools that allow perturbational screens and multi-modal views (e.g., proteomics, transcriptomics, metabolomics, etc.) to obtain an abundance of interventional data or different views on the same system. In the language of causality, such a perturbation is equivalent to interventions. Some examples of interventions can be setting the expression of a certain gene to zero (i.e., gene knockouts), chemical perturbations (adding drugs), and mechanical perturbations. Given access to such data, learning what causal variables underlie a system along with their causal relationships can have significant implications for scientific discovery. There have been several recent works that explore identifiable causal representation learning (Squires et al., 2023; Zhang et al., 2023b) and intervention design in causal models (Zhang et al., 2023a) applied to sequencing data.

## 8 Challenges and Open Problems

**Causal Representations under Nonlinear General Form SCMs.** Although there have been efforts in causal discovery for general nonlinear SCMs, it has been limited to bivariate settings. It is still desirable to develop causal discovery algorithms and identifiability theory in this setting, especially in the purely observational setting. In general, causal structure identifiability has been explored for linear (and post-nonlinear) additive Gaussian noise SCMs. Despite the expressivity of this class of models, it can be quite a restricting assumption for universal approximability. In the context of causal representation learning, we seek to identify the causal factors and their rich causal structure. However, if we do not have access to interventional data, learning the causal structure with general-form nonlinear models is challenging. Even linear causal discovery methods (Zheng et al., 2018; Yu et al., 2019) have their limitations and can only recover a supergraph of the true causal graph. As we try to learn causal representations with purely observational data, it becomes crucial to expand the scope of latent SCMs that we can model. An interesting direction that ties the causal discovery task to representation learning is using the Jacobian of invertible mechanisms to derive causal structure (Reizinger et al., 2023). Further, it may sometimes be more feasible to assume that there exists an inherent causal structure that latent variable models can extract from the training data (Leeb et al., 2022).

**Causal Representation Learning from Weaker Forms of Supervision.** A natural question in causal representation learning is how we learn identifiable representations as we decrease direct supervision. von Kügelgen et al. (2023) show that causal representation learning is possible without any parametric assumptions. However, there are still open questions about how interventions on multi-node causal latent variables can generalize to arbitrary causal graphs. We note that there is still a lack of practical causal representation learning algorithms. Another challenge in learning causal representations is the so-called chicken-and-egg problem. That is, without causal variables, it is difficult to learn the causal structure and vice-versa. As highlighted in this survey, the data-generating paradigms fall into the observational, interventional, and counterfactual rungs of Pearl's Causal Hierarchy. Thus, it is important to understand the limitations of each paradigm. For instance, the observational data paradigm may be restricted to simpler causal models and often requires stronger supervision signals such as labels. Interventional data can sometimes be hard to obtain in many domains and identifiability can be easily violated unless we have interventions for every causal variable in the system. Ground-truth counterfactual data pairs are impossible to obtain in the real world since they represent hypothetical samples, but they may be used as an inductive bias through con-

trastive learning. Due to these constraints, it is important to assess what type of models are most feasible for real-world use cases to extract meaningful insights. A practical outlook on CRL from weak supervision is using multi-domain observational data, which can often be obtained due to access to large quantities of multi-modal data in the real world. Thus, results from multi-domain CRL methods (Sturma et al., 2023; Morioka & Hyvärinen, 2023; Ahuja et al., 2024) are especially of interest to bridge the gap between theory and practice in CRL. For example, there is plentiful multi-modal and multi-view data collected in the biological sciences from a variety of different tools. Some recent work formulates a general framework for causal representation learning from multiple distributions under a sparsity constraint on the recovered causal graph in a completely nonparametric setup and without the assumption of interventional data (Zhang et al., 2024). This is a promising direction indicating the applicability of CRL methods in real-world systems.

**Diffusion-based Causal Representation Learning.** With the emergence of diffusion probabilistic models as an effective generative modeling framework, even outperforming GANs in image sample quality, a natural research direction is to explore if we can learn causal representations using diffusion and score-based generative models. Preechakul et al. (2022) and Mittal et al. (2023) explore learning semantically meaningful representations in the diffusion framework. Mittal et al. (2023) propose learning a latent representation for each timestep in the diffusion process to capture richer representations. This area is quite underexplored and an interesting direction could study the disentanglement of causal representations under different data-generating assumptions and compare the robustness of representations with those learned from VAE-based models. Studying counterfactual image generation using diffusion-based causal representations is another interesting research direction.

**Practical Assumptions for Causal Generative Models in Real-world Applications.** Although there has been rapid progress in theoretical results for a range of weakly supervised CRL settings, many assumptions are violated in real-world applications. Relaxing such assumptions to make way for models that can show tangible results on real-world datasets such as medical images is still necessary. Further, causal representation learning has not been explored in the natural language domain, which may require different assumptions and algorithms altogether. Further, it is worth considering how to relax identifiability assumptions for downstream tasks such as domain generalization and transfer learning. Since CRL methods often depend on strong assumptions, such as causal sufficiency, they are prone to the effects of confounding in complex datasets. Thus, practical models should also focus on how to model confounders in a latent SCM to mitigate the effects of induced spurious correlations.

**Metrics to Evaluate Causal Representations w.r.t Counterfactual Quality.** Although correlated metrics such as DCI and Mean Correlation Coefficient (MCC) have been used to evaluate disentanglement of causal factors, there is still a lack of standardized metrics to evaluate the disentanglement and counterfactual generation quality of representations and models. Future work should systematically study evaluation procedures that rigorously test the causal model's ability to learn disentangled representations, generate quality counterfactual samples, and achieve robust downstream OOD performance. Metrics such as the Frechet Inception Distance (FID) (Heusel et al., 2017) could be modified to consider counterfactual quality. Since knowing the underlying data-generating process is not always feasible, exploring how to evaluate counterfactual quality without access to the underlying data-generating process is also an important direction. For instance, Monteiro et al. (2023) propose a general framework for evaluating image counterfactuals and derived distance metrics to compare counterfactual inference models.

**Lack of Benchmark Datasets.** The majority of the work in causal generative modeling, especially representation learning, considers synthetic data for which we know the underlying data-generating process. However, no standardized benchmark real-world dataset exists to study causal generative models. A push toward evaluating causal models on a set of benchmarks would go a long way to enable consistent and trackable progress in the field. However, it is important to note that obtaining image data from causal interventions can be difficult in practice. Interventions from randomized control trials may not always be feasible. Further, obtaining counterfactual data for evaluating generated counterfactuals is impossible. Thus, the community must work towards a unified framework for evaluating causal generative models. Furthermore, it could be worthwhile to explore more robust datasets that are prone to confounding and learning causal models that violate causal sufficiency. For example, Reddy et al. (2022b) construct CANDLE, a dataset for causal disentanglement that includes confounding between generative factors. Moreover, con-

structing datasets useful for a variety of downstream tasks is also an important direction. For instance, Liu et al. (2023) construct CausalTriplet, a real-world intervention-centric causal representation learning benchmark dataset. Each sample in the dataset consists of a pair of images captured before and after a high-level action. The dataset features high visual complexity, actionable counterfactual observations, and downstream interventional reasoning, which are lacking in existing datasets for causal representation learning.

**Causality in Large-scale Generative Models.** Recently, large-scale generative models, such as large language models (LLMs) (Zhao et al., 2023) and diffusion models (Yang et al., 2023), have shown impressive performative capability in language and vision tasks, respectively. Concepts are defined at a higher level of abstraction in large-scale generative models since they are trained on large and diverse data. Thus, perfect recoverability of concepts is impossible to achieve. Theoretical properties of the representation space of large-scale generative models would be an interesting direction to explore since even weak identifiability results could suggest the robustness and stability of such models to be used reliably for downstream tasks. The latent space of LLMs is certainly rich in structure that has not been explored well yet, especially through a causal lens. There has been some preliminary work using counterfactual language to understand model control in LLMs and pretrained text-to-image diffusion models by leveraging the linear representation space (Park et al., 2023; Wang et al., 2023). Another thrust of work studies mechanistic interpretability in large-scale models through causal abstractions (Wu et al., 2023). Looking forward, in addition to developing CRL algorithms from a set of assumptions, proving and empirically studying properties of the representation space of pretrained large-scale generative models through a causal interpretation can lead to a better understanding of model control and increase the applicability of these models in real-world systems.

**Open Source Software.** To help the transition from theory to the application of causal generative models in real-world systems and to share a framework for the research community, it is important to implement an open-source library for causal representation learning and causal generative models. The `disentanglement_lib`[6] package (Locatello et al., 2019b) is an open-source library developed for disentangled representation learning. The library consists of model training pipelines, pre-trained disentanglement models, data processing modules, evaluation metrics, and visualizations. The package contains standard unsupervised generative models, weakly supervised models, and applications to fairness and abstract reasoning tasks. A similar library developed for causal representation learning and counterfactual generation methods would greatly benefit the community and perhaps be a step towards application-oriented causal generative modeling research. The library should provide comprehensive metrics and algorithms that domain users and developers can incorporate into their real systems and/or design custom causal representation learning models via the provided APIs.

# 9 Conclusion

In this paper, we presented a technical survey on causal generative modeling focusing on identifiable causal representation learning and controllable counterfactual generation. We taxonomized causal representation learning methods based on the assumed data-generating process from Pearl's causal hierarchy and counterfactual generation methods based on the type of generative model. We targeted our survey for both beginners to the field of causal generative modeling and experts looking for a review and provided an intuitive introduction, with respect to theory and methodology, to fundamental approaches in the field. In addition to methods, We covered common datasets and metrics used in causal representation learning and controllable counterfactual generation. We outlined promising applications of causality-based generative models in trustworthy AI, precision medicine, and biology. Finally, we discussed several current challenges and potential directions for future work.

# Acknowledgements

This work is supported in part by National Science Foundation under awards 1910284, 1946391 and 2147375, the National Institute of General Medical Sciences of National Institutes of Health under award P20GM139768, and the Arkansas Integrative Metabolic Research Center at University of Arkansas.

---

[6]https://github.com/google-research/disentanglement_lib

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
