# OpenReview forum: "From Identifiable Causal Representations to Controllable Counterfactual Generation: A Survey on Causal Generative Modeling"
_TMLR — Accepted by TMLR_

### Review · Reviewer_pB3s · 2023-12-25

**Summary Of Contributions:**

The paper provides a thorough review of recent generative models that either focus on causal representation learning (CRL) or controllable counterfactual generation (CCG). The review is preceded by an extended introduction that 1- defines central notions regarding statistical causality, 2- familiarizes the reader with various architectures in generative modeling, and 3- discusses the notions of identifiability and disentanglement, and how these relate to causal representation learning. The reviewed methods are examined and tabulated with respect to the architectures they rely on, main assumptions on the actual data generating distribution as well as observed variables, and the identifiability results provided in the case of CRL. The paper finishes off with a summary of commonly used datasets, important applications of the reviewed methods, and open research questions.

**Audience:**

Yes

**Broader Impact Concerns:**

I believe the contents of this paper does not require a Broader Impact Statement.

**Claims And Evidence:**

Yes

**Requested Changes:**

Here, in support of the weaknesses subsection above, I present expanded comments, as well as mentioning some minor points. I consider it important, yet not critical, for the issues raised above and here to be addressed for the final paper:
-  Identifiability and Disentanglement w.r.t. Causality
	- In Section 2.6.2, how these methods lead to identifiability can be discussed at least in the form of an example. This does not need to be exhaustive for all three methods presented therein, but still an example or high-level description should be present for completeness.
	- Section 3.1.4 can be expanded to more thoroughly describe the increase in complexity of the problem of identifiability as we transition to causal representation learning, with the assumption of independent factors representing a trivial SCM, and what kind of additional information and/or assumptions this might require.
- Comparison of CRL and CCG with alternatives:
	- In CRL, this would take the form of comparing it with representation learning methods in general, wherein the advantages could include the authors describing in what ways CRL-produced representations are likely to be more robust to distribution shifts, and the disadvantages could include the additional requirements or assumptions on the data at hand (e.g. observed auxiliary labels, domain annotation, fixed SCM class), as well as increased computational costs whenever applicable. Although these dis/advantages are presented throughout the paper within specific methods, a more high level comparison and commentary would be beneficial for the reader.
- Minor points:
	- The Example 1 on Pg. 10 is too abriged to be easily understandable.
	- Definition 4: "domain of parameters": Please be explicit that members of $\Theta$ parametrize the mixing functions and densities.
	- "Reduced form SCM" is not defined.
	- Given the level of detail present in the paper, at least an informal definition of "distribution shift" should be provided.
	- Section 3.1.3: The beginning of this subsection is repetitive.
	- Definition 1 is referenced in Section 3.1.3, which does not exist. This must be a mix-up regarding the existence of Principles 1 and 2, as Definitions start with Definition 3.
	- The following sentence is unclear and would benefit from expansion: "However, such an approach does not isolate the learned factors and is effectively semantically meaningless; hence, the impossibility implication for arbitrary unsupervised disentanglement (Locatello et al. (2019b))."
	- The following sentence is unclear: "That is, the multi-domain setup "completes the picture" of the causal model in some sense since relations can arise from different domains, as shown in Figure 11."
	- Please be explicit about the notion of un/paired data in places where this is used as qualifier.
	- MorphMNIST is referred to without any context before its introduction in datasets section.
	- The same model is mentioned under two different names $\beta$-VAE and BetaVAE

**Strengths And Weaknesses:**

### Strengths
- Overall this is an informative, thorough, and well-written review.
- The technical preamble is succinct yet informative, and informs the reader regarding the nuances of important concepts such as identifiability and disentanglement.
- The reviewed methods are presented accessibly and within a logical categorization, the reader is usually directed to a (recreated) graphical model representing the modeling assumptions of the method in question, and the interrelations between methods are highlighted frequently. The tabulation of the presented work in Tables 2 and 3 is additionally helpful.
- The inclusion of commonly used datasets, important applications, and open questions increase the value of the paper as a reference for the topics in question.
- The paper is very well and carefully written, and is very easy to follow overall.

### Weaknesses
- As appropriate as the inclusion of an extensive introductory discussion of the issues of identifiability and disentanglement is, I think some parts of this discussion, as well as how they transfer to the context of CRL are underdeveloped, given the level of detail present throughout the paper. See below for details.
- Although both CRL and CCG methods are diligently categorized within each other, the paper could include an additional (sub)section that compares these class of methods to their alternatives more broadly, both positively and negatively. See below for details.
- Why the authors focus on a subset of research is not always sufficiently discussed, and the reader is not always directed to other surveys (if exists), for research that falls out of scope of the paper. This is the case in the authors limiting their survey to CRL and CCG in Section 1 and limiting CRL to VAE-based models in Section 3.1.4. The authors also declare non-medical applications out-of-scope without any references for the interested reader.
- The metrics section lack a consistent evaluation or comparison of metrics with each other, as well as how frequently they are used in the literature or in the papers reviewed.

---

> ### Author Response · Authors · 2024-02-07
> **Response to Reviewer pB3s**
>
> We thank the reviewer for the detailed feedback to improve our survey. We have incorporated your feedback and requested changes with $\textbf{color code orange}$ in the revised version.
>
> We answer your concerns in the order listed in the Weaknesses and Requested Changes sections
>
> 1. a.) The various strategies have often been used in traditional nonlinear ICA and disentangled representation learning to achieve identifiability of latent factors of variation. We have added a brief discussion for each strategy to discuss the high-level intuition behind how they can be used to achieve identifiability $\textbf{(Section 2.6.2)}$.
> b.) We have added a paragraph explaining the increase in complexity from standard representation learning to causal representation learning including the challenge of identifiability of causal variables, identifying latent causal structure, and strong data-generating assumptions in $\textbf{Section 3.1.4.}$
>
> 2. We have now added a section after the CRL $\textbf{(Section 3.8)}$ and CCG $\textbf{(Section 4.5)}$ sections, respectively, discussing their advantages and limitations compared to standard non-causal approaches. We also discussed the feasibility of varying assumptions on the data-generating process in real-world settings.
>
> 3. We emphasize in our introduction that our focus in this survey is causal representation learning and counterfactual generation. We do not cover causal discovery since it is a parallel effort that very sparsely utilizes generative models. However, we do refer readers to surveys on causal discovery whenever appropriate (Vowels et al, 2021; Squires & Uhler, 2022). In our previous version, we focused on medical applications (in addition to fairness, privacy, and OOD generalization) due to explainability being a critical component in medicine. In the revised version, we have additionally included applications of CRL in the biological and genomics fields due to the abundance of interventional and multi-modal data often available $\textbf{(Section 7)}$.
>
> 4. We note that some of the metrics presented are not used by any of the methods surveyed. We have added a brief section discussing the evaluation metrics. Specifically, we point out that metrics such as DCI and MCC have been used extensively in the literature. However, other recent metrics, such as CDS, are promising metrics that are potentially more robust at evaluation. We also note that most controllable counterfactual generation methods are evaluated qualitatively by visual inspection and discuss two recent metrics that are theoretically grounded in counterfactual analysis as promising candidates for future methods to use for evaluating counterfactuals. We now include this discussion in $\textbf{Section 5.3}$.
>
>
> 5. We address the minor points as follows
>
>    * We have extended Example 1 to explain the intuition behind why the Darmois construction indeed yields unidentifiability of nonlinear ICA with a simple 2-variable example.
>
>    * We have now explicitly stated that the elements of the parameter space parameterize the mixing functions and densities in $\textbf{Definition 4}$.
>
>    * We have added a footnote to define the notion of a reduced form SCM (pg. 16).
>
>    * We note that distribution shift is defined in the application section that discusses OOD generalization $\textbf{(Section 7)}$.
>
>    * We have reworded some of $\textbf{Section 3.1.3}$ to remove redundancies.
>
>     * Thank you for pointing this out. Yes, we intended to refer to Principle 1 (ICM principle). We have fixed this in the revised version.
>
>     * We have reworded this sentence to emphasize that we cannot manipulate the factors in a controllable fashion without disentangling them.
>
>     * We have elaborated a bit more in this section to make this concept clear. The idea is that different domains may consist of different latent variables. Thus, the joint representation when we use all the domains would give us the picture of the complete (or perhaps more accurate) causal model.
>
>     * We have defined unpaired/paired when describing methods that have them as an assumption. We have also included a description of what paired/unpaired data means in the interventional regime $\textbf{(Section 3.4)}$.
>
>     * We have cited the MorphoMNIST dataset and explained the features in the DSCM and CausalHVAE sections.
>
>     * We have fixed this inconsistency in the revised version.
>
>
> (Vowels et al, 2021) Matthew J Vowels, Necati Cihan Camgoz, and Richard Bowden. D’ya like dags? a survey on structure
> learning and causal discovery. arXiv preprint arXiv:2103.02582, 2021.
>
> (Squires \& Uhler, 2022) Chandler Squires and Caroline Uhler. Causal structure learning: A combinatorial perspective. Foundations
> of Computational Mathematics, pp. 1–35, 2022.

---

### Review · Reviewer_Q6jz · 2024-01-12

**Summary Of Contributions:**

The work presents a survey on causal representations and controllable counterfactual generation using generative models.
It includes a comprehensive review of fundamental theory, methodology and (evaluation) metrics, and has a brief discussion on datasets, applications and open problems.
There is no new knowledge claimed nor presented in this work.

**Audience:**

Yes

**Claims And Evidence:**

Yes

**Requested Changes:**

I have the following requested changes:

Critical
1. The survey is very much based on Pearl’s SCMs and Pearl's causal hierarchy. While they are widely adapted in causality research, they are not the only tools for causal inference and ways to categorize related concepts. I think this should be reflected in the title somehow to set the right expectation for readers. Just to make it clear, I'm questioning this choice and not asking the author(s) to change the way the survey is done, I'm suggesting to reflect this in the title somehow.
2. Citation format is wrong for the entire paper. Please use \citet and \citep properly, e.g. `\citet{} proposes XXX` and `such as XXX \citep{}`.
3. While not being actively maintained, the [`disentanglement_lib`](https://github.com/google-research/disentanglement_lib) should at least be mentioned in the "Open Source Software" as it has been used in many follow-up work.

Non-critical but would be good to do to improve the draft
1. Having an in-depth comparison between methods is highly encouraged. For example, the current table 3 is not that useful, especially the "Key Features" simply gives some keywords. Perhaps the authors can find a few important properties to form a set of columns and have each row tick some of the properties the corresponding method satisfies.
2. The diagrams seems to be distorted in the PDF at certain zoom levels. They can be improved by using TIkZ or anything software produces vector graphics.
3. Another useful table to have is one where the rows are methods and columns are datasets (or types of datasets). We can use this table to provide a view of which datasets have already be studied on which methods. This might be a good starting point to catch up the missing common benchmark datasets issue.

**Strengths And Weaknesses:**

Strengths
- The previous works that the submission surveyed is comprehensive.
- Compared to previous surveys, the submission also includes a good summary of identifiability theory and challenging open problems.

Weaknesses
- The summary and discussion of previous methods are relatively shallow. Mostly the problem setting is given and the method is described. However, there is no much in-depth discussion to compare the pros/cons, when certain methods should be preferred than others, etc.
- The contents for datasets, applications and open problems are relatively small, compared to the other like fundamental theory, methodology and metrics.

---

> ### Author Response · Authors · 2024-02-07
> **Response to Reviewer Q6jz**
>
> We thank the reviewer for the detailed feedback in the weakness section to improve our survey. We have incorporated your feedback and requested changes with $\textbf{color code cyan}$ in the revised version.
>
> We answer your concerns in the order listed in the Requested Changes section
>
> Critical
> 1. Thank you for your feedback regarding the title of our survey. We note that most work in CRL and CCG utilizes the SCM framework since it is a data-generating model. However, there has been some work inspired by the potential outcomes (PO) framework. In Section 4.4, we cover a method that is inspired by the PO framework but still utilizes ideas from the SCM (CAGE). To clarify for readers, we have added a paragraph to the introduction explaining the reason for focusing on the SCM framework.
>
> 2. We have fixed the citation format to reflect narrative and parenthetical citations properly using \citet{} and \citep{}. Thank you for pointing this out.
>
> 3. We have now added the disentanglement_lib as a reference in the “Open Source Software” section and emphasize the lack of such a library for causal representation learning.
>
> Non-critical
>
> 4. We have reformatted the table for controllable counterfactual generation methods $\textbf{(Table 3)}$ according to the reviewer’s suggestion. We have added the headers Causal Mechanism, Counterfactual Inference, and Key Assumptions. We hope this serves as a more useful table to compare the different methods. We have also added $\textbf{Section 3.8}$ and $\textbf{Section 4.5}$ to discuss in more detail the pros and cons of CRL and CCG, respectively.
>
> 5. Thank you for your feedback. We have modified the figure in the Datasets section with the causal graphs (Figure 30) to be clearer by increasing the font size.
>
> 6. We thank the reviewer for suggesting the dataset-specific table for CRL and CCG methods. We have now included a table with methods as rows and datasets as columns to illustrate which datasets have been studied by which method $\textbf{(Table 4)}$. We have also added more datasets for completeness as requested by Reviewer A2sk in $\textbf{Section 6}$.

---

### Review · Reviewer_A2sk · 2024-01-23

**Summary Of Contributions:**

The paper provides a thorough and insightful technical survey on causal generative modeling, specifically focusing on causal representation learning and controllable counterfactual generation methods. The contributions are as follows:
1. The author categorizes causal generative modeling into identifiable causal representation learning and controllable counterfactual generation tasks. A detailed classification of causal representation learning methods based on Pearl's Causal Hierarchy is provided, along with a breakdown of controllable generation approaches by the type of generative model used.
2. This paper discusses evaluation metrics, benchmark datasets, and results for causal generative models.

**Audience:**

Yes

**Broader Impact Concerns:**

Not applicable.

**Claims And Evidence:**

Yes

**Requested Changes:**

See the weaknesses above.

**Strengths And Weaknesses:**

Strengths:

1. The author effectively categorizes causal generative modeling, offering clarity on identifiable causal representation learning and controllable counterfactual generation.
2. The paper demonstrates a strong understanding of causal representation learning by organizing methods and providing a comprehensive discussion on evaluation metrics, datasets, and results.

Weaknesses:

1. The authors did not list the existing datasets in full. Many of the referenced papers include some datasets. Including these datasets in the paper would make the survey more comprehensive.
2. Causal generative modeling is also employed to address issues related to learning causal relationships among observed variables, why do the authors only focus on the causal representation learning and controllable counterfactual generation? Moreover, it would be better to provide the relationship between causal representation learning and controllable counterfactual generation.
3. There are also some works on using generative models to address causal representation learning that have not been included in the paper, such as LEAP[1], TDRL[2], and so on.

[1] Yao, Weiran, et al. "Learning Temporally Causal Latent Processes from General Temporal Data." International Conference on Learning Representations. 2021.
[2] Yao, Weiran, Guangyi Chen, and Kun Zhang. "Temporally disentangled representation learning." Advances in Neural Information Processing Systems 35 (2022): 26492-26503.

4. Although a symbol table is provided in the paper, there is some inconsistency in the use of symbols. For instance, in Figure 7, "A" is used to represent a variable, while in Equation 34, "A" is used to denote the causal adjacency matrix. This could lead to confusion in understanding.

---

> ### Author Response · Authors · 2024-02-07
> **Response to Reviewer A2sk**
>
> We thank the reviewer for the detailed feedback in the weakness section to improve our survey. We have incorporated your feedback and requested changes with $\textbf{color code red}$ in the revised version.
>
>  We answer your concerns in the order listed in the Weaknesses section:
>
> 1. We have now included more datasets from the papers for completeness and have included a table showing which methods experiment on each dataset $\textbf{(Table 4)}$. We have also split the datasets into synthetic image, real-world image, temporal video, and biological/medical categories $\textbf{(Section 6)}$. We also mention a synthetic non-visual dataset generated from sampled random graphs (Erdos-Renyi random graph model) often used for CRL experiments.
>
> 2. In this survey, we focus our attention on causal representation learning and counterfactual generation due to recent research focusing on modeling underlying data-generating processes of high dimensional data. We do not focus on causal structure learning as it is a parallel effort. Other surveys thoroughly cover the landscape of causal discovery, including those methods that utilize generative models (Vowels et al, 2021). Regarding the relationship between causal representation learning and controllable counterfactual generation, we have added a paragraph describing their similarities and differences in the introduction.
>
> 3. We thank the reviewer for pointing out the temporal disentanglement works in the observational setting. We have now included these in the observational CRL section of our survey $\textbf{(Section 3.3)}$. Further, we have also included some more recent work in the observational and interventional paradigms of causal representation learning for completeness (please see global comment for full list). $\textbf{Table 1}$ has been updated accordingly.
>
> 4. We have revised our manuscript to ensure consistency of notation as much as possible.
>
>
> (Vowels et al, 2021) Matthew J Vowels, Necati Cihan Camgoz, and Richard Bowden. D’ya like dags? a survey on structure learning and causal discovery. arXiv preprint arXiv:2103.02582, 2021.

---

### Author Response · Authors · 2024-02-07
**Global Response Summarizing Main Changes**

We thank all reviewers for their incredibly useful feedback and suggestions. We appreciate the reviewers acknowledging that our work is informative, thorough, and well written (Reviewer pB3s), comprehensive with a good summary of topics (Reviewer Q6jz and A2sk), and offers clarity and insightfulness (Reviewer A2sk).

We have updated the submission with a revised version with general changes reflected by $\textbf{magenta color coding}$ and requested changes from reviewers reflected by different color coding Reviewer pB3s -> orange, Reviewer Q6jz -> cyan, Reviewer A2sk -> red. Since submitting our initial version, there were a few more recently published CRL works that we deemed as important enough to include in our survey for completeness. Thus, we have added around 7 more papers, including temporal methods pointed out by Reviewer A2sk.

The following are the main revisions we have made incorporating all reviewers’ comments.

1. Added new methods and updated Figure 1 accordingly

    * Grouped Causal Representation Learning (Morioka \& Hyvarinen, 2023) - $\textbf{Section 3.2.6}$

    * Multiview CRL (Yao et al, 2023) - $\textbf{Section 3.2.7}$

    * Latent Temporal Causal Process (Yao et al, 2022b) - $\textbf{Section 3.3.1}$

    * Temporally Disentangled Representation Learning (Yao et al, 2022a) - $\textbf{Section 3.3.2}$

    * Temporally Disentangled Representation Learning under Unknown Nonstationarity (Song et al, 2023) - $\textbf{Section 3.3.3}$

    * Causal Disentanglement under Soft Interventions (Zhang et al, 2023) - $\textbf{Section 3.4.4}$

    * Score-based CRL (Varici et al, 2023) - $\textbf{Section 3.4.6}$

2. Added a "Discussion" section after CRL $\textbf{(Section 3.8)}$ and CCG $\textbf{(Section 4.5)}$ sections, respectively, detailing advantages & limitations

3. Added a more comprehensive list of datasets and categorized them into synthetic image datasets, real-world image datasets, temporal video datasets, and biological/medical datasets $\textbf{(Section 6)}$

4. Added a new table with methods as rows and datasets as columns $\textbf{(Table 4)}$

5. Added high-level intuition in $\textbf{Section 2.6.2}$ regarding how the strategies lead to identifiability

6. Explained the increase in complexity from standard RL to CRL $\textbf{(Section 3.1.4)}$

7. Explained in a bit more detail the connection between CRL and CCG $\textbf{(Introduction)}$

8. Added a section discussing the different metrics and their usage in the literature $\textbf{(Section 5.3)}$

9. Reformatted and updated $\textbf{Table 2}$ and split into multiple columns including assumptions on mixing function, causal model, and other key assumptions

10. Fixed citation format to use \citet{} and \citep{} as appropriate

11. Described disentangled_lib as an Open Source disentanglement package and the lack of such a library for CRL and causal generative modeling in general $\textbf{(Section 8)}$

12. Added an application of CRL in biology and genomics $\textbf{(Section 7)}$

13. Updated $\textbf{Table 3}$ to include more columns describing features of CCG methods

References

(Morioka \& Hyvarinen, 2023) Hiroshi Morioka and Aapo Hyvärinen. Causal representation learning made identifiable by grouping of
observational variables, 2023.

(Yao et al, 2023) Dingling Yao, Danru Xu, Sébastien Lachapelle, Sara Magliacane, Perouz Taslakian, Georg Martius, Julius
von Kügelgen, and Francesco Locatello. Multi-view causal representation learning with partial observability. arXiv preprint arXiv:2311.04056, 2023.

(Yao et al, 2022b) Weiran Yao, Yuewen Sun, Alex Ho, Changyin Sun, and Kun Zhang. Learning temporally causal latent
processes from general temporal data. In International Conference on Learning Representations, 2022b.

(Yao et al, 2022a) Weiran Yao, Guangyi Chen, and Kun Zhang. Temporally disentangled representation learning. In Advances
in Neural Information Processing Systems, 2022a.

(Song et al, 2023) Xiangchen Song, Weiran Yao, Yewen Fan, Xinshuai Dong, Guangyi Chen, Juan Carlos Niebles, Eric Xing,
and Kun Zhang. Temporally disentangled representation learning under unknown nonstationarity. In
Advances in Neural Information Processing Systems, 2023.

(Zhang et al, 2023b) Jiaqi Zhang, Kristjan Greenewald, Chandler Squires, Akash Srivastava, Karthikeyan Shanmugam, and Caroline Uhler. Identifiability guarantees for causal disentanglement from soft interventions. In Advances in
Neural Information Processing Systems, 2023b.

(Varici et al, 2023a) Burak Varıcı, Emre Acarturk, Karthikeyan Shanmugam, Abhishek Kumar, and Ali Tajer. Score-based causal
representation learning with interventions. arXiv preprint arXiv:2301.08230, 2023a.

(Varici et al, 2023b) Burak Varıcı, Emre Acartürk, Karthikeyan Shanmugam, and Ali Tajer. Score-based causal representation
learning from interventions: Nonparametric identifiability. In Causal Representation Learning Workshop
at NeurIPS 2023, 2023b.

---

### Decision · Action_Editor_FMaS · 2024-05-02

**Recommendation:** Accept as is

**Comment:**

This paper presents a technical survey on causal generative modeling, focusing on causal representation learning and controllable counterfactual generation methods. Specifically, the authors categorize causal representation learning methods according to the type of data, and controllable counterfactual generation methods according to the type of used generative model, respectively. Then, they discuss the common metrics, datasets, applications, and future research directions.

All reviewers acknowledged this survey is informative, comprehensive, and well-written. Also, the authors successfully addressed reviewers’ comments during the revision, and all of the reviewers agreed to the acceptance. Therefore, AE recommends a clear acceptance of this paper.

**Audience:**

Yes

**Claims And Evidence:**

Yes

---

> ### Author Response · Authors · 2024-05-22
> **Camera-Ready**
>
> We would like to thank the AE and reviewers for the great feedback that helped improve our paper. The camera-ready version of the paper has now been uploaded.